*Resource*

# The dynamics and functional impact of tRNA repertoires during early embryogenesis in zebrafish

Madalena M Reimão-Pinto [iD][1✉], Andrew Behrens [iD][2,4], Sergio Forcelloni [iD][2,4], Klemens Fröhlich [iD][1], Selay Kaya [iD][2] & Danny D Nedialkova [iD][2,3✉]

## Abstract

Embryogenesis entails dramatic shifts in mRNA translation and turnover that reprogram gene expression during cellular proliferation and differentiation. Codon identity modulates mRNA stability during early vertebrate embryogenesis, but how the composition of tRNA pools is matched to translational demand is unknown. By quantitative profiling of tRNA repertoires in zebrafish embryos during the maternal-to-zygotic transition, we show that zygotic tRNA repertoires are established after the onset of gastrulation, succeeding the major wave of zygotic mRNA transcription. Maternal and zygotic tRNA pools are distinct, but their reprogramming does not result in a better match to the codon content of the zygotic transcriptome. Instead, we find that an increase in global translation at gastrulation sensitizes decoding rates to tRNA supply, thus destabilizing maternal mRNAs enriched in slowly translated codons. Translational activation and zygotic tRNA expression temporally coincide with an increase of TORC1 activity at gastrulation, which phosphorylates and inactivates the RNA polymerase III repressor Maf1a/b. Our data indicate that a switch in global translation, rather than tRNA reprogramming, determines the onset of codon-dependent maternal mRNA decay during zebrafish embryogenesis.

**Keywords** tRNA; Translation Regulation; Maternal-to-zygotic Transition; TORC1; Zebrafish
**Subject Categories** Development; RNA Biology

## Introduction

Embryonic development is a dynamic period that encompasses cellular proliferation and fate acquisition. After fertilization, metazoan embryos are transcriptionally quiescent and rely entirely on maternal factors to drive embryogenesis. Developmental control is then handed over to the zygotic genome during the maternal-to-zygotic transition (MZT). Maternally deposited mRNAs are cleared and zygotic mRNAs are de novo synthesized by RNA Polymerase II (Pol II) in a temporally coordinated manner during the MZT (Despic and Neugebauer, 2018; Vastenhouw et al, 2019; Yartseva and Giraldez, 2015). Apart from changes in mRNA synthesis and degradation, developmental transitions are accompanied by dramatic shifts in global mRNA translation (Saba et al, 2021; Teixeira and Lehmann, 2019). Embryogenesis entails a gradual transition to a translationally active state (Bachvarova and De Leon, 1977; Brandis and Raff, 1978; Chassé et al, 2018; Danilchik and Hille, 1981; Kronja et al, 2014; Leesch et al, 2023; Winkler et al, 1985; Woodland, 1974; Xiong et al, 2022; Zhang et al, 2022; Zhu et al, 2022), suggesting that global protein synthesis and differentiation are coordinated during early development (Buszczak et al, 2014; Saba et al, 2021).

Transfer RNAs (tRNAs) are central to the process of translating mRNA into proteins, but how their expression is regulated during embryogenesis is largely unknown. The availability of these short (~76 nt) non-coding RNAs, which are synthesized by RNA Polymerase III (Pol III), is critical for decoding speed, as codons matching lowly abundant tRNAs are translated more slowly in yeast and human cells (Gao et al, 2024; Wu et al, 2019a). While the bulk of Pol II-dependent zygotic mRNA expression occurs around 3 h post-fertilization (hpf) in zebrafish embryos (Aanes et al, 2011; Baia Amaral et al, 2024; Bhat et al, 2023; Chan et al, 2019; Harvey et al, 2013; Heyn et al, 2014; Lee et al, 2013; Mathavan et al, 2005), the timing of activation of tRNA synthesis by Pol III remains unclear (Heyn et al, 2014). A subset of maternal mRNAs is degraded in a translation-dependent manner based on their codon composition during the zebrafish MZT (Bazzini et al, 2016; Medina-Muñoz et al, 2021; Mishima et al, 2022; Mishima and Tomari, 2016). Whether the temporal onset of this process is due to changes in tRNA abundance and decoding speed, and how tRNA supply and codon demand evolve during the MZT is unknown.

The complexity of zebrafish tRNA pools has so far precluded their accurate quantification. The zebrafish nuclear genome contains 8676 predicted tRNA genes, which can potentially yield 3157 mature tRNA transcripts from 47 tRNA anticodon families (Chan and Lowe, 2016). The ubiquitous chemical modifications of tRNA molecules can block reverse transcription or lead to nucleotide misincorporations during cDNA synthesis (Motorin et al, 2007). This results in cDNA libraries with a large proportion

[1]Biozentrum, University of Basel, 4054 Basel, Switzerland. [2]Mechanisms of Protein Biogenesis Laboratory, Max Planck Institute of Biochemistry, 82152 Martinsried, Germany. [3]Technical University of Munich, TUM School of Natural Sciences, Department of Bioscience, 85748 Garching, Germany. [4]These authors contributed equally: Andrew Behrens, Sergio Forcelloni. ✉E-mail: madalena.pinto@unibas.ch; nedialkova@biochem.mpg.de

of reads that are truncated or contain multiple mismatches to the genomic reference. Combined with the inordinately high (>98%) sequence identity among many tRNAs (Chan and Lowe, 2016), this often leads to incorrect or unsuccessful sequencing read alignment (Behrens et al, 2021; Padhiar et al, 2024; Sas-Chen and Schwartz, 2019). Failure to account for these complexities during library construction or data analysis can substantially impact the accuracy and resolution of tRNA abundance measurements (Behrens et al, 2021). Accordingly, 23 of the 47 zebrafish tRNA anticodon families were not resolved in a recent study (Rappol et al, 2024), precluding the analysis of their regulation and functional impact during embryogenesis.

To address the challenges inherent to tRNA quantification, we recently developed modification-induced misincorporation tRNA sequencing (mim-tRNAseq). This method combines a workflow for efficient full-length cDNA library construction from tRNAs by the bacterial group II-intron-derived enzyme TGIRT that retains modification signatures (Katibah et al, 2014; Mohr et al, 2013; Zheng et al, 2015) with a computational pipeline tailored to the complexity of the resulting sequencing data (Behrens and Nedialkova, 2022; Behrens et al, 2021). These improvements have enabled the accurate quantification of tRNA levels and modification status with single-transcript resolution in cells from diverse eukaryotes (Behrens et al, 2021; Gao et al, 2024). Here, we used mim-tRNAseq to probe the dynamics of tRNA repertoires during early zebrafish embryogenesis, and combined these measurements with recent improvements to ribosome profiling (Wu et al, 2019a) to establish the functional impact of tRNA pools on mRNA decoding rates and stability. We find that the bulk of zygotic tRNA expression is established as the embryo enters gastrulation (~5 hpf), succeeding the major wave of zygotic mRNA transcription at ~3 hpf. Although we find that a subset of tRNA anticodon families differ significantly in abundance between maternal and zygotic pools, these differences are not sufficient to appreciably alter codon translation rates of endogenous mRNAs. Instead, maternal mRNAs with a codon content poorly matched to tRNA pools are destabilized by slower decoding upon a global increase in protein synthesis at the onset of gastrulation. Global translational activation and tRNA gene derepression temporally coincide with an increase in target of rapamycin complex 1 (TORC1) signaling during the MZT, which phosphorylates the Pol III repressor Maf1a/b to render it inactive (Michels et al, 2010; Shor et al, 2010). Collectively, our data indicate that a switch in global translational activity, rather than a switch in codon optimality driven by changes in tRNA anticodon pools, underlies the timing of translation-dependent maternal mRNA destabilization during zebrafish embryogenesis.

## Results

### Quantitative profiling of tRNA repertoires during early zebrafish embryogenesis

During the zebrafish MZT, the maternal mRNA pool is gradually replaced by zygotic transcripts synthesized upon zygotic genome activation (ZGA). To establish how tRNA pools evolve in this developmental time-frame, we generated mim-tRNAseq libraries after size-selecting 60–100 nt RNA from unfertilized eggs (0 hpf), and consecutive blastula (256-cell, 2.5 hpf; 1000-cell, 3 hpf; sphere,

4 hpf) and gastrula (shield, 6 hpf; bud, 10 hpf) embryonic stages (Kimmel et al, 1995) (Fig. 1A). We did not analyze time-points beyond the end of gastrulation (10 hpf) as embryos already contain many distinct cell types (Farrell et al, 2018; Wagner et al, 2018), complicating the interpretation of data obtained by bulk sequencing approaches.

We next leveraged recent improvements to the mim-tRNAseq computational pipeline (Behrens and Nedialkova, 2022) to analyze the resulting tRNA-derived sequencing libraries. We first built a reference of mature tRNA sequences (with appended 3′-CCA, 5′-G for tRNA-His, and without introns) from the 8676 high-scoring zebrafish reference tRNA genes in the current zebrafish genome assembly (GRCz11) we obtained from the Genomic tRNA Database (Chan and Lowe, 2016) and the 22 mitochondrially encoded (mt-) zebrafish tRNAs. To minimize read multi-mapping due to high sequence similarity, we clustered tRNAs within each anticodon family by a sequence identity (ID) threshold while maintaining separation between nuclear-encoded and mt-tRNAs. Alignment to the representative cluster parents substantially reduces multi-mapping of reads to nearly identical tRNAs while retaining full resolution for all tRNA anticodon families (Behrens and Nedialkova, 2022; Behrens et al, 2021). A cluster ID of 1, corresponding to the common approach of collapsing identical tRNA genes into a single reference, resulted in >17% multi-mapped reads for tRNA libraries from unfertilized eggs (Fig. EV1A). Multi-mapping was markedly reduced (to 5.6%) at a cluster ID of 0.93 (Fig. EV1A), which we used for the rest of our analyses. To further resolve individual tRNA transcripts, we used a cluster deconvolution algorithm that reassigns reads to unique tRNAs based on mismatches to the cluster parent sequence (Behrens and Nedialkova, 2022).

Readthrough by TGIRT results in highly specific misincorporation signatures at eight Watson-Crick face modifications common in tRNAs: N1-methyladenosine (m$^1$A), N1-methylguanosine (m$^1$G), N2,N2-dimethylguanosine (m$^{2,2}$G), N3-methylcytosine (m$^3$C), inosine (I), N1-methylinosine (m$^1$I), wybutosine (yW), and N3-(3-amino-3-carboxypropyl)-uridine (acp$^3$U) (Behrens et al, 2021; Pan, 2018). Nearly all tRNAs carry one or more of these modifications, which are necessary for tRNA folding, stability, and accurate and efficient mRNA decoding (Biela et al, 2023; Suzuki, 2021). As a result, sequencing reads derived from mature tRNAs contain multiple mismatches to the genomic reference. Since there are no modifications annotated in zebrafish tRNAs so far (Cappannini et al, 2024), we used the read alignment algorithm GSNAP (Wu and Nacu, 2010) in SNP-tolerant mode for two successive rounds of alignment. In the first round, we detected potentially modified sites by a mismatch rate of >10% and the presence of a specific misincorporation signature (Behrens et al, 2021), and indexed these newly annotated sites in the reference. We then re-aligned all reads to the indexed reference with a more stringent mismatch tolerance outside of potentially modified sites, which we have shown substantially improves alignment efficiency and accuracy (Behrens et al, 2021). Using this approach, we obtained 80–84% uniquely mapped and <6.7% multi-mapped reads across early zebrafish embryogenesis (Fig. 1B). Our subsequent analysis using all uniquely mapped reads provided experimental evidence for the expression of all 47 tRNA anticodon families and 607 of the 2528 unique transcripts we resolved (>0.005% of tRNA-mapped reads in at least one sample; Dataset EV1). Seventy-six of the remaining 652 non-resolved tRNA transcripts (12%) did not pass this expression

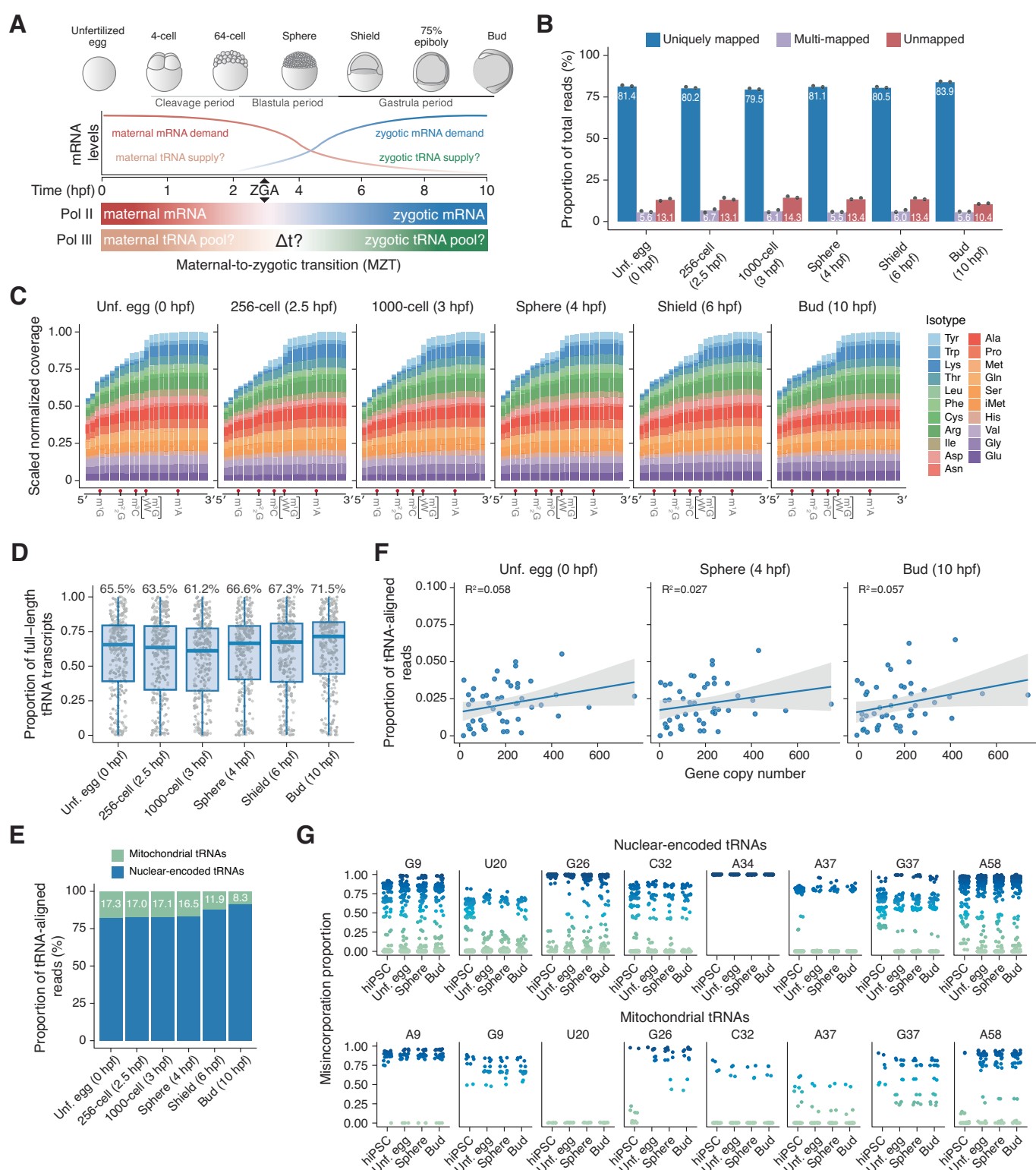

threshold in any sample, while 464 (71%) could not be deconvoluted from the cluster parent because they differ only at positions that may also carry misincorporation-inducing modifications. In such cases, a mismatch due to misincorporation at a modified site is indistinguishable from a mismatch between cluster

members (Behrens and Nedialkova, 2022). All uniquely mapped reads (regardless of deconvolution status) were included in quantitative analyses of tRNA anticodon families.

In line with the high proportion of modification readthrough in the mim-tRNAseq workflow (>95%) (Behrens et al, 2021), we

**Figure 1.  Characterization of zebrafish tRNA abundance and modification status during the maternal-to-zygotic transition.**

(A) Schematic of zebrafish embryogenesis from the unfertilized egg to bud stages of development (0–10 hours post-fertilization, hpf), during which maternal mRNAs (red) are gradually replaced by zygotic mRNAs (blue) with the onset of zygotic genome activation (ZGA). The timing (Δt) of bulk zygotic tRNA gene expression activation, the composition of maternal (beige) and zygotic (green) tRNA pools, and the relationship between tRNA demand and supply are unknown. (B) Alignment statistics for mim-tRNAseq datasets of zebrafish samples at the unfertilized egg (0 hpf), 256-cell (2.5 hpf), 1000-cell (3 hpf), sphere (4 hpf), shield (6 hpf) and bud (10 hpf) stages. Bars represent the average proportions of mapped reads ($n = 2$ for each stage). (C) Metagene analysis of scaled sequence coverage across nuclear-encoded tRNA isotypes by differences between 3′ and 5′ coverage, for each developmental time-point sequenced ($n = 1$). The major known barriers to reverse transcription during tRNA library generation are indicated in red. (D) Box plots of full-length fraction per tRNA transcript in datasets from (B); center line and label correspond to the median ($n = 2$), box limits correspond to upper and lower quartiles and whiskers represent 1.5 × interquartile range. (E) Alignment statistics for mitochondrial tRNAs (mt-tRNAs) and nuclear-encoded tRNAs at each developmental stage. (F) Correlation plots for uniquely aligned read proportions for each tRNA anticodon family and the corresponding unique tRNA gene copy number from the unfertilized egg, sphere, and bud stages ($n = 1$). Blue lines represent the linear regression model; shaded gray depicts the 95% confidence interval. (G) Misincorporation proportions per canonical nucleotide position and identity in zebrafish embryonic tRNA pools ($n = 2$) in comparison to tRNA pools from human induced pluripotent stem cells (hiPSC) ($n = 2$, data from Gao et al (2024)).

obtained uniform tRNA sequence coverage from zebrafish embryonic tRNA pools (Fig. 1C). A median of 61–72% uniquely mapped reads from nuclear-encoded tRNAs were full-length (Fig. 1D), and nearly all (99%) contained the post-transcriptionally added 3′-CCA sequence (Fig. EV1B), indicating they are derived from mature tRNAs.

The proportion of reads aligning to the 22 mt-tRNAs gradually decreased from 17.3% in unfertilized eggs (0 hpf) to 8.3% at the bud stage (10 hpf) (Fig. 1E). These data agree with the comparatively higher number of mitochondrial DNA molecules present in mature zebrafish oocytes (Otten et al, 2016), and support the notion of baseline transcriptional activity of the mitochondrial genome immediately after fertilization (Bhat et al, 2023; Heyn et al, 2014). The decrease in the proportion of mt-tRNAs at the shield and bud stages (6 and 10 hpf, respectively) could indicate a relative increase in nuclear-encoded tRNA levels throughout MZT, or reflect a decrease in the number of mitochondria per cell as embryonic development progresses (Van Blerkom, 2009).

Gene copy number is a good predictor of tRNA abundance in yeast (Behrens et al, 2021; Jacob et al, 2019; Tuller et al, 2010), but less so in metazoans (Behrens et al, 2021). The proportion of reads uniquely aligned to each zebrafish tRNA anticodon family did not significantly correlate with tRNA gene copy number at any stage before or after ZGA (adjusted $R^2 = 0.027$–$0.058$, Figs. 1F and EV1C). In human cells, inactive tRNA genes are marked by near-complete CpG methylation (Gao et al, 2024). Analysis of published datasets from whole-genome bisulfite sequencing (Jiang et al, 2013) revealed that a large proportion of CpG in predicted zebrafish tRNA genes are highly methylated in both 1000-cell (3 hpf) and germ-ring (5.7 hpf) embryos (Fig. EV1D). CpG methylation was near-complete for tRNA genes encoded in large clusters on chromosomes 4 (5227 tRNA genes, 60%), 8 (353 genes, 4%), and 3 (261 genes, 3%). These data indicate that metrics relying on tRNA gene copy number to estimate tRNA abundance like the tRNA adaptation index (tAI) (dos Reis et al, 2004; Sabi and Tuller, 2014) do not reflect intracellular tRNA levels during zebrafish embryogenesis.

We next leveraged TGIRT misincorporation signatures to map modifications in zebrafish tRNAs. We used a mismatch frequency threshold of >10% to identify potentially modified sites in tRNAs with a coverage of >0.01% of uniquely mapped reads. Modification type was predicted by combining canonical tRNA position, reference nucleotide type, and misincorporation signature in comparison with known Watson-Crick face-modified sites in human tRNAs (Behrens et al, 2021). We annotated 327

modifications in nuclear-encoded tRNAs and mt-tRNAs, and we predicted the identity of 312 of these (95%, Dataset EV1). We have previously shown that misincorporation proportions in mim-tRNAseq datasets also reflect modification stoichiometry (Behrens et al, 2021), so we asked how these compare between zebrafish and human tRNAs. Upon examining the most conserved modified tRNA sites (9, 20, 26, 32, 34, 37, and 58) in nuclear-encoded and mt-tRNAs, we found that the majority (77%) of zebrafish mt-tRNAs display high misincorporation proportions (>0.7) at G26 and A58, which are modified to $m^{2,2}G$ and $m^1A$, respectively. By contrast, G26 and A58 are largely unmodified in mt-tRNAs from human induced pluripotent stem cells (hiPSC) (Fig. 1G). These variations may reflect a larger sequence divergence among zebrafish and human mt-tRNAs than among nuclear-encoded tRNAs, or unique substrate specificities of the enzymes that install these modifications in the two organisms. While $m^1A58$ deposition in nuclear-encoded and mt-tRNAs is catalyzed by distinct enzymes, $m^{2,2}G26$ is deposited by a common one: TRMT1 (de Crécy-Lagard et al, 2019). Indeed, misincorporation rates at G26 were also more distinct among nuclear-encoded human and zebrafish tRNAs, with zebrafish profiles displaying a larger range of stoichiometries in contrast to the clear separation of tRNAs with unmodified or ~100% modified $m^{2,2}G26$ (Fig. 1G). We also found variations in $m^1G37$, which is installed by TRMT5 (Powell et al, 2015). There was no detectable misincorporation at G37 in nuclear-encoded zebrafish tRNA-His-GUG transcripts (Fig. 1G; Dataset EV1), while human tRNA-His-GUG molecules carry this modification (Behrens et al, 2021). These data indicate that there are differences in substrate specificity between human TRMT1 and TRMT5 and their zebrafish homologs.

## Zebrafish maternal and zygotic tRNA pools are distinct

We next asked whether the composition of zebrafish tRNA pools changes significantly during embryogenesis. This could result from altered levels of individual tRNA isodecoders (transcripts that share the same anticodon sequence but have distinct tRNA body sequences, Goodenbour and Pan, 2006) (Fig. 2A). If the impacted isodecoders are sufficiently abundant, their altered levels could change the abundance of the tRNA anticodon family to which they belong. We therefore performed DESeq2 (Love et al, 2014) to quantify the variation between mim-tRNAseq datasets from distinct developmental time-points at the unique tRNA transcript and tRNA anticodon level. Principal component (PC) analysis for

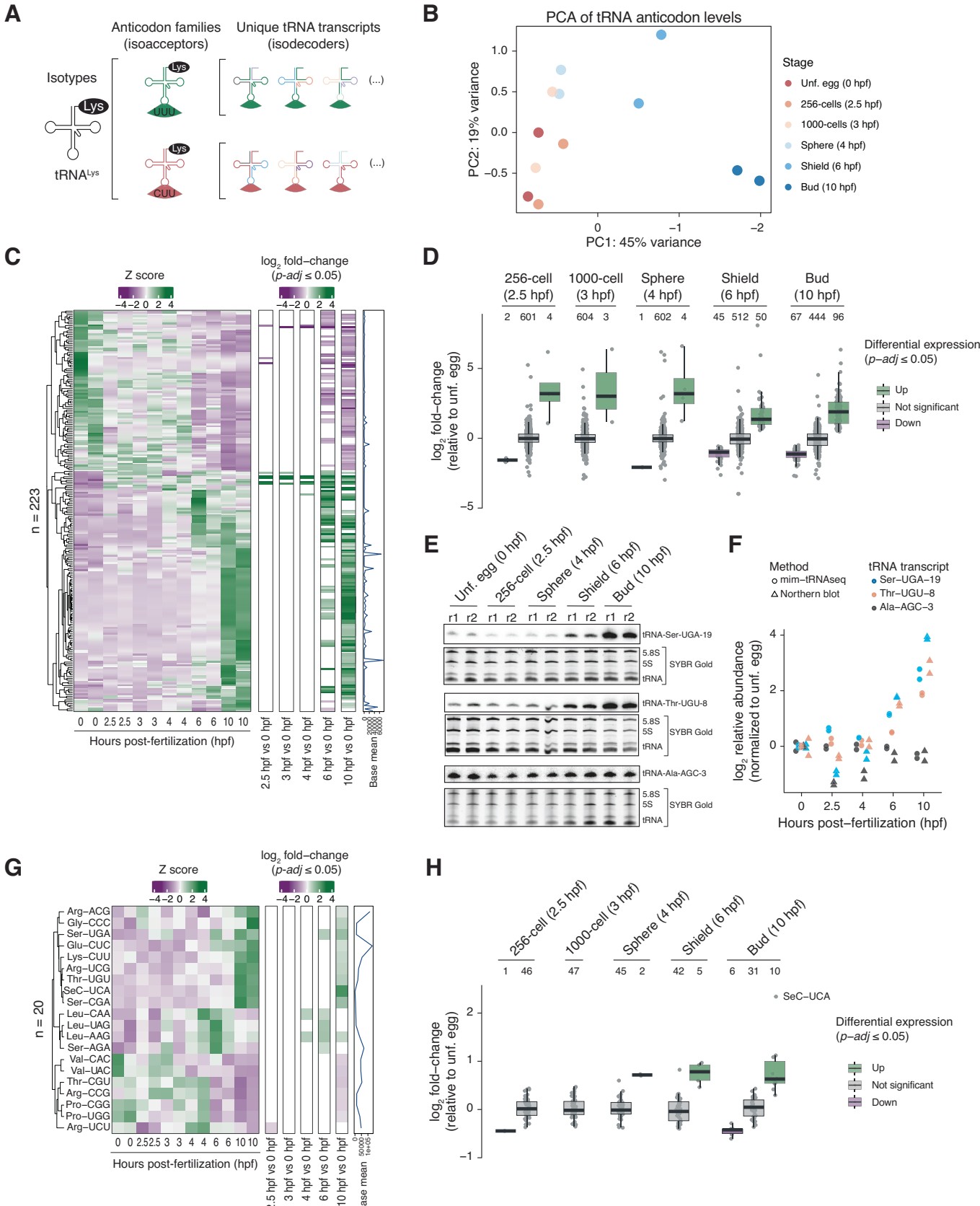

**Figure 2. The zebrafish maternal and zygotic tRNA pools are distinct.**

(A) Schematic of tRNA classification into isoacceptor families with multiple isodecoders in each family. (B) Principal component analysis (PCA) plot of normalized tRNA anticodon levels for all 12 mim-tRNAseq samples. (C) Left panel: hierarchically clustered expression heatmaps showing scaled z score of normalized unique tRNA transcript ($n = 223$) counts in all mim-tRNAseq samples, arranged by chronological developmental time. Right panel: differential expression relative to the unfertilized egg (0 hpf). Values represent log2 fold-changes estimated by DESeq2 ($n = 2$ for each time-point, p-values from Benjamini–Hochberg-adjusted Wald test). (D) Box plots of log2 fold-changes in unique tRNA transcript levels relative to the unfertilized egg (0 hpf) time-point. Only tRNA transcripts with detectable expression in at least one developmental stage (≥0.005% of tRNA-mapped reads) separated by significant up-regulation (green), significant down-regulation (purple), or no significant change (gray) (Benjamini–Hochberg-adjusted Wald test; p-adj ≤ 0.05) are plotted. Center line and label: median; box limits: upper and lower quartiles; whiskers: 1.5 × interquartile range. (E) Northern blot analysis of tRNA-Ser-UGA-19, tRNA-Thr-UGU-8, and tRNA-Ala-AGC-3 during embryogenesis ($n = 2$, matched total RNA samples to those used for mim-tRNAseq); r1: replicate 1, r2: replicate 2. Gels stained with SYBR Gold prior to transfer are also shown. (F) Relative abundances of tRNA-Ser-UGA-19, tRNA-Thr-UGU-8, and tRNA-Ala-AGC-3 measured by northern blot (quantified by densitometry; rectangles) side-by-side to tRNA abundances measured by mim-tRNAseq (dots), normalized to the respective mean value for the unfertilized egg stage (0 hpf). (G) Differential expression analysis as in (C) for counts per tRNA anticodon family. (H) Box plots of log2 fold-changes in tRNA anticodon family as in (D). Source data are available online for this figure.

the 607 fully deconvoluted unique tRNA transcripts with detectable expression, as well as for the 47 zebrafish tRNA anticodon families, revealed high reproducibility between biological replicates from the same time-point (Figs. 2B and EV2A). Surprisingly, the first PC accounted for 38% to 45% of the variance in unique transcript or tRNA anticodon levels, indicating that maternal and zygotic tRNA pools are distinct. 223 of the 607 fully resolved tRNA transcripts (37%) were differentially expressed (p-adj ≤ 0.05, Benjamini–Hochberg-adjusted Wald test) at one or more stages of zebrafish embryogenesis in comparison to unfertilized eggs (Fig. 2C). There were only 3 to 6 transcripts (0.5–1%) with differential abundance at early ZGA stages (2.5, 3 and 4 hpf), and the majority of changes were detected after the onset of gastrulation (6 and 10 hpf) (Fig. 2C,D). Notably, the magnitude of changes for upregulated tRNA transcripts (up to ~64-fold) was much larger than for downregulated ones (up to ~4-fold) (Fig. 2D). The differences in abundance we measured by mim-tRNAseq were highly concordant with northern blotting analysis we performed for three transcripts as a validation: the levels of tRNA-Ser-UGA-19 and tRNA-Thr-UGU-7 increased at the onset of gastrulation, while tRNA-Ala-AGC-3 levels did not increase (Fig. 2E,F). Taken together, these data suggest that zygotic tRNA expression in zebrafish embryos ensues at the onset of gastrulation, in agreement with findings obtained by complementary methods (Chen et al, 2021; Rappol et al, 2024).

When uniquely tRNA-mapped reads were aggregated by anticodon (before analysis), we found that 20 out of the 47 tRNA anticodon families are differentially expressed at one or more stages during (2.5, 3, and 4 hpf) or after ZGA (6 and 10 hpf) in comparison to unfertilized eggs (p-adj ≤ 0.05; Fig. 2G). In contrast to the large magnitude of expression changes from some unique tRNA transcripts (Fig. 2D), the levels of tRNA anticodon families varied by <2-fold during embryogenesis (Fig. 2H). An exception to this was tRNA-Sec-UCA, which inserts selenocysteine at specific UGA codons and was upregulated by 4-fold in bud (10 hpf) stage embryos (Fig. 2H). Clustering analysis using Mfuzz (Kumar and Futschik, 2007) identified three groups of tRNAs with distinct expression dynamics both at the unique tRNA transcript and tRNA anticodon family level during embryogenesis (Fig. EV2B,C; Dataset EV2): a cluster for which tRNA levels decrease (cluster 1), another for which tRNA levels increase (cluster 2), and a third for which tRNA levels remain relatively unchanged throughout embryogenesis (cluster 3). For cluster 2, the increase in expression levels is observed from the shield stage (6 hpf) onwards. Collectively, these data indicate that the bulk of zygotic tRNA expression ensues at the onset of gastrulation, shortly after robust zygotic Pol II-

mediated gene expression has been established (~3 hpf) (Baia Amaral et al, 2024; Bhat et al, 2023; Harvey et al, 2013; Heyn et al, 2014; Lee et al, 2013), and that zygotic tRNA expression reshapes zebrafish embryonic tRNA pools while maintaining tRNA anticodon levels largely stable.

We next investigated whether the stoichiometry of misincorporation-inducing modifications in zebrafish tRNAs varies significantly between stages of embryogenesis before and after gastrulation. To test this, we calculated log odds ratios of misincorporation proportions across tRNA positions we predicted as modified in nuclear-encoded tRNAs (Dataset EV1) at the end of gastrulation (bud, 10 hpf) in comparison to blastula (sphere, 4 hpf) (Fig. EV2D). The stoichiometry of the vast majority of misincorporation-inducing tRNA modifications did not differ significantly (Chi-square FDR-adjusted $p \leq 0.01$), indicating that most zebrafish tRNAs are modified to a similar extent in maternal and zygotic tRNA pools. Twenty-two tRNA transcripts and two unsplit clusters, however, exhibited significantly lower levels of $m^1G9$ (three tRNA transcripts and one unsplit cluster), $m^1A14$ (two transcripts), $m^{2,2}G26$ (nine transcripts and one unsplit cluster), or $m^1A58$ (eight transcripts) (Dataset EV1). Most of these changes were of a small magnitude, or occurred in tRNAs with an overall low modification stoichiometry where these chemical groups may be less functionally important (Figs. EV2E and 1G). The transcript-specific decreases in modification levels did not correlate with a selective reduction in the expression of the enzymes predicted to deposit them, which we measured by RNA-Seq of sphere (4 hpf) and buf (10 hpf) samples (Fig. EV2F). In fact, the mRNA levels of all enzymes that deposit misincorporation-inducing tRNA modifications were lower at the end of gastrulation (Fig. EV2F). However, more than half (13/24) of the tRNAs with reduced modification stoichiometry increased significantly in abundance at the bud stage (10 hpf) in comparison to sphere (4 hpf) (Fig. EV2G). These data suggest that a subset of newly made tRNAs is disproportionately sensitive to decreased expression of tRNA-modifying enzymes at the end of zebrafish gastrulation.

## An increase in global translation at gastrulation sensitizes decoding rates to tRNA supply

We next investigated the functional impact of zebrafish tRNA repertoires on decoding rates. Most ribosomes in the early zebrafish embryo (1 hpf) are in a translationally inactive state and global translation gradually increases during the MZT, with a clear increase in translational efficiency at 6 hpf (Leesch et al, 2023). Accordingly, we

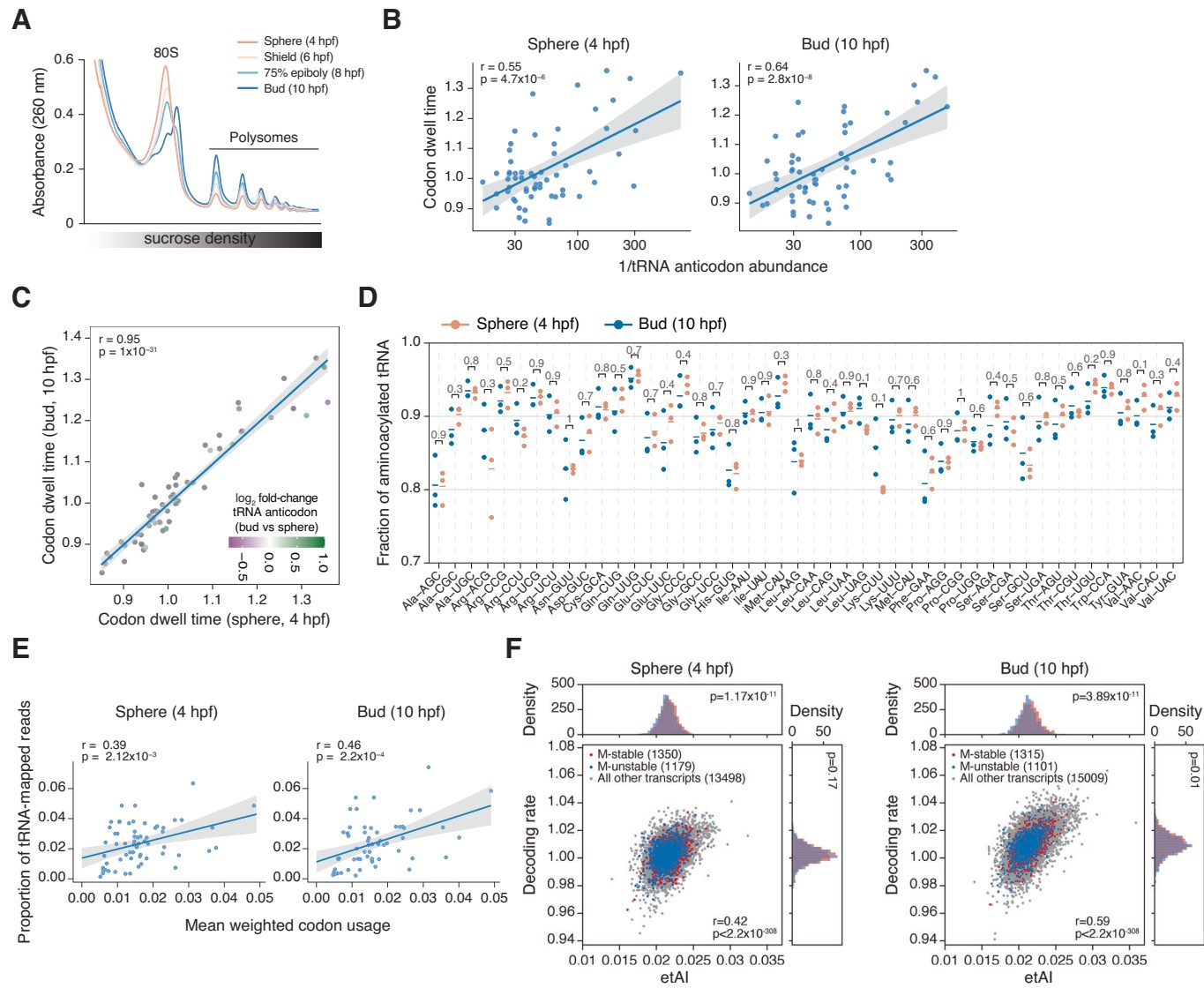

**Figure 3. An increase in global translation at gastrulation sensitizes decoding rates to tRNA supply.**

(A) Polysome profiles of pooled zebrafish embryos at the sphere (4 hpf), shield (6 hpf), 75% epiboly (8 hpf) and bud (10 hpf) stages of embryogenesis. (B) Correlation plot of codon-specific dwell times calculated with Scikit-ribo and cognate tRNA levels determined by mim-tRNA seq for the sphere (4 hpf) and bud (10 hpf) stages. Pearson's *r* values and their associated *p*-values are shown. (C) Correlation plot of codon dwell times at the sphere (4 hpf) and bud (10 hpf) stages of development. Codons are colored according to significant log2 fold-changes in matching tRNA anticodon abundance in bud (10 hpf) versus sphere (4 hpf) time-point estimated by DESeq2 (Benjamini–Hochberg-adjusted Wald test, p-adj ≤ 0.05). (D) Dot plot of aminoacylated tRNA fractions per anticodon family at the sphere (4 hpf) and bud (10 hpf) stages (*n* = 3 for each stage; line: mean; *p*-values from unpaired t-test). tRNA anticodons and corresponding amino acid are indicated. (E) Correlation plots of mean codon usage to mean tRNA anticodon abundance scaled to the proportions of total tRNA-mapped reads at the sphere (4 hpf) and bud (10 hpf) stages (*n* = 2 for each stage). Pearson's *r* values and their associated p-values are shown. (F) Correlation plots of expression-based tRNA adaptation index (etAI) values and decoding rates at the sphere (4 hpf) and bud (10 hpf) stages for all detectable mRNA transcripts (mean TPM > 0.5 in RNA-Seq, *n* = 16,027 at 4 hpf, *n* = 17,425 at 10 hpf). Red and blue dots represent maternal (M)-stable and M-unstable transcripts, respectively (Bhat et al, 2023); gray dots denote all other detectable transcripts. Marginal density histograms for transcript decoding rates calculated from matched ribosome profiling data and etAI values are shown for M-stable and M-unstable transcripts (*p*-values from a Mann–Whitney U test). Source data are available online for this figure.

detected an increase in polysomal fractions as embryos proceed through gastrulation (Fig. 3A), suggesting that global translation and tRNA expression increase concurrently during zebrafish embryogenesis. To determine the relationship between tRNA abundance and codon translation rates, we generated matched ribosome profiling datasets (Ingolia et al, 2009) of embryos undergoing bulk ZGA (sphere, 4 hpf) and when the zygotic transcriptome has been established (bud,

10 hpf). During mRNA decoding, ribosomes with open A-sites accumulate at codons matching lowly abundant tRNA species in yeast and human cells (Gao et al, 2024; Weinberg et al, 2016; Wu et al, 2019a). To recover footprints from such pre-tRNA accommodation ribosomes, we flash-froze zebrafish embryos in liquid nitrogen and lysed them in the presence of cycloheximide and tigecycline. This cocktail of elongation inhibitors yields ribosome profiling data that

faithfully recapitulates the inverse relationship between tRNA abundance and codon translation rates in yeast (Wu et al, 2019a) and human cells (Gao et al, 2024). Nearly all footprints we obtained were 29–34 nt in length (Fig. EV3A) and displayed the three-nucleotide periodicity that is a hallmark of active translation (Ingolia et al, 2009) (Fig. EV3B). We then identified the A-site codon in each coding sequence-mapped ribosome footprint with Scikit-ribo (Fang et al, 2018) and examined the relationship between relative A-site codon dwell time (DT) and matching tRNA anticodon abundance at the same developmental stage. Codon DTs were significantly anti-correlated with cognate tRNA levels at both 4 hpf (Pearson's $r = 0.55$, $p = 4.7 \times 10^{-6}$) and 10 hpf (Pearson's $r = 0.64$, $p = 2.8 \times 10^{-8}$) stages (Fig. 3B; Dataset EV3), indicating that tRNA availability limits decoding speed in zebrafish embryos both before and after gastrulation onset. By contrast, the correlation between tRNA anticodon levels and codon DTs calculated from published datasets obtained without tigecycline (Chew et al, 2013) correlated much more weakly (4.3 hpf) or not significantly (10 hpf) (Fig. EV3C,D; Dataset EV3). These findings, together with recent data from human cells (Gao et al, 2024), underscore the importance of including tigecycline in ribosome profiling workflows when probing the relationship between tRNA levels and codon translation rates in eukaryotes.

We next asked if the changes in tRNA anticodon levels we detected at gastrulation (Fig. 2G) were of sufficient magnitude to alter decoding rates at their cognate codons. A-site codon DTs calculated by ribosome profiling, however, were highly concordant between bud (10 hpf) and sphere (4 hpf) embryos (Pearson's $r = 0.95$, $p = 1 \times 10^{-31}$) (Fig. 3C; Dataset EV3) and the correlation between differences in codon DT and changes in tRNA anticodon levels was weak (Pearson's $r = -0.28$, $p = 0.0018$; Fig. EV3E). Global quantitative analysis of tRNA charging at sphere (4 hpf) and bud (10 hpf) stages showed high (>80%) aminoacylation levels for all tRNA anticodon families at both stages, and no significant change in tRNA charging before and after gastrulation (Fig. 3D). These data indicate that the reprogramming of tRNA pools at gastrulation is unlikely to be an active regulatory mechanism for altering decoding rates of specific codons or compensating for poor aminoacylation during the MZT.

We then calculated the global "demand" for each tRNA anticodon by computing the global usage of each codon weighted for mRNA expression levels from sphere (4 hpf) and bud (10 hpf) RNA-seq data. Codon usage remained largely unchanged between the two stages (coefficient of variation 0.15–13.68%), as did the ratio between codon demand and tRNA anticodon supply (coefficient of variation 0.44–43.18%; Fig. EV4A; Dataset EV3). The positive correlation between codon usage and tRNA anticodon abundance was also of a similar strength (Pearson's $r = 0.39$–$0.46$; Fig. 3E) in sphere (4 hpf) and bud (10 hpf) stage embryos, indicating that tRNA pools are comparably well adapted to the codon demand of these two developmental stages. Collectively, these data indicate that the changes to tRNA repertoires after the onset of gastrulation do not render tRNA pools more suited to the embryo's codon demand.

Apart from decoding rates, codon identity also impacts mRNA stability (Wu and Bazzini, 2023). This link is also established via the ribosomal A-site, which when left vacant due to low tRNA availability can slow down translation and result in mRNA decay (Buschauer et al, 2020). A subset of maternal mRNAs is destabilized in a codon-dependent manner at later stages of the zebrafish MZT (6 hpf), an effect that depends on active translation (Bazzini et al, 2016; Medina-Muñoz et al, 2021; Mishima et al, 2022;

Mishima and Tomari, 2016). We therefore investigated whether the differential tRNA anticodon levels established during gastrulation could underlie the timing of codon-dependent maternal mRNA decay. To test this, we focused on maternal zebrafish mRNAs that are degraded with different kinetics: maternal-stable, which persist throughout MZT, and maternal-unstable, whose levels decrease from ~4 hpf onward (Bhat et al, 2023). To determine how well adapted mRNA transcripts are to the tRNA pools in sphere (4 hpf) or bud (10 hpf) stage embryos, we calculated an expression-based tAI (etAI, see Methods) based on tRNA anticodon levels measured by mim-tRNAseq at each of these developmental stages. Then, we examined the relationship between stage-dependent etAI and the decoding rate (calculated as the inverse of codon DT from ribosome profiling) of each mRNA for genes with detectable expression at the corresponding developmental stage (average TPM > 0.5 in RNA-seq; Dataset EV4). Remarkably, etAI and transcript decoding rates correlated more strongly at the bud stage (10 hpf, Pearson's $r = 0.59$, $p < 2.2 \times 10^{-308}$) than in sphere (4 hpf, Pearson's $r = 0.42$, $p < 2.2 \times 10^{-308}$; Fig. 3F). The difference in magnitude between these two correlations is statistically significant ($p < 0.001$, Fisher's tests on z-statistics (Diedenhofen and Musch, 2015)) suggesting that the global increase in translational activity at the onset of gastrulation (Fig. 3A) may render tRNA supply more limiting. This difference was much less pronounced for the codon adaptation index (CAI), a metric that defines optimal codons as those over-represented in highly expressed protein-coding genes (Sharp and Li, 1987; Zhou et al, 2009) and does not take into account tRNA abundance. CAI values we calculated from the top 5% most highly expressed genes at each developmental stage we assayed correlated similarly with transcript decoding rates at both stages (Pearson's $r = 0.44$, $p = 2.2 \times 10^{-308}$ at 4 hpf; $r = 0.5$, $p = 2.2 \times 10^{-308}$ at 10 hpf) (Fig. EV4B; Dataset EV4). CAI and etAI also had a significant positive correlation (Pearson's $r = 0.76$, $p < 10^{-4}$; Fig. EV4C), suggesting an adaptation of zebrafish tRNA pools for fast translation of codons overrepresented in highly abundant mRNAs. Maternal-stable mRNAs had significantly higher CAI and etAI values and were thus better matched to tRNA pools than maternal-unstable transcripts at both developmental stages (Figs. 3F and EV4B). Remarkably, however, maternal-unstable mRNAs were decoded significantly more slowly than maternal-stable ones only at the bud stage (10 hpf, $p = 0.01$ versus $p = 0.17$ at 4 hpf). The slower decay of maternal-stable mRNAs may thus stem in part from tRNA pools that are better suited to their codon composition throughout embryogenesis. The global increase in translation during gastrulation, on the other hand, could amplify the codon-mediated destabilization of maternal-unstable transcripts at later stages of the MZT.

## Slowly decoded codons destabilize maternal mRNAs during the zebrafish MZT

We next asked which codons contribute to the differential stability of maternally deposited zebrafish mRNAs, and how this contribution is modulated by their decoding rates and the levels of their cognate tRNAs during the MZT. To address these questions, we used the codon stabilization coefficient (CSC), a metric that represents the Pearson correlation coefficient between the occurrence of each codon in individual mRNA transcripts and their half-life (Presnyak et al, 2015). Previous studies derived zebrafish CSCs

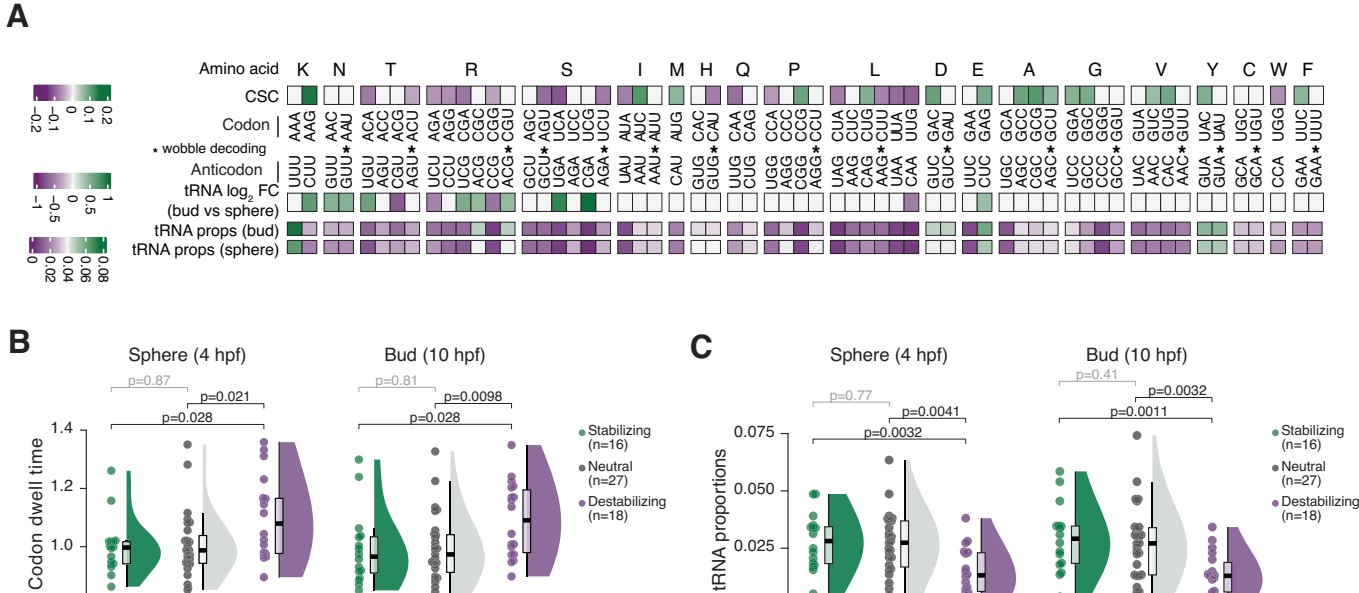

**Figure 4. Slowly decoded codons destabilize maternal mRNAs during the zebrafish maternal-to-zygotic transition as a function of tRNA levels.**

(A) Heatmaps depicting codon stabilization coefficient (CSC) scores, tRNA log2 fold-changes in anticodon supply of the bud (10 hpf) relative to sphere (4 hpf) stage, and tRNA proportions for each anticodon tRNA at both stages of development. Codon–anticodon pairs decoded by wobble base pairing are marked with *. (B) Raincloud plots of codon dwell times ($n = 1$ for each time-point) for codons scoring as stabilizing ($n = 16$), neutral ($n = 27$) and destabilizing ($n = 18$) at the sphere (4 hpf) and bud (10 hpf) stages of development. Center line of box plots: median; box limits: upper and lower quartiles; whiskers: 1.5 × interquartile range. Statistical significance (p-value) between the means was determined by a Mann–Whitney U Test. (C) Raincloud plots of tRNA proportions for codons scoring as stabilizing ($n = 16$), neutral ($n = 27$) and destabilizing ($n = 18$) at the sphere (4 hpf) and bud (10 hpf) stages of development. Center line of box plots: median; box limits: upper and lower quartiles; whiskers: 1.5 × interquartile range. Statistical analysis performed as in (B). Average tRNA proportions were calculated from mim-tRNAseq data ($n = 2$ for each time-point).

from half-life measurements of synthetic mRNA reporters (Bazzini et al, 2016; Mishima et al, 2022) or of endogenous mRNAs after transcriptional shut-off with α-amanitin (Bazzini et al, 2016; Mishima and Tomari, 2016). We calculated half-lives for maternal (stable and unstable) mRNAs from an RNA-Seq time-course of wild-type embryos in unperturbed conditions generated after rRNA depletion (Medina-Muñoz et al, 2021). This approach circumvents confounding factors from the inhibition of both Pol II and Pol III by α-amanitin (Austoker et al, 1974; Beebee and Butterworth, 1974; Benecke and Seifart, 1975; Schwartz et al, 1974; Weinmann and Roeder, 1974), as well as from differences in mRNA polyA tail length that are particularly pronounced during the zebrafish MZT (Begik et al, 2023; Subtelny et al, 2014). Importantly, this approach captured the accelerated decay of maternal-unstable transcripts from ~4 hpf onward (Fig. EV4D). Sixteen codons showed a significant stabilizing effect (CSC > 0, $p < 0.05$) and eighteen showed a significant destabilizing effect (CSC < 0, $p < 0.05$) for maternally deposited zebrafish mRNAs (Fig. 4A; Dataset EV4). Our CSC scores generally agree with previous CSC calculations in zebrafish embryos after α-amanitin transcriptional shut-off (Bazzini et al, 2016), with a significant positive correlation (Spearman's rho = 0.633, $p < 10^{-4}$) and 37 out of 61 codons displaying matching properties (Fig. EV4E,F; Dataset EV4).

Previous studies in zebrafish and human cell lines have reached conflicting conclusions about the relationship between CSC, decoding rates, and tRNA abundance (Bazzini et al, 2016; Forrest et al, 2020; Mishima et al, 2022; Wu et al, 2019b). We reasoned that this could arise from the technical challenges to tRNA

quantification by standard sequencing approaches (Padhiar et al, 2024), combined with the use of ribosome profiling datasets generated in the absence of tigecycline that do not accurately capture footprints from decoding ribosomes (Fig. EV3D; Gao et al, 2024; Wu et al, 2019a). We therefore examined the relationship between CSC and codon dwell times calculated from ribosome profiling that we performed with tigecycline (Fig. 4B). Stabilizing codons displayed significantly lower A-site codon DTs than destabilizing codons at both sphere (4 hpf) and bud (10 hpf) stages ($p < 0.05$, Mann–Whitney U Test), indicating that they are indeed decoded faster in zebrafish embryos. Accordingly, stabilizing codons were associated with significantly higher tRNA abundances than destabilizing codons at both developmental stages (Fig. 4C). Overall, CSC scores and matching tRNA anticodon levels were positively correlated (Spearman's rho = 0.448, $p = 3 \times 10^{-4}$ at 4 hpf; Spearman's rho = 0.491, $p < 10^{-4}$ at 10 hpf) (Fig. EV4G), demonstrating a relationship between tRNA abundance and codon-dependent mRNA stability effects. In line with this, maternal-unstable transcripts had a significantly higher proportion of destabilizing ($p < 2 \times 10^{-16}$, Mann–Whitney U Test) and neutral ($p = 2.9 \times 10^{-8}$, Mann–Whitney U Test) codons than maternal-stable transcripts and conversely, maternal-stable transcripts were significantly enriched in stabilizing codons ($p < 2 \times 10^{-16}$, Mann–Whitney U Test) (Fig. EV4H). Thus, tRNA levels modulate decoding speed, which in turn affects maternal mRNA stability in a codon-dependent manner during the zebrafish MZT.

Despite the clear relationship between tRNA levels, decoding speed, and CSC (Figs. 4B,C and EV4G), some codons still scored as

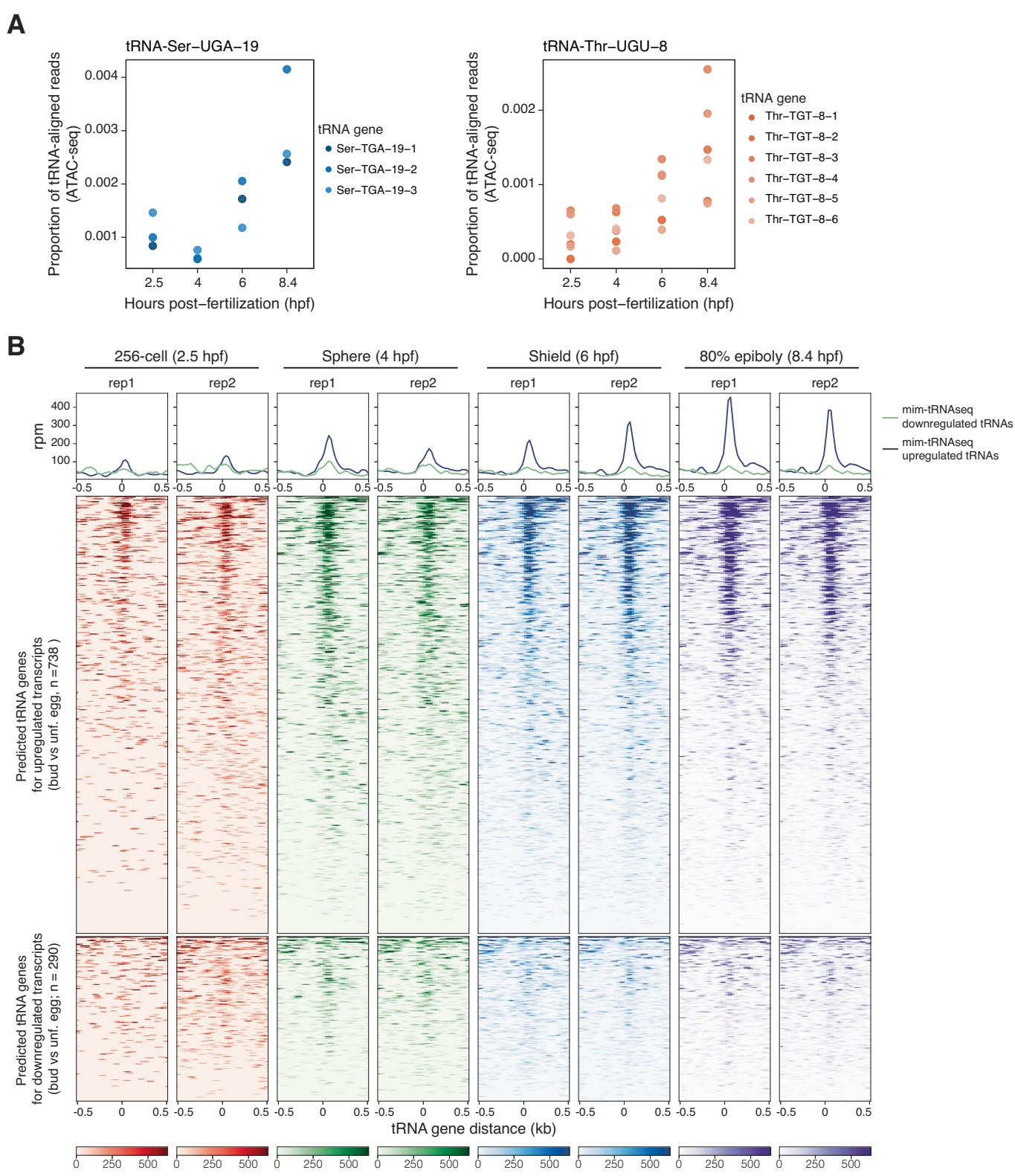

**Figure 5. Chromatin accessibility changes at tRNA genes during the zebrafish MZT.**

(A) Proportion of ATAC-seq NFR reads aligning to tRNA-Ser-TGA-19 and tRNA-Thr-TGT-8 genes at the developmental time-points specified. (B) Top: Average normalized ATAC-seq NFR signal intensity at genes encoding upregulated (n = 738) and downregulated (n = 290) tRNA transcripts in bud (10 hpf) relative to egg (0 hpf). Bottom: TSS-centered heatmaps of chromatin accessibility at upregulated and downregulated tRNA genes. Bulk ATAC-seq data from Pálfy et al (2020).

stabilizing or neutral despite comparatively low levels of their cognate tRNAs and consequently higher codon DTs in zebrafish embryos. For example, Pro-CCG and Ala-GCG scored as stabilizing (Fig. 4A; Dataset EV4), but were among the top 10% most slowly decoded codons at both sphere (4 hpf) and bud (10 hpf) stages (Dataset EV4). We asked whether this could be linked to wobble pairing during decoding, which can slow down translation in bacteria (Curran and Yarus, 1989; Sørensen and Pedersen, 1991), yeast (Letzring et al, 2010) and nematodes and HeLa cells (Stadler and Fire, 2011). To test this, we compared codon DTs for the 16 pairs of codons that are decoded by the same tRNA anticodon family by either Watson-Crick or wobble pairing in zebrafish. However, we found no significant difference between A-site ribosome occupancy in these two groups ($p = 0.49$ at 4 hpf and $p = 0.4$ at 10 hpf, Wilcoxon rank-sum test; Fig. EV4I), indicating that wobble pairing does not slow down decoding in zebrafish embryos. High GC content, however, may in some cases override the impact of tRNA levels on that codon's effect on stability: the majority of codons that scored as stabilizing in our analysis are GC3 codons (14/16), and the two remaining codons are GC-rich (GC% = 67%). Conversely, most destabilizing codons are AT3 codons (14/18) (Dataset EV4). These data are in agreement with prior findings that GC3 codons and highly structured coding sequences can stabilize reporter mRNAs and boost protein expression in trypanosomatids and mammalian cells (Hia et al, 2019; Jeacock et al, 2018; Mauger et al, 2019). Codon content can thus influence mRNA stability through both tRNA-dependent and tRNA-independent mechanisms.

## Chromatin accessibility increases at tRNA genes upregulated at the onset of gastrulation

We next investigated whether the reprogramming of tRNA repertoires following gastrulation results from selective tRNA gene expression. During the zebrafish ZGA, an increase in chromatin accessibility at promoters generally precedes Pol II transcription (Pálfy et al, 2020), and nucleosome-free region (NFR) signal measured by the assay for transposase-accessible chromatin with sequencing (ATAC-seq) positively correlates with mature tRNA abundance in human cells (Gao et al, 2024). We therefore assessed NFR signal at tRNA genes in published ATAC-seq datasets collected at consecutive stages of early zebrafish embryogenesis (Pálfy et al, 2020). We first focused on two tRNA transcripts that are strongly and significantly upregulated from the shield (6 hpf) stage onwards: tRNA-Ser-UGA-19 and tRNA-Thr-UGU-8 (Dataset EV1, Fig. 2E). The proportion of tRNA-aligned NFR reads mapped to the three tRNA Ser-TGA-19 genes and five tRNA-Thr-TGT-8 genes gradually increased from the shield (6 hpf) stage onward (Fig. 5A), matching the timing of increased tRNA-Ser-UGA-19 and tRNA-Thr-UGU-8 expression we detected by mim-tRNAseq and northern blotting (Fig. 2E,F). Globally, NFR signal gradually increased at a subset of tRNA genes encoding transcripts significantly upregulated by the end of gastrulation from the sphere stage (4 hpf) onward, whereas genes encoding downregulated tRNAs remained inaccessible throughout the MZT (Fig. 5B). The majority of the remaining zebrafish tRNA genes had no detectable NFR signal even in epiboly stage embryos (8.4 hpf) (Fig. EV5A), in line with the high degree of CpG methylation we found at most tRNA loci (Fig. EV1D) and the poor correlation between tRNA

gene copy number and mature tRNA levels (Fig. 1E). Of note, CpG methylation at tRNA genes encoding upregulated tRNA transcripts was significantly lower than that at genes encoding downregulated or unchanged tRNAs already in 1000-cell embryos (3 hpf, Fig. EV5B), suggesting that the onset of their transcription is not linked to a change in DNA methylation status. Taken together, these data indicate that most predicted tRNA genes are inactive or do not contribute substantially to mature tRNA pools in early zebrafish embryos, while tRNA gene activation at gastrulation is accompanied by chromatin remodeling.

## TORC1 activation and tRNA gene derepression coincide temporally during the zebrafish MZT

Since the time frame of global translation activation and tRNA expression both coincide with gastrulation onset in zebrafish, we asked how these two events are coordinated temporally. Both processes are controlled by the target of rapamycin complex 1 (TORC1) serine/threonine kinase in response to growth and differentiation cues (Battaglioni et al, 2022). Mammalian TORC1 (mTORC1) regulates translation initiation by phosphorylating 4E-binding proteins (4EBPs), which are inhibitors of the mRNA cap-binding protein eukaryotic initiation factor 4E (eIF4E). When mTORC1 is active, hyperphosphorylated EIF4EBP1 dissociates from eIF4E, which licenses translation initiation by enabling the recruitment of eIF4G and eIF4A to mRNA 5′ ends (Ma and Blenis, 2009). TORC1 also regulates tRNA gene expression via phosphorylation of the Pol III repressor MAF1 (Willis and Moir, 2018). We therefore postulated that the gradual activation of global translation and tRNA gene derepression during zebrafish embryogenesis could be coordinated by TORC1. To test this, we probed TORC1 activity by immunoblotting of whole embryo protein lysates throughout early embryogenesis (Fig. 6A). We analyzed the phosphorylation status of EIF4EBP1, which is a direct target of TORC1 (Gingras et al, 1999), and the 40S ribosomal protein S6 (RPS6), which is phosphorylated by ribosomal protein S6 kinase (RPS6K) that is in turn a direct target of TORC1 (Wullschleger et al, 2006) (Fig. 6A). Phosphorylation of zebrafish Eif4ebp1 and Rps6 was markedly reduced during cleavage and early blastula periods (1–4 hpf) in comparison to 0.2 hpf, and increased again when gastrulation was underway (from 6 hpf onward) (Fig. 6A; Appendix Fig. S1A). These data show that timing of tRNA gene expression and the increase in global translational activity temporally coincide with TORC1 activation during zebrafish embryogenesis.

In human cells, activated mTORC1 phosphorylates the Pol III inhibitor MAF1 at serines 60 (S60), 68 (S68), and 75 (S75), which renders it unable to repress Pol III transcription (Michels et al, 2010; Shor et al, 2010). Zebrafish encode two MAF1 homologs: Maf1a and Maf1b, which share 70% sequence similarity with the human protein (Fig. 6B). The three human MAF1 sites targeted by TORC1 are conserved in Maf1a (S60, S68 and S75) and Maf1b (S60, S69, S76). To investigate whether Maf1a and/or Maf1b are phosphorylated in a TORC1-dependent manner during the zebrafish MZT, we performed a mass spectrometry time-course by collecting embryos at the 1000-cell stage (3 hpf; during ZGA and before bulk tRNA expression activation), and during gastrulation at the sphere and bud stages (6 and 10 hpf, respectively) (Fig. 6C). Maf1a levels increased gradually, suggesting de novo protein synthesis during the MZT. By contrast, Maf1b levels remained

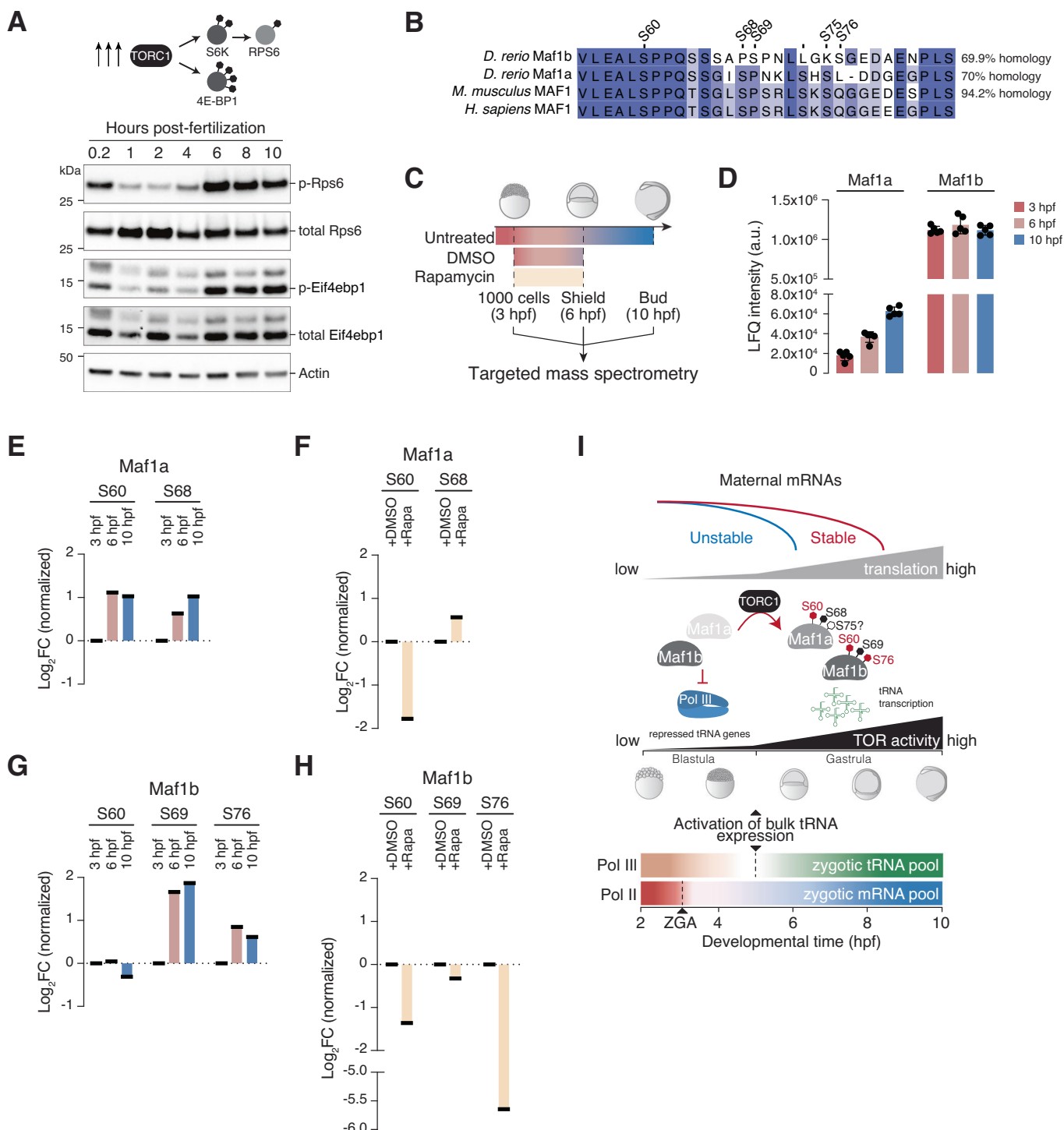

constant throughout the time-course, indicating that it is likely maternally deposited (Fig. 6D). The label-free quantification (LFQ) intensity for Maf1b was ~20-fold higher than that for Maf1a by the end of gastrulation (10 hpf) (Fig. 6D), suggesting that Maf1b is the functionally relevant paralog during zebrafish embryogenesis.

To assess TORC1-dependent changes in Maf1a and Maf1b phosphorylation, we treated embryos with rapamycin and collected embryos shortly after the onset of gastrulation (6 hpf), when

TORC1 is active (Fig. 6A–C). Analysis of the phosphorylation status of TORC1 targets Rps6 and Rps6k by immunoblotting and targeted mass spectrometry confirmed the inhibition of TORC1-mediated phosphorylation in vivo by rapamycin (Appendix Fig. S1B–F; Dataset EV5). By measuring phosphosite intensity at conserved residues that could be targeted by TORC1, we found that Maf1a is phosphorylated during gastrulation, with a 2-fold increase in phosphosite intensity at S60 and S68 ($p < 0.005$, one-way

◄ **Figure 6.  tRNA gene derepression and TORC1 activity increase coincide temporally during the zebrafish MZT.**

(A) Representative immunoblots of wild-type zebrafish embryo lysates during the maternal-to-zygotic transition (MZT) (0.2 to 10 hpf). Protein molecular weight markers are specified on the left and antibodies are indicated on the right. Actin was used as a loading control. (B) Pairwise alignment of human MAF1, mouse MAF1 and zebrafish Maf1a and Maf1b protein sequences. Sites known to be phosphorylated in human MAF1 and corresponding conserved residues are depicted. Percentage identity between protein sequences is depicted using the default Percentage Identity color scheme of Jalview. (C) Schematic depiction of the strategy used for collecting embryos developing in untreated, vehicle-treated (control; DMSO) and rapamycin-treated (Rapamycin; TORC1 inhibition) conditions for mass-spectrometry analysis. (D) Label-free quantification (LFQ) intensity (arbitrary units, a.u.) of Maf1a and Maf1b peptides in embryos at the 1000-cell (3 hpf), shield (6 hpf) and bud (10 hpf) stages of development. Bars represent mean ± SD ($n = 5$ for each time-point). (E) Average log2 fold-change of normalized phosphosite intensity for Maf1a (S60 and S68) at each developmental time-point and condition. Values reported for 6 hpf and 10 hpf are normalized to the 3 hpf time-point ($n = 5$) (F) Average log2 fold-change of normalized phosphosite intensity for Maf1a (S60 and S68) for each condition at 6 hpf. Values reported for rapamycin (Rapa)-treated samples are normalized to the DMSO-treated control ($n = 4$). (G) Plot as in (E) for Maf1b phosphosites (S60, S69 and S76). (H) Plot as in (F) for Maf1b phosphosites (S60, S69 and S76). (I) Schematics depicting the temporal coordination of global translational activation, TORC1 activity increase, and increase in tRNA gene expression by polymerase III during the zebrafish maternal-to-zygotic transition. Maf1a/b phosphosites sensitive to TORC1 inhibition are colored in red. See also Appendix Fig. S1. Source data are available online for this figure.

ANOVA) at 10 hpf relative to 3 hpf (Fig. 6E; Appendix Fig. S1G). Rapamycin treatment resulted in a 4-fold decrease in phosphorylation levels of Maf1a S60, while phosphorylation of Maf1a S68 increased, suggesting it is insensitive to rapamycin or targeted by another kinase (Fig. 6F; Appendix Fig. S1G; Mann–Whitney U test, $p < 0.05$). We could not determine the phosphorylation status of Maf1a S75 probably due to the poor ionization efficiency of the corresponding peptide (see Methods). Maf1b S60 was phosphorylated throughout the MZT and rapamycin treatment decreased phosphorylation levels by 2.6-fold ($p < 0.05$, Mann–Whitney U test; Fig. 6G,H; Appendix Fig. S1H), indicating that maternal Maf1b is inherited in a phosphorylated form and that TORC1 maintains Maf1b phosphorylation at S60 during the MZT. Interestingly, similarly to Maf1a S68, phosphorylation at Maf1b S69 significantly increased during embryogenesis ($p < 0.01$, one-way ANOVA), but it was not affected by rapamycin (Fig. 6H; Appendix Fig. S1H). In contrast, Maf1b S76 phosphorylation, which also increased significantly at gastrulation, was completely abolished by rapamycin treatment (Fig. 6H; Appendix Fig. S1H). Both Maf1a and Maf1b are thus phosphorylated in a TORC1-dependent manner as zebrafish embryos enter gastrulation, suggesting that TORC1 activity may temporally coordinate the global increase in translation with the onset of Pol III-mediated tRNA expression (Fig. 6I).

## Discussion

By combining recent advances in tRNA sequencing (Behrens et al, 2021) and ribosome profiling (Wu et al, 2019a), here we provide a quantitative characterization of tRNA repertoires and their functional impact on decoding rates and codon-mediated mRNA decay during early zebrafish embryogenesis. We find that zygotic tRNA expression ensues during gastrulation, subsequent to Pol II-mediated zygotic gene expression (Baia Amaral et al, 2024; Bhat et al, 2023; Harvey et al, 2013; Heyn et al, 2014; Lee et al, 2013), consistent with findings in a recent related study (Rappol et al, 2024). Our data suggest that this delay may be due in part to an increase in TORC1 activity we find during gastrulation, which we postulate alleviates Pol III repression by Maf1a/b through phosphorylation (Michels et al, 2010; Shor et al, 2010; Towpik et al, 2008). The dampening in TORC1 activity we show occurs prior to gastrulation (1–4 hpf) is reminiscent of the transient drop observed for multiple stem cell types at the onset of differentiation (Saba et al, 2021). At gastrulation onset, increased TORC1 signaling

inactivates both Maf1a/b and Eif4ebp1 through phosphorylation, suggesting it may concomitantly relieve Pol III inhibition and global translation repression. This coordinated switch could boost tRNA levels during lineage commitment, which requires enhanced protein synthesis capacity (Saba et al, 2021; Teixeira and Lehmann, 2019). Together with previous reports of TORC1 functions in mammalian cell differentiation (Baser et al, 2019; Bulut-Karslioglu et al, 2016; Castilho et al, 2009; Easley et al, 2010; Gao et al, 2024; Gui et al, 2023; LaFever et al, 2010; Sanchez et al, 2016; Sun et al, 2010) and mouse and sea urchin early embryogenesis (Chassé et al, 2018; Chassé et al, 2016; Cormier et al, 2001; Gangloff et al, 2004; Hentges et al, 2001; Murakami et al, 2004; Oulhen et al, 2007; Salaün et al, 2003), our data point to TORC1 as a conserved player coordinating the timing of increased global translation rates and tRNA expression as embryos begin to establish distinct cell lineages.

We find that zygotic tRNA expression modulates embryonic tRNA levels, resulting in distinct maternal and zygotic tRNA repertoires. The majority of tRNA transcripts differentially expressed during the MZT are upregulated, which likely results from active Pol III transcription from the sphere stage onwards (>4 hpf). In support of this notion, chromatin accessibility at a subset of genes encoding upregulated tRNAs increases during early embryogenesis. Differences in internal promoter or 5′ flanking sequences could account for this selective activation, as observed for human tRNA genes during cellular differentiation (Gao et al, 2024). Some zebrafish tRNA transcripts also decrease significantly in abundance as gastrulation progresses, indicating a dilution of maternally deposited tRNAs or their selective decay through unknown mechanisms. Of note, we find that zebrafish tRNA gene copy number does not correlate with tRNA levels at any stage of early embryogenesis, underscoring the need to quantify mature tRNA abundance during dynamic cellular transitions. This lack of correlation could be due to selective expression of tRNA genes inactive during embryogenesis at later developmental stages, or only in rare cell types. Alternatively, it may stem from the genomic distribution of predicted zebrafish tRNA genes: 60% of them are located on chromosome 4, which consists almost entirely of heterochromatin in all tissues (Yang et al, 2020) and has a remarkably high repeat content due to an insertion preference of transposable elements and low recombination (Chang et al, 2022; Howe et al, 2013). We note that tRNA gene predictions in the GtRNAdb (Chan and Lowe, 2016) are only available for the current GRCz11 build of the zebrafish genome, which however has known

gaps and possible duplications, particularly on the tRNA-rich chromosome 4 (Chernyavskaya et al, 2022; Yang et al, 2020). Nevertheless, a colocalization with heterochromatin regions is consistent with the lack of chromatin accessibility we observed at the majority of predicted tRNA genes in ATAC-Seq datasets (Pálfy et al, 2020).

We find that ribosome density at individual codons in endogenous mRNAs inversely correlates with the availability of their cognate tRNAs in early zebrafish embryos. Previously, such a correlation has been observed only for triplets encoding polar amino acids in synthetic mRNA reporters (Mishima et al, 2022), which could be due to challenges in capturing footprints from decoding ribosomes in the absence of tigecycline (Gao et al, 2024; Wu et al, 2019a). We further show that the reprogramming of tRNA repertoires during the zebrafish MZT does not reflect a regulatory adaptation to the codon demand of the zygotic transcriptome. Since most maternal mRNAs are zygotically re-expressed (Bhat et al, 2023; Fishman et al, 2024; Fishman et al, 2023) and we show that global codon demand remains invariant, our data are at odds with the notion that tRNA repertoire remodeling during the MZT serves to modulate codon-dependent mRNA decay. Instead, we propose that tRNA availability is an intrinsic rather than a regulatory determinant of maternal mRNA stability in zebrafish embryos. In line with this notion, we find that the codon content of maternal-stable mRNAs is better matched to both maternal and zygotic tRNA pools than the codon content of maternal-unstable mRNAs. However, maternal-stable transcripts are decoded significantly faster than maternal-unstable ones only at later stages of gastrulation, following a global increase in translation. These observations are in agreement with previous findings that transcripts with a higher content of rare codons can be targeted for deadenylation by the CCR4-NOT complex during the zebrafish MZT, presumably as a consequence of a prolonged vacancy of the ribosomal A-site (Buschauer et al, 2020; Hanson et al, 2018; Mishima et al, 2022). We note that maternal-stable and maternal-unstable transcripts are also differentially enriched in other features that can affect mRNA half-life during the zebrafish MZT such as 3′ UTRs and poly(A) tail length (Bhat et al, 2023), and that our data suggest that the stabilizing effects of some codons may be related to mRNA nucleotide composition rather than tRNA availability and decoding rates. Collectively, our findings support a model in which increased translational activity sensitizes decoding rates to tRNA levels during zebrafish gastrulation, rationalizing the late onset of codon-dependent effects on mRNA half-life (Bazzini et al, 2016; Despic and Neugebauer, 2018; Mishima and Tomari, 2016).

# Methods

### Reagents and tools table

| Reagent/Resource | Reference or Source | Identifier or Catalog Number |
|---|---|---|
| **Experimental Models** | | |
| Zebrafish (*Danio rerio*) | TL/AB | |
| **Antibodies** | | |
| Phospho-S6 Ribosomal Protein (Ser240/244) (D68F8) XP® | Cell Signaling Technology | Rabbit mAB #5364 |
| S6 Ribosomal Protein (5G10) | Cell Signaling Technology | Rabbit mAB #2217 |
| Phospho-4E-BP1 (Thr37/46) (236B4) | Cell Signaling Technology | Rabbit mAB #2855 |
| 4E-BP1 (53H11) | Cell Signaling Technology | Rabbit mAB #9644 |
| Anti-Actin Antibody, clone C4 | Sigma-Aldrich | MAB1501 |
| **Oligonucleotides and other sequence-based reagents** | | |
| RT primer for sequencing library construction | 5′-pRNAGATCGGAAGAGCGTCGTGTAGG GAAAGAG/iSp18/GTGACTGGAGTT CAGACGTGTGCTC-3′ | IDT |
| rRNA depletion oligo #1 | 5′-/5BiotinTEG/ GAGGGACCTGTGGCGTACGGAGGGCCGC-3′ | IDT |
| rRNA depletion oligo #2 | 5′-/5BiotinTEG/ ATCCCGGGGCCACGCCTGTCTGAGGGTCGCC-3′ | IDT |
| rRNA depletion oligo #3 | 5′-/5BiotinTEG/ ACCCGGGGACGCGTGCATTTATCAGA-3′ | IDT |
| rRNA depletion oligo #4 | 5′-/5BiotinTEG/ TCGTTTCCTCACTTACCCGGTGAGGCGGGG-3′ | IDT |
| rRNA depletion oligo #5 | 5′-/5BiotinTEG/ CCTACCCGCCGCCGTCCGCCCCCG-3′ | IDT |
| rRNA depletion oligo #6 | 5′-/5BiotinTEG/ ATGTTAGGGATAACAGGGTAATGCGA-3′ | IDT |
| rRNA depletion oligo #7 | 5′-/5BiotinTEG/ ACCGCCGGTCCCGGACCCCCAGGCCT-3′ | IDT |
| rRNA depletion oligo #8 | 5′-/5BiotinTEG/ CTGCGCCGCGCCGCCCAGGGCGGG-3′ | IDT |
| Northern blotting probe for tRNA-Ser-UGA-19 | 5′-CGCTGCGAGCAGGGTTCGAACCTGC-3′ | IDT |
| Northern blotting probe for tRNA-Thr-UGU-8 | 5′-GAACTCGCGACCCCTGGTTTACAAGAC-3′ | IDT |
| Northern blotting probe for tRNA-Ala-AGC-3 | 5′-TGGAGAATGCGGGCATCGATCCCGCTAC-3′ | IDT |
| **Chemicals, Enzymes, and other reagents** | | |
| Sequencing Grade Modified Trypsin | Promega | V5113 |
| Glu-C, Sequencing Grade | Promega | V165A |
| Glu-Glu peptide | Sigma-Aldrich | G3640 |
| cOmplete™, Mini, EDTA-free Protease Inhibitor Cocktail | Roche | 11836170001 |
| PhosStop™ | Roche | 4906837001 |
| RNasin® ribonuclease inhibitor | Promega | N2515 |
| Nupage™ 4–12% Bis-Tris Protein Gels | Invitrogen | NP0322BOX |
| TRIzol Reagent | Thermo Fisher Scientific | 15596026 |
| Formamide, deionized | Sigma-Aldrich | F9037 |
| Acrylamide/Bis 19:1, 40% (w/v) solution | Thermo Fisher Scientific | AM9022 |
| Glycogen | Thermo Fisher Scientific | AM9510 |
| Low Range ssRNA ladder | New England Biolabs | N0364S |
| T4 Polynucleotide Kinase | New England Biolabs | M0201L |
| T4 RNA ligase 2, truncated KQ (200 U/µL) | New England Biolabs | M0373L |

| Reagent/ Resource | Reference or Source | Identifier or Catalog Number |
|---|---|---|
| SUPERase•In RNase Inhibitor (20 U/μL) | Thermo Fisher Scientific | AM2694 |
| TGIRT | InGex | TGIRT50 |
| CircLigase ssDNA ligase | Lucigen | CL4115K |
| KAPA HiFi DNA Polymerase (1U/μL) | Roche | KK2102 |
| **Software** | | |
| cutadapt v3.5 | (Martin, 2011) | https://cutadapt.readthedocs.io/en/stable/ |
| mim-tRNAseq v1.2 | (Behrens et al, 2021) | https://github.com/nedialkova-lab/mim-tRNAseq |
| STAR v2.6.1c | (Dobin et al, 2013) | |
| Scikit-ribo v0.2.4bl customized for use on large genomes | (Fang et al, 2018) | https://github.com/nedialkova-lab/scikit-ribo-ext |
| Mfuzz | (Kumar and Futschik, 2007) | https://www.bioconductor.org/packages/release/bioc/html/Mfuzz.html |

## Zebrafish husbandry and embryo rearing

Wild-type zebrafish embryos (TLAB strain) were grown in standard housing conditions (28 °C at a 14/10 h light/dark cycle). The TLAB strain was generated by crossing zebrafish natural variant TL (Tupfel Longfin) with AB strain. Embryos were incubated in E3 medium (5 mM NaCl, 0.17 mM KCl, 0.33 mM CaCl$_2$, 0.33 mM MgSO$_4$, pH 7.2) in standard conditions (28 °C at a 14/10 h light/dark cycle). All experiments were conducted according to federal guidelines for animal research and approved by the Kantonales Veterinäramt of Kanton Basel-Stadt (under the Animal Holding License Form 1035H).

## Western blotting

Protein lysates were prepared using 1× Lysis buffer (30 mM HEPES–KOH pH 7.4, 100 mM potassium acetate, 2 mM magnesium acetate, 5 mM DTT, 0.5% NP-40, 5% glycerol) containing protease inhibitor cocktail (cOmplete mini, 11836170001, Roche) and phosphatase inhibitor (PhosStop, 4906837001, Roche). Lysates (30 μg/well) were separated by SDS-PAGE using NuPAGE 4–12% Bis-Tris gradient gels and wet-transferred onto nitrocellulose membranes (GE Healthcare). Blotted membranes were blocked for 1 h at room-temperature in 5% BSA (in TBS-Tween 0.1%) prior to antibody incubation. Primary antibodies were incubated at 4 °C overnight, secondary antibodies were incubated at room-temperature for 2 h. Antibodies were used at a dilution of 1/1000 for Phospho-S6 Ribosomal Protein (Ser240/244) (clone D68F8, rabbit mAb #5364, Cell Signaling Technology), Total-S6 Ribosomal Protein (clone 5G10, rabbit mAb #2217, Cell Signaling Technology), Phospho-4E-BP1 (Thr37/46) (clone 236B4, rabbit mAb #2855, Cell Signaling Technology) and Total-4E-BP1 (clone 53H11, rabbit mAb #9644, Cell Signaling Technology) or 1/10,000 for Actin (clone C4, MAB1501, Sigma-Aldrich), and detected by secondary HRP-antibody conjugates. Maf1 antibodies

tested that did not yield signal for zebrafish Maf1a/b were MAF1 (H-2) sc-515614 (Santa Cruz Biotechnology), MAF1 A15531 (ABclonal) and MAF1 GTX106776 (GeneTex). For sequential detection of phosphorylation levels, total protein levels, and loading control using the same blotted membrane, membranes were stripped for 10 min using Restore Western Blot Stripping buffer (Thermo Scientific), washed and blocked between primary antibody incubation steps. Western blot images were acquired on a ChemiDoc MP Imaging System (BioRad).

## Polysome profiling

For each developmental time-point (4, 6, 8, and 10 hpf), 150 embryos were staged (Kimmel et al, 1995) and collected per sample and immediately flash-frozen in liquid nitrogen. Immediately before polysome gradient preparation, embryos were lysed in 450 μL polysome gradient buffer (20 mM Tris-Cl pH 7.5, 30 mM MgCl$_2$, 100 mM NaCl, 0.25% Igepal-630 [v/v], 100 μg/mL cycloheximide, 0.5 mM dithiothreitol [DTT], 1 mg/mL heparin and 40 U/mL recombinant RNasin ribonuclease inhibitor [Promega]) with ~40 strokes of a motor-driven "B" pestle while on ice. After lysis, 14U of TURBO DNase (Invitrogen) were added and the embryo lysate was incubated for 10 min at 4 °C on a rotating mixer. Samples were centrifuged at 13,000 rpm for 10 min at 4 °C, and 400 μL of clarified lysate were loaded onto a 5–50% (w/v) sucrose gradient prepared in TMS buffer (20 mM Tris-Cl pH 7.5, 5 mM MgCl$_2$, 140 mM NaCl, 100 μg/mL cycloheximide, 1 mM DTT). Gradients were centrifuged in a SW40 Ti rotor (Buschauer et al) at 4 °C for 150 min and profiles were analyzed using a Gradient Station (Biocomp Instruments) with continuous recording of optical density (OD) at 260 nm.

## Sample collection and total RNA isolation

50–200 embryos per sample were collected at the 256-cell (2.5 hpf), 1000-cell (3 hpf), sphere (4 hpf), shield (6 hpf), and bud (10 hpf) stage (Kimmel et al, 1995). For collecting non-activated, unfertilized eggs, female zebrafish were anesthetized by immersion in water containing 50 mg/ml of tricaine methanesulfonate (methyl-*m*-aminobenzoate, MS 222) and eggs were obtained by gently pressing on the sides of females, moving from behind the pectoral fins towards the tail. Non-activated eggs were transferred into 1.5 mL falcon tubes. The replicate developmental time-point samples used for mim-tRNAseq and matched northern blotting were collected on different days. For measuring tRNA abundance, embryos and non-activated eggs were lysed in 800 μl LiDS/LET buffer (5% lithium dodecyl sulfate in 20 mM Tris-HCl pH = 7.4, 100 mM LiCl, 2 mM EDTA, 5 mM DTT) upon collection. Lysates were supplemented with 100 μg/ml Proteinase K, incubated at 60 °C for 10 min, and triturated through a 1-ml syringe with a 26G needle. Two volumes of cold acid phenol (pH 4.3), 1/10 volume 1-Bromo-3-chloropropane and 50 μg glycogen (Thermo Fisher Scientific, #AM9510) were then added to each sample. Samples were mixed vigorously by vortexing, followed by centrifuging at 10,000 × *g*/4 °C for 5 min. After transfer of the aqueous phase to a new tube, the phenol/BCP extraction was repeated, and RNA was then precipitated from the aqueous phase by the addition of 3 volumes of 100% ethanol at −20 °C for 30 min. Following centrifugation at 16,000 × *g*, 4 °C for 20 min, pellets were washed

with 80% ethanol, air-dried, resuspended in RNase-free water, and stored at −80 °C. For measuring tRNA charging, embryos were collected in Trizol and total RNA was isolated under mild acidic conditions as previously described (Behrens and Nedialkova, 2022).

## mim-tRNAseq library construction

Libraries were prepared from total RNA using the mim-tRNAseq workflow (Behrens and Nedialkova, 2022). For measuring tRNA abundance, 10 ng synthetic *E. coli* tRNA-Lys-UUU-CCA and E.coli tRNA-Lys-UUU-CC in a 3:1 ratio were spiked into total RNA and the mixture was incubated in 75 mM Tris-HCl, pH = 9.0 at 37 °C for 45 min to deacylate tRNAs. For measuring tRNA aminoacylation, total RNA purified under mild acidic conditions was subjected to oxidation and beta-elimination as previously described (Behrens and Nedialkova, 2022). RNA 3′ ends were dephosphorylated with T4 PNK (NEB, #M0201S) and RNA was purified by ethanol precipitation, and RNA of 60–100 nt was size-selected on denaturing 10% polyacrylamide/7 M urea/1×TBE gels. Following elution from gel slices and purification by ethanol precipitation, the size-selected RNA was ligated to one of eight pre-adenylated, barcoded 3′-adapters (Behrens and Nedialkova, 2022) in 1×T4 RNA ligase buffer, 25% PEG-8000, 20 U Superase In (Thermo Fisher Scientific, #AM2696) and 1 µl T4 RNA Ligase 2, truncated KQ (NEB, #M0373S) for 3 h at 25 °C. Ligation products were pooled and purified by size selection on a 10% polyacrylamide/7 M urea/1×TBE gel, followed by elution from gel slices and ethanol precipitation. 100 ng pooled adapter-ligated RNA was annealed with 1 µl of 1.25 µM RT primer at 82 °C for 2 min and 25 °C for 5 min. Reverse transcription was performed in 50 mM Tris-HCl pH = 8.3, 75 mM KCl, 3 mM MgCl$_2$, 5 mM DTT (from a freshly made 100 mM stock), 1.25 mM dNTPs and 20 U Superase In with 500 nM TGIRT (InGex, #TGIRT50) at 42 °C for 16 h. NaOH was then added to 0.1 M and the samples were incubated for 5 min at 90 °C to hydrolyze the template RNA. Samples were then loaded on a 10% polyacrylamide/7 M urea/1×TBE gel and cDNAs >10 nt longer than RT primer were excised after SYBR Gold staining. cDNA was eluted from crushed gel slices for 60 min at 1500 rpm/70 °C in 1TE buffer. Following ethanol precipitation, gel-purified cDNA was incubated with CircLigase ssDNA ligase (Lucigen) in 1× reaction buffer supplemented with 0.05 mM ATP, 2.5 mM MnCl$_2$, and 1 M betaine for 3 h at 60 °C, followed by enzyme inactivation for 10 min at 80 °C. One-fifth of the circularized cDNA was directly used for library construction PCR with KAPA HiFi DNA Polymerase (Roche) as previously described (Behrens and Nedialkova, 2022). Following purification with DNA Clean&Concentrator 5 (Zymo Research), the resulting libraries quantified with the Qubit dsDNA HS kit (Thermo Fisher Scientific, #Q32851) and sequenced for 150 cycles on an Illumina NextSeq 500 platform, generating 1.3–6.5 million reads per library.

## mRNA sequencing library construction

Total RNA from sphere (4 hpf) and bud (10 hpf) samples from the same batch used for mim-tRNAseq library construction was subjected to DNase treatment with the Zymo RNA Clean & Concentrator kit (Zymo Research, #R1014) and mRNA sequencing libraries were constructed with the Zymo-Seq RiboFree Total RNA Library Kit (Zymo Research, #R3000) from 250 ng input. Fragment

size was determined on an Agilent TapeStation. Libraries were quantified with the Qubit dsDNA HS assay sequenced for 80 cycles on an Illumina NextSeq 500 platform, generating 27–31 million reads per library.

## Ribosome profiling library construction

For each developmental time-point (4 and 10 hpf), 200 embryos were collected per sample and immediately flash-frozen in liquid nitrogen. Ribosome profiling libraries were prepared based on Wu et al (2019a) with some modifications. Embryos were mixed with 1 ml drip-frozen footprint lysis buffer (20 mM Tris pH = 7.4, 150 mM NaCl, 5 mM MgCl$_2$, 1% Triton-X100, 1 mM DTT) supplemented with 100 µg/ml cycloheximide (Sigma-Aldrich, #C1988) and 100 µg/ml tigecycline (Sigma-Aldrich, #PZ0021), and lysed by cryogenic grinding at 5 cps for 2 min in a SPEX 6775 Freezer/Mill® cryogenic grinder. The powder was dissolved in 15 ml ice-cold footprint lysis buffer supplemented with 100 µg/ml CHX and 100 µg/ml TIG and the lysates were cleared by centrifugation at 3000 × g/4 °C for 5 min. Ribosomes were subsequently pelleted from cell extracts through 3 ml of a sucrose cushion (1 M sucrose, 20 mM Tris pH = 8.0, 140 mM KCl, 5 mM MgCl$_2$, 1 mM DTT) by centrifuging the layered solutions in a Type 70 Ti rotor (Beckmann) at 50,000 rpm/4 °C for 120 min. Ribosome pellets were rinsed once with 1 ml footprint lysis buffer and dissolved in 300 µl footprint lysis buffer. RNA concentration was measured by Nanodrop, and 150 U RNase I (Thermo Fisher Scientific, #AM2295) per A260 unit of extract were added to each sample. After incubation for 60 min at 500 rpm/25 °C, extracts were supplemented with 100 U Superase In (Thermo Fisher Scientific, #AM2694) and loaded on 10–45% sucrose gradients in 20 mM Tris-Cl pH = 8.0, 150 mM KCl, 5 mM MgCl$_2$, 0.5 mM DTT. Gradients were centrifuged in a TH-641 rotor (Sorvall) for 90 min at 35,000 rpm/4 °C and fractionated on a Gradient Station (Biocomp). Monosome fractions were collected, supplemented with SDS to a final concentration of 1%, and snap-frozen. RNA isolation from monosome fractions was performed by addition of two volumes of hot acid phenol (pH = 4.3) and incubation at 65 °C for 5 min in a water bath. One-tenth volume of BCP was added to each sample, followed by centrifugation at 10,000 × g for 5 min. After transfer of the aqueous phase to a new tube, the phenol/BCP extraction was repeated, and RNA was then precipitated from the aqueous phase by the addition of 3 volumes of 100% ethanol at −20 °C for 30 min. Following centrifugation at 16,000 × g for 20 min, pellets were air-dried, resuspended in RNase-free water, and stored at −80 °C.

5 µg of RNA extracted from monosome fractions were mixed with loading dye, denatured for 3 min at 90 °C and separated on a 15% polyacrylamide/7 M urea/1×TBE gel. Fragments in the range of 19 to 32 nucleotides were excised from the gel and crushed with disposable 1.5-ml pestles. RNA was eluted by heating for 10 min at 65 °C in 400 µl gel elution buffer (0.3 M NaOAc pH = 4.5, 0.25% SDS, 1 mM EDTA pH = 8.0), followed by snap-freezing on dry ice for 10 min. The gel slurry was then thawed for 5 min at 65 °C and incubated overnight at room temperature on a rotating wheel. Samples were centrifuged through a Spin-X filter (Corning) and RNA was purified from the flowthrough by ethanol precipitation.

Size-selected RNA was dephosphorylated with 10 U T4 PNK (NEB, #M0201S) for 45 min at 37 °C. Following purification by

ethanol precipitation, the RNA was ligated to a pre-adenylated adapter containing 5 random nucleotides at its 5′ end followed by a unique barcode (McGlincy and Ingolia, 2017) in 1× T4 RNA ligase buffer, 25% PEG-8000, Superase In and 1 µl T4 RNA Ligase 2, truncated KQ (NEB, #M0373S) for 3 h at 25 °C. Ligation products were purified by size selection on a 12% polyacrylamide/7 M urea/1×TBE gel. Following elution from gel slices, samples with unique barcodes were pooled, and 50 ng of the linker-ligated RNA was annealed with RT primer) at 65 °C and reverse-transcribed for 30 min at 50 °C in reaction containing 1× Protoscript II Buffer, 0.5 mM dNTPs, 10 mM DTT, 20 U Superase In and 200 U Protoscript II (NEB, #E6560S). NaOH was added to 0.1 M for 5 min at 90 °C to hydrolyze RNA templates. cDNA products were purified by size selection on a 12% polyacrylamide/7 M urea/1×TBE gel. After elution from gel slices, gel-purified cDNA was incubated with CircLigase ssDNA ligase (Lucigen) in 1× reaction buffer supplemented with 1 mM ATP, 50 mM $MgCl_2$, and 1 M betaine for 3 h at 60 °C, followed by enzyme inactivation for 10 min at 80 °C. Ribosomal RNA contaminants were depleted from circularized cDNA using a mixture of 5′-biotinylated oligos) as described (Ingolia et al, 2012). Libraries were constructed from circularized cDNA using the same workflow and primers as for tRNA library construction, and eight cycles of PCR in 1× KAPA HiFi buffer. After quantification with the Qubit dsDNA High Sensitivity kit, 150-bp single-end sequencing was performed on an Illumina NextSeq 500 platform, generating 25.7 million reads (sphere, 4 hpf) and 37.1 million reads (bud, 10 hpf).

## Northern blotting

Northern blotting was performed as described (Behrens et al, 2021). Equal amounts of total RNA from each sample (0.5–2 µg depending on the probe) was separated on 10% polyacrylamide in 7 M urea and 1×TBE gels. Following staining with SYBR Gold to confirm equal loading, RNA was transferred to Immobilon NY+ membranes (Millipore) in 1×TBE for 40 min at 4 mA cm$^{-2}$ with a TransBlot Turbo apparatus (Bio-Rad). RNA was cross-linked to membranes at 0.04 J in a Stratalinker UV light crosslinker. Following incubation at 80 °C for 1 h, the membranes were pre-hybridized in hybridization buffer (20 mM $Na_2HPO_4$ pH 7.2, 5×SSC, 7% SDS, 2×Denhardt's solution and 40 µg/ml sheared salmon sperm DNA) at 55 °C for 4 h, followed by hybridization with 10 pmol of a 5′-end $^{32}$P-labeled probes in hybridization buffer at 55 °C overnight. The membranes were washed three times in 25 mM $Na_2HPO_4$ pH 7.5, 3×SSC, 5% SDS, 10×Denhardt's solution, and once in 1×SSC, 10% SDS. Signal was detected by phosphor imaging on a Typhoon FLA 9000 (GE Healthcare) and band intensity was quantified using ImageJ.

## Mass spectrometry sample preparation

Embryos were staged (Kimmel et al, 1995) and grown in 15 cm dishes (150 embryos/plate) in standard conditions. Approximately 30 min prior to the desired stage, embryos were dechorionated in pre-warmed (28 °C) E3 medium containing 1 mg/mL Pronase (P5147, Sigma-Aldrich) for 6–9 min under constant orbital shaking (40 rpm) at room temperature in a glass dish. Embryos were then thoroughly washed (5 times) in pre-warmed E3 medium to remove all residual Pronase and to release the embryos from the Pronase-treated

chorions. Dechorionated embryos were deyolked according to Link et al (2006) with some modifications. Briefly, embryos were transferred into Protein LoBind Eppendorf tubes containing 1.5 ml pre-chilled deyolking buffer (55 mM NaCl, 1.8 mM KCl, 1.25 mM $NaHCO_3$, 5% glycerol). For each replicate sample at the 1000-cell stage (3 hpf), 500–600 embryos were collected. For each replicate sample at the shield (6 hpf) or bud (10 hpf) stages, 300–400 embryos were collected. Embryos were brought into single-cell suspensions by gently pipetting up and down with a P1000 tip while on ice. Samples were shaken at 1100 rpm for 30 s at 4 °C in a thermal shaker, following gentle centrifugation at $200 \times g$ for 30 sec at 4 °C. The supernatant was discarded, and three to five rounds of washing with 1.5 ml pre-chilled wash buffer (110 mM NaCl, 3.5 mM KCl, 10 mM Tris pH 8.0, 2.7 mM $CaCl_2$, 5% glycerol, 1 tablet/10 ml buffer of protease inhibitor cocktail [cOmplete mini, 11836170001, Roche] and 1 tablet/10 ml phosphatase inhibitor [PhosStop, 4906837001, Roche]) were performed, until yolk residues were no longer apparent. After the last wash, the supernatant was discarded and the single-cell solution was resuspended in 50 µl pre-chilled wash buffer and immediately stored at −80 °C. Five replicates were collected per time-point. For TORC1-inhibition experiments, DMSO (1% final concentration) or Rapamycin (1 µM final concentration) were added to pre-warmed (28 °C) E3 medium and 1000-cell stage embryos were transferred to 15 cm dishes containing either DMSO or Rapamycin. Embryos were grown in standard conditions until the shield stage (6 hpf), followed by deyolking and dissociation as described above. Four replicates of 300–400 embryos each were collected per condition at the shield stage (6 hpf).

For digestion of the native proteome, each sample was lysed in 60 µl lysis buffer (8 M urea, 0.1 M ammonium bicarbonate, 5 mM TCEP, 1 tablet/10 ml buffer of protease inhibitor cocktail [cOmplete mini, 11836170001, Roche] and 1 tablet/10 ml phosphatase inhibitor [PhosStop, 4906837001, Roche]). Samples were sonicated using a BioRuptor (Diagenode) (10 cycles, 30 s on, 30 s off) and then incubated for 10 mins at 50 °C with shaking at 300 rpm in a thermal shaker. To each sample, 1 µl of 0.75 M chloroacetamide was added and incubated for 1 h at 37 °C with shaking at 600 rpm. Afterwards, 540 µl 0.1 M ammonium bicarbonate was added to each sample, for a final concentration of urea of 0.8 M. The pH of each sample was checked (~pH 8), and Glu-Glu peptide ($C_{10}H_{16}N_2O_7$; G3640, Sigma-Aldrich) was added to a final concentration of 0.5 mM. Then, 2.5 µg of GluC protease (V165A, Promega) were added per sample (~1:50 protein ratio) and incubated for 4 h at 37 °C. After 4 h incubation, another 2.5 µg of GluC protease (V165A, Promega) and 2.5 µg Trypsin protease (V5113, Promega) were added to each sample, followed by incubation overnight at 37 °C. After acidification using 50 µl of 20% TFA (28904, Thermo Fisher), the peptides were desalted on C18 reversed-phase spin columns according to the manufacturer's instructions (Macrospin, Harvard Apparatus, Holliston, MA, USA) and dried under a vacuum. Peptide samples were enriched for phosphorylated peptides using Fe(III)-IMAC cartridges on an AssayMAP Bravo platform as recently described (Post et al, 2017). Peptides were stored at −20 °C until measurement.

## Mass spectrometry data acquisition

In a first step, parallel reaction-monitoring (PRM) assays (Peterson et al, 2012) were generated from a mixture containing ~50 fmol of each

proteotypic heavy reference peptide. Peptides were subjected to LC–MS/MS analysis using an Orbitrap Exploris 480 Mass Spectrometer fitted with an Vanquish Neo (both Thermo Fisher Scientific) and a custom-made column heater set to 60 °C. Peptides were resolved using a RP-HPLC column (75 μm × 30 cm) packed in-house with C18 resin (ReproSil-Pur C18–AQ, 1.9 μm resin; Dr. Maisch GmbH) at a flow rate of 0.2 μL/min. Separation of peptides was achieved by using the following gradient: 4% Buffer A to 10% Buffer B in 3 min, 10% Buffer A to 35% Buffer B in 30 min, 35% Buffer A to 50% Buffer B in 7 min. Buffer A was 0.1% formic acid in water and buffer B was 80% acetonitrile, 0.1% formic acid in water. The mass spectrometer was operated in either DIA or PRM acquisition mode with a total cycle time not exceeding approximately 4 s. For PRM, the following MS1 parameters were set: Resolution: 120,000 FWHM (at 200 $m/z$), Scan Range: 350–1600 $m/z$, Injection time: 25 ms, Normalized AGC Target: 300%. The following MS2 parameters were set: Isolation Window: 0.4 $m/z$, HCD Collision Energy (normalized): 30, Normalized AGC target: 3000%, DataType: Centroid. For synthetic isotopically heavy labeled peptides, 50 ms injection time and 15,000 resolution (at 200 $m/z$) were set whereas for endogenous peptides 120,000 resolution and 250 ms injection time were set. For DIA, the following MS1 parameters were set: Resolution: 120,000 FWHM (at 200 $m/z$), Scan Range: 400–1200 $m/z$, Injection time: 25 ms, Normalized AGC Target: 300%. The following MS2 parameters were set: Isolation Window: 12 $m/z$, HCD Collision Energy (normalized): 30, Normalized AGC target: 3000%, DataType: Centroid. All raw-files were either imported into Skyline (Version 23.1) for protein/peptide quantification or analyzed by DIA-NN. The synthetic peptides used for quantification of the phosphorylation status of *D. rerio* maf1a/Q6PGU2 at positions S73 and S75 (Dataset EV5) did not allow us to confidently distinguish the two phosphosites due to low ionization efficiency of the synthetic and endogenous peptides corresponding to these phosphosites. For that reason, we do not report their phosphorylation levels.

## mim-tRNAseq data analysis

Read demultiplexing (based on unique barcodes in 3′ adapters) and 3′ adapter removal were performed with cutadapt v3.5 (Martin, 2011). Indels were not allowed during adapter matching (*--no-indels*) and both read ends were quality-trimmed (*-q 30,30*). Only trimmed reads were retained (*--trimmed-only*) as all tRNA-derived reads are expected to contain adapters after 150 cycles of sequencing. Reads were further trimmed to remove the two 5′ nucleotides introduced by circularization from the RT primer with *-u 2*. Reads shorter than 10 nt were discarded using *-m 10* during all trimming steps. tRNA expression and modification analysis was performed with v1.2 of the mim-tRNAseq computational package (https://mim-trnaseq.readthedocs.io/en/latest/index.html). Briefly, the precompiled GtRNAdb GRCz11 reference was used (*--species Drer*) with a cluster ID of 0.93, maximum mismatch tolerance at a number of nucleotides equal to 7.5% read length for the first round of alignment and realignment, a deconvolution coverage ratio of 0.4 at mismatch sites to allow accurate cluster deconvolution, and a minimum coverage threshold of 0.1% total reads per transcript for low coverage transcript filtering (*mimseq --species Drer --cluster-id 0.93 --threads 40 --min-cov 0.001 --deconv-cov-ratio 0.4 --max-mismatches 0.075 --remap --remap-mismatches 0.075 --control-condition Egg -n drer_-mix --out-dir drer_0.075_remap0.075_ID0.93_deconv0.4_mimseqv12 --max-multi 10 sampleData_zt_devReduced.txt*). For anticodon-level

analysis, DESeq2 (Love et al, 2014) was performed with all uniquely mapped reads. For transcript-level analysis, DESeq2 (Love et al, 2014) was rerun using only tRNAs with single-transcript resolution after read deconvolution.

## mRNA-seq data analysis

A custom reference transcriptome annotation was built for the GRCz11.108 genome using the best-scoring transcript annotated in APPRIS (Rodriguez et al, 2013) the principal isoform for each gene. In the case of multiple best-scoring transcripts associated with the same gene, the longest one was selected; when multiple best-scoring transcripts had the same length, a random one was chosen as the principal isoform. This reference set was further filtered to retain only transcripts with coding regions starting with an AUG codon, ending with a UAG/UAA/UGA stop codon, a length that is a multiple of three, and not containing unidentified bases or an additional stop codon in the open reading frame. The resulting reference transcriptome contained 21,552 protein-coding transcripts.

Trim Galore v0.6.4 with default settings was used to find and remove 3′ adapters from sequencing reads. Trimmed reads of length ≥20 were then aligned to the GRCz11.108 genome using STAR v2.6.1c (Dobin et al, 2013) with the following parameters: *--outSAMtype BAM SortedByCoordinate --outFilterMultimapNmax 1 --outFilterMismatchNmax 1 --quantMode TranscriptomeSAM GeneCounts*. To calculate coding-gene expression in transcripts per million (TPM), we used Kallisto 0.44.0 with parameters *-b 100 --single -l 180 -s 20 -t 40*.

## Ribosome profiling data analysis

Sequencing reads were demultiplexed based on unique barcodes in 3′ adapters, and adapter sequences were removed with Cutadapt v2.5 (Martin, 2011), discarding reads with indels in the alignment to the adapter sequence or without adapters, and removing low-quality bases from both 5′ and 3′ ends (*--no-indels --trimmed-only -q 30,30*). The two 5′ nucleotides introduced by cDNA circularization were subsequently trimmed (*-u 2*). To filter out rRNA contaminants, trimmed reads of length ≥10 were aligned to a zebrafish rRNA reference using Bowtie v1.2.2 (Langmead et al, 2009) with options *-p 40 -S --best*. The rRNA-filtered reads were then aligned to the GRCz11.108 zebrafish genome build using STAR v2.6.1c with the following options: *--outFilterMultimapNmax 1 --seedSearchStartLmax 15 --outSAMtype BAM SortedByCoordinate --outFilterMismatchNmax 2 --alignEndsType EndToEnd --quantMode TranscriptomeSAM --outSAMattributes NH HI AS nM NM MD*. This resulted in 4.0 million (sphere) and 10.6 (bud) million preprocessed reads aligning to the reference set of principal coding isoforms in the GRCz11.108 transcriptome.

The ribosome A-site location was identified in each mapped read with Scikit-ribo (Fang et al, 2018). This software uses a random forest with recursive feature selection for accurate A-site prediction and a generalized linear model to estimate the dwell time of each codon based on matched ribosome profiling and RNA-Seq datasets. To run scikit-ribo, we first omitted all the RNAfold dependencies in Scikit-ribo. Moreover, the index was built by running *scikit-ribo-build.py* separately for each chromosome to avoid memory errors due to the large size of the zebrafish genome and the presence of multiple transcript isoforms. Finally, the GTF

file with the genome annotation was made compatible with Scikit-ribo, removing transcript/UTR annotations. Specifically, the start codon in the first exon of each transcript and the stop codon in the last exon were adjusted to represent transcript start and end coordinates, taking into account the gene strand. To run Scikit-ribo and estimate codon dwell times (DTs), we focused on 30–34 nts ribosome footprints, running *scikit-ribo-run.py* with option *-s 30 -l 34*. These ribosome footprint lengths correspond to the peak in the ribosome footprint distribution and show the best frame information. Decoding rates for each codon were calculated as the inverse of codon DTs (1/DT) (Fang et al, 2018).

## Codon usage analysis

TPM expression values for coding genes in the custom reference set of APPRIS-annotated transcripts were estimated using RSEM v1.3.1. An RSEM reference for read alignment using STAR and the GRCz11 reference genome was built using *rsem-prepare-reference* with the *--star* option enabled. As input, this reference and trimmed RNA-seq reads were used with *rsem-calculate-expression* command. Finally, a merged table of transcript names and TPM values was generated from the outputs of each RSEM analysis. The resulting TPM expression values per sample per gene were used to weigh the frequencies of each of the 61 sense codons in each transcript in the custom annotation. Start AUG and coding sequence AUG codons were counted and normalized separately. The values were summed across all transcripts to generate usages per codon, and normalized per codon by the sum of all codons to generate proportional codon usage.

To compare weighted codon usage to tRNA anticodon abundance, proportional values for each tRNA anticodon family were calculated by dividing the raw reads uniquely aligned to a tRNA anticodon by the sum of reads mapped across all anticodons (similar to proportional values for codon usage). Codons were matched to cognate tRNAs based on codon:anticodon sequence complementarity. In cases where two synonymous codons are decoded by the same tRNA anticodon family via wobble (e.g. UUU and UUC codons decoded by tRNA-Phe-GAA, CAU and CAC decoded by tRNA-His-GTG, eight pairs U- and C-ending codons decoded by the eight tRNAs with inosine at position 34 across eukaryotes), the appropriate tRNA anticodon abundance was duplicated and matched with the wobble-decoded codon.

The Codon Adaptation Index (CAI) (Sharp and Li, 1987) was used to quantify the codon usage similarities between coding sequences in individual transcripts and a reference set composed by the top 5% most expressed genes (based on average TPM values in RNA-seq) at sphere (4 hpf) and bud (10 hpf) stages, respectively. CAI ranges between 0 and 1, with higher values indicating a CDS contains a higher proportion of codons used more frequently in the CDS of the top 5% most highly expressed genes at the corresponding developmental stage. To estimate the codon usage adaptation of each CDS to the tRNA pool at different developmental stages, an expression-based tRNA Adaptation Index (etAI) was calculated based on the tAI metric (dos Reis et al, 2004). In its original form, tAI measures the adaptation of a CDS to intracellular tRNA levels calculated from the number of nuclear tRNA genes predicted to encode tRNA transcripts with a given anticodon. As this index is static and does not account for fluctuations in tRNA abundance during development, we redefined tAI based on empirical tRNA abundance measurements by replacing gene copy number with the proportion of tRNA-mapped reads for each anticodon family from mim-tRNAseq datasets in the calculation. No selective constraints were imposed on the efficiency of codon–anticodon interactions during wobble decoding, as these are species-specific and unknown for zebrafish (Sabi et al, 2017). The relative adaptiveness values for each codon were also not scaled to the maximum one, as the identity of the most highly expressed tRNA anticodon family can differ at each developmental stage. As a result, higher etAI values for a given CDS reflect a higher proportion of codons matching highly abundant tRNA anticodon families at a particular developmental stage, but unlike tAI, etAI does not range from 0 to 1. The etAI, CAI, and decoding rate of each mRNA was calculated as the geometric mean of the etAI, CAI, and decoding rate values for codons in the corresponding CDS.

## mRNA half-life and CSC calculations

RNA-seq datasets from a time-course of WT embryos generated after rRNA depletion and in the absence of transcription inhibitors (Medina-Muñoz et al, 2021) were used for mRNA half-life calculations. From the 2539 mRNA transcripts annotated as M-stable or M-unstable in Bhat et al (2023), only those with TPM values > 10 at t = 0 and decreasing over time (with lower and upper bounds on the slope of the linear regression from *-inf to 0*) and with a linear fit that predicts well the temporal trend of TPM values ($p$-value < 0.05 with confidence interval 95%) were retained for analysis ($n = 1780$). mRNA half-lives for these transcripts were calculated by fitting a linear regression between log2(TPM) and time (0, 1, 2, 3, 4, 5, 6, 7, 8 h) using the *curve_fit* function in Scipy (v. 1.5.3). The mRNA decay rate constant ($k$) for each transcript was derived from the slope of the linear regression, and the mRNA half-life ($t_{1/2}$) = ln2/$k$. The Codon Stabilization Coefficient (CSC) was calculated as the Pearson correlation coefficient between occurrences of each codon in individual transcripts and their mRNA half-lives (Presnyak et al, 2015). Codons with a statistically significant ($p < 0.05$), positive Pearson correlation coefficient were defined as stabilizing, and codons with a statistically significant ($p < 0.05$) negative Pearson correlation coefficient were defined as destabilizing. Codon DT distributions at stabilizing or destabilizing codons were plotted with Raincloud (https://gabrifc.shinyapps.io/raincloudplots/).

## Whole-genome bisulfite sequencing and ATAC-seq data analysis

Public whole-genome bisulfite sequencing datasets (Jiang et al, 2013) for 1000-cell and germ ring ($n = 2$ for each stage) were obtained from GEO accession GSE44075 and public ATAC-seq datasets (Pálfy et al, 2020) were obtained from GEO accession GSE130944 for 256-cell (GSM3756599, GSM3756600), sphere (GSM3756614, GSM3756615), shield (GSM3756612, GSM3756613), and 80%-epiboly (GSM3756601, GSM3756602) stages using SRA Toolkit prefetch v3.0.0. Paired-end fastq files were then generated using *fastq-dump --split-files*. Methylation analysis was performed with the the nf-core/methylseq v2.6.0 pipeline using the Nextflow v23.10.1 client with default parameters.

For ATAC-seq analysis, Trim Galore v0.6.4 with default settings was used to find and remove 3′ adapters from sequencing reads. Trimmed reads of length ≥20 were then aligned to the GRCz11.108 genome using STAR v2.6.1c with the following parameters: up to

one mismatch per read, a maximum of 100 alignment positions, end-to-end alignment, prohibited introns, only one alignment reported per read (*--outSAMtype BAM SortedByCoordinate --outFilterMultimapNmax 100 --outSAMmultNmax 1 --outFilterMismatchNmax 1 --alignIntronMax 1 --alignendsType EndToEnd*).

Reads potentially originating from PCR duplicates were detected and removed using Picard Tools MarkDuplicates v2.17.10 with REMOVE_DUPLICATES=true to enable direct filtering of duplicates in the output bam file. Reads aligning to the mitochondrial genome ("MT" chromosome) were also filtered using samtools v1.11. Technical replicates for both 80%-epiboly and shield biological replicates, and biological replicate 1 of sphere were merged using samtools merge (remaining samples had only one technical replicate). deepTools alignmentSieve v3.4.0 was used to shift and filter these merged alignments. Specifically, shifts of +4 bp and −5 bp were applied for positive and negative strand alignments, respectively. This accounts for transposon dimerization before insertion (Buenrostro et al, 2013). Alignments were split by size using alignmentSieve, and nucleosome-free regions (NFRs) were retained as fragments less than 100 nt in length. Both shifting and length filtering operations were performed simultaneously using the *--ATACshift* and *--maxFragmentLength 100* parameters.

To compare nucleosome-free region occurrence at different developmental stages, bam files were converted into normalized bigWig signal tracks using deepTools v3.5.1. Normalization factors were first calculated using read counts around predicted tRNA genes (+−125 bp). Counts were computed using featureCounts v1.6.2, where fragments based on paired-end alignments were counted with permission for multi-mapping, and feature assignment was decided based on the tRNA feature with the largest overlap with the read (*-M -p --largestOverlap*). Then, using edgeR v3.34.1, these counts were provided as input for the calcNormFactors function using the "RLE" method. All reads aligned to tRNA features were summed per sample and scaled per million reads, and edgeR normalization factors were further multiplied by these library size factors, and the reciprocal of this product was used for normalized signal generation.

Normalized signal files were generated using deepTools bamCoverage with a normalization bin size of 1 bp (*--binSize 1*), the scale factors previously calculated (*--scaleFactor*), and read extension (*--extendReads*). For plotting this signal at different sets of tRNA genes, we extracted the set of transcripts significantly up- and down-regulated in bud (10 hpf) relative to unfertilized egg (0 hpf) generated by mim-tRNAseq. We then matched these transcripts to all of the genes which encode them in the zebrafish reference tRNA gene set. Plotting of this signal was performed with deepTools computeMatrix (in reference-point mode) and plotHeatmap, using the tRNA gene start as a reference (*--referencePoint TSS*), bed files of up- and down-regulated tRNAs as described above as regions (*-R*), and either 500 bp flanking the tRNA gene start (*-a 500 -b 500*).

## Data availability

Proteomics data have been deposited at MASSIVE (dataset MSV000094166, accession link https://massive.ucsd.edu/) and high-throughput sequencing datasets have been deposited in the Gene Expression Omnibus Database (dataset GSE241755, accession link https://www.ncbi.nlm.nih.gov/geo/query/acc.cgi?acc=GSE241755). The source data of this paper are collected in the following database record: biostudies:S-SCDT-10_1038-S44318-024-00265-4.

## Peer review information

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

## Acknowledgements

We thank K. Strasser for assistance with tRNA library preparation, M. N. Hall's lab for sharing antibodies, S. Shetty for invaluable technical know-how regarding polysome profiling, Alex Schier and Elena Conti for resources and advice, and the Schier lab and the Basel RNA Club for critical input and discussions. High-throughput sequencing was performed at the NGS Facility in the Department of Totipotency at the MPI of Biochemistry. Mass spectrometry data acquisition was performed at the Proteomics facility of the Biozentrum, University of Basel. MMR-P was supported by fellowships from EMBO (ALTF 691-2019) and MSCA-IF (H2020-MSCA-IF-2019 No. 898218) and SF by a postdoctoral fellowship from Alexander von Humboldt Foundation. This work was funded by the Max Planck Society, the European Research Council under the European Union's Horizon 2020 Research and Innovation Programme (ERC Starting Grant No. 803825-TransTempoFold, DDN), SFB 1035 (German Research Foundation DFG, Sonderforschungsbereich 1035, Project number 201302640, Project B14, DDN) and the EMBO Young Investigator Program (YIP 4833).

## Author contributions

**Madalena M Reimão-Pinto**: Conceptualization; Formal analysis; Investigation; Visualization; Methodology; Writing—original draft; Writing—review and editing. **Andrew Behrens**: Data curation; Software; Formal analysis; Investigation; Visualization; Methodology; Writing—review and editing. **Sergio**

**Forcelloni**: Data curation; Software; Formal analysis; Investigation; Visualization; Methodology; Writing—review and editing. **Klemens Fröhlich**: Formal analysis; Investigation; Visualization; Methodology; Writing—review and editing. **Selay Kaya**: Formal analysis; Investigation; Methodology; Writing—review and editing. **Danny D Nedialkova**: Conceptualization; Formal analysis; Supervision; Funding acquisition; Investigation; Visualization; Methodology; Writing—original draft; Writing—review and editing.

Source data underlying figure panels in this paper may have individual authorship assigned. Where available, figure panel/source data authorship is listed in the following database record: biostudies:S-SCDT-10_1038-S44318-024-00265-4.

## Funding

## Disclosure and competing interests statement

AB and DDN are inventors on a patent application filed by the Max Planck Society pertaining to the mim-tRNAseq technology. The other authors declare no competing interests.

# Expanded View Figures

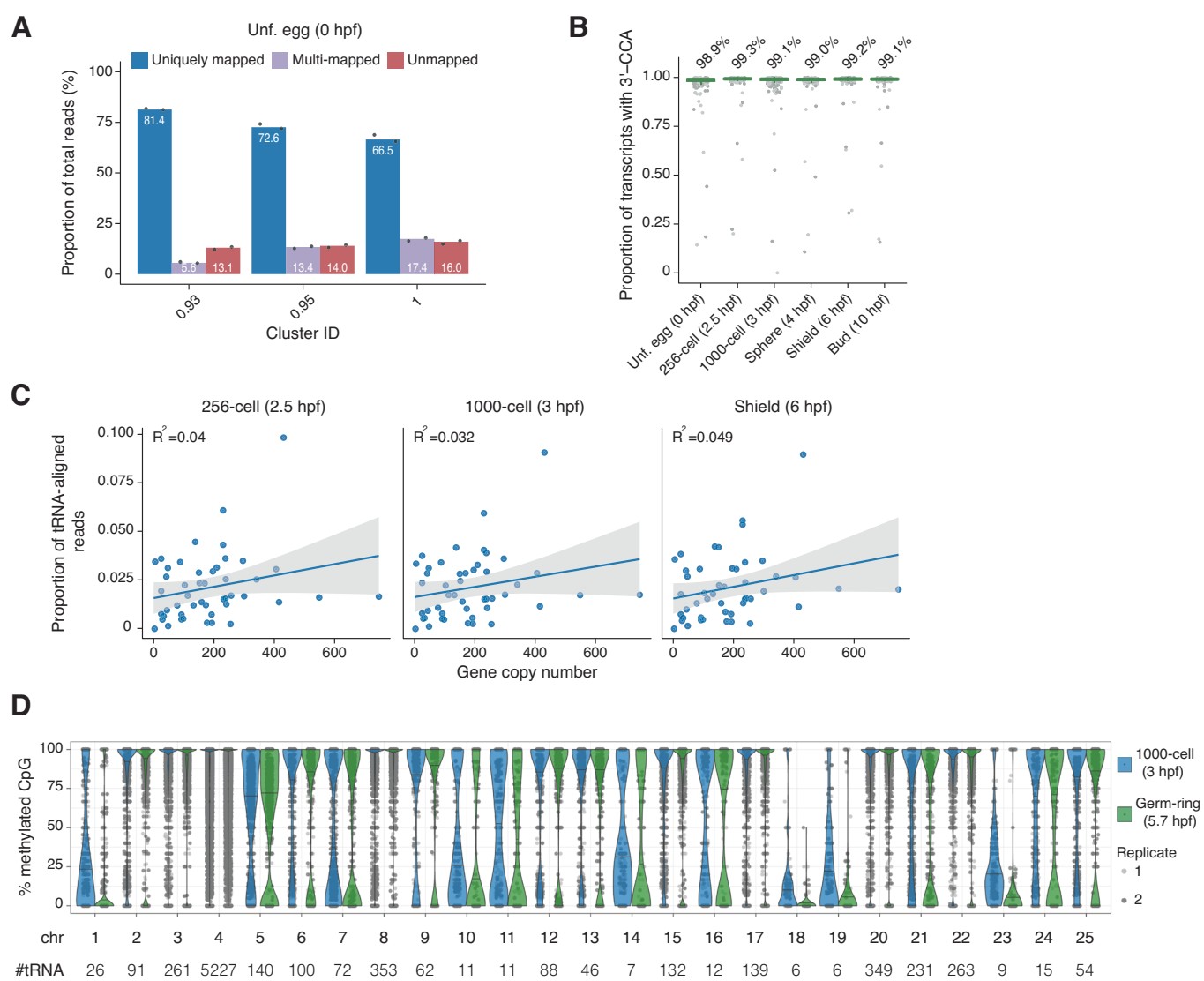

**Figure EV1.   Alignment rates of zebrafish tRNA sequencing libraries and correlation of tRNA levels with gene copy number.**

(**A**) Alignment statistics for the unfertilized egg (0 hpf) samples ($n = 2$) for different cluster ID thresholds using the mim-tRNAseq computational pipeline (Behrens and Nedialkova, 2022). (**B**) Box plots of the proportion of tRNA transcripts containing post-transcriptional 3′-CCA additions at each of the developmental time-points determined by mim-tRNA seq ($n = 2$; central line and label: median; box limits: upper and lower quartiles; whiskers: 1.5 × interquartile range). (**C**) Correlation plots of unique tRNA gene copy number and corresponding proportion of uniquely aligned tRNA reads in single replicates ($n = 1$) for the indicated developmental time-points. Blue lines: linear regression model; shaded gray: 95% confidence interval (CI); r1 and r2: replicate 1 and 2, respectively. (**D**) Violin plots of CpG methylation proportions at tRNA genes (+125 bp upstream sequence; center line: median, $n = 2$) per standard chromosome (chr) measured by whole-genome bisulfite sequencing of 1000-cell (3 hpf) and germ ring (5.7 hpf) embryos in two biological replicates (data from Jiang et al (2013)). The number of predicted high-scoring tRNA genes (Chan and Lowe, 2016) on each chromosome in the GRCz11 genome assembly is indicated.

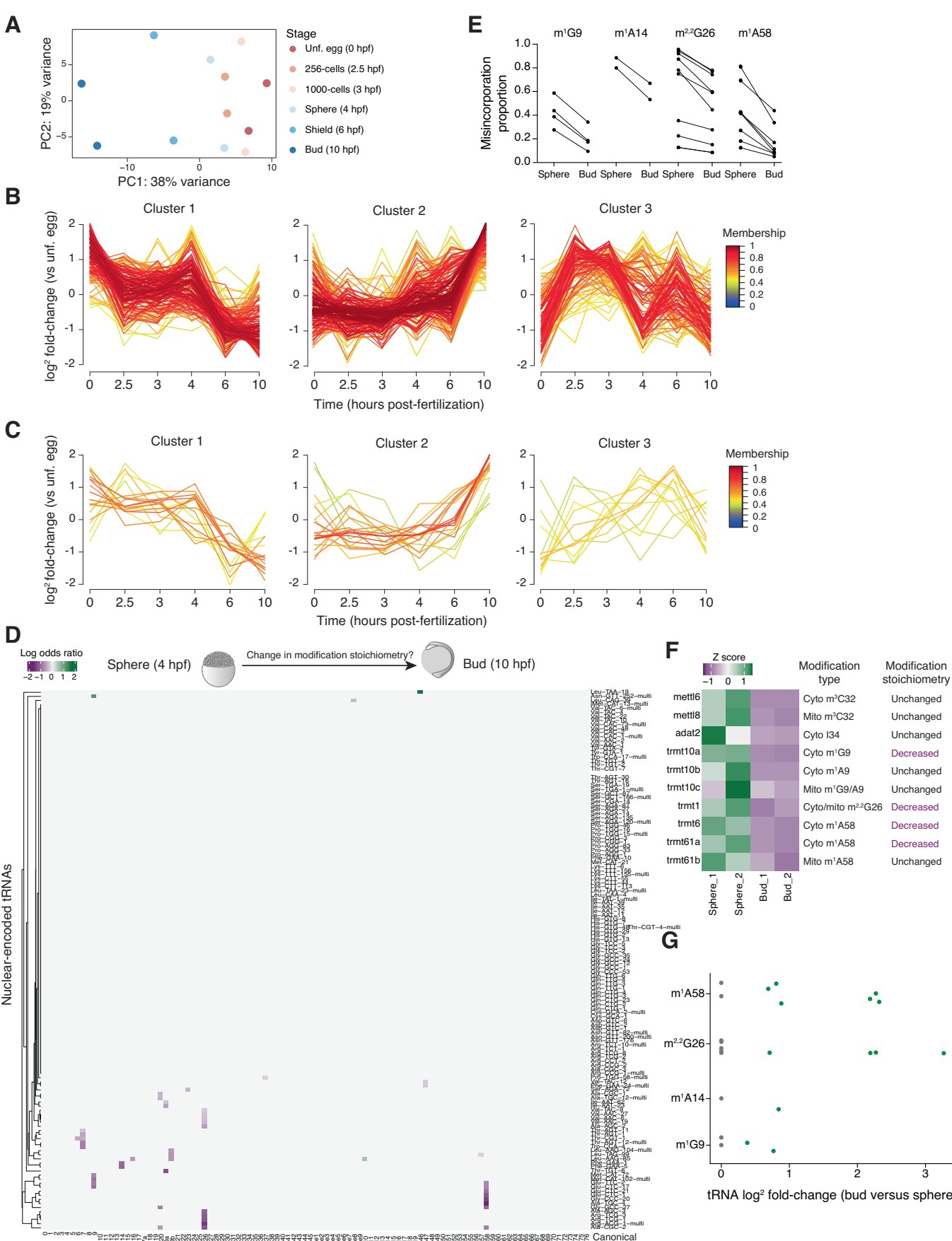

**Figure EV2. Analysis of tRNA transcripts, anticodon families, and differential modification stoichiometry during the zebrafish MZT.**

(A) Principal Component Analysis (PCA) plot of normalized unique tRNA transcript levels for all 12 mim-tRNAseq samples. (B) Fuzzy c-means soft clustering analysis of expression changes during the zebrafish MZT at the level of unique tRNA transcripts and (C) tRNA anticodon families. tRNAs for which cluster membership value is >0.6 were considered as part of the same cluster. (D) Hierarchically clustered global heatmap of log odd ratios of average misincorporation ($n = 2$) in unique cytosolic tRNA transcripts in bud (10 hpf) versus sphere (4 hpf) stages. Log odd ratios were filtered for significance (Chi-square FDR-adjusted $p$-value ≤ 0.01) and effect size (average misincorporation log2 fold-change ≥0.5) for sites detected as modified by mim-tRNAseq. Column names show canonical tRNA position. (E) Scatter plot of average ($n = 2$) misincorporation proportions in sphere (4 hpf) and bud (10 hpf) for the indicated modified sites in tRNAs with significantly different log odds ratios from (D). (F) Gene expression heatmaps for selected tRNA-modifying proteins in sphere (4 hpf) and bud (10 hpf) stages ($n = 2$). Standardized Z scores were calculated from DESeq2-normalized RNA-seq counts per gene across samples. Modification type and position in nuclear-encoded (cyto) and mitochondrial (mito) tRNA is shown along with annotation of significant changes in stoichiometry inferred from misincorporation rates analysis in (D). (G) Scatterplots of log2 fold change in abundance estimated by DESeq2 for tRNA transcripts and clusters with significant changes in misincorporation frequency at the indicated modified sites in bud versus sphere (from Fig. EV2D).

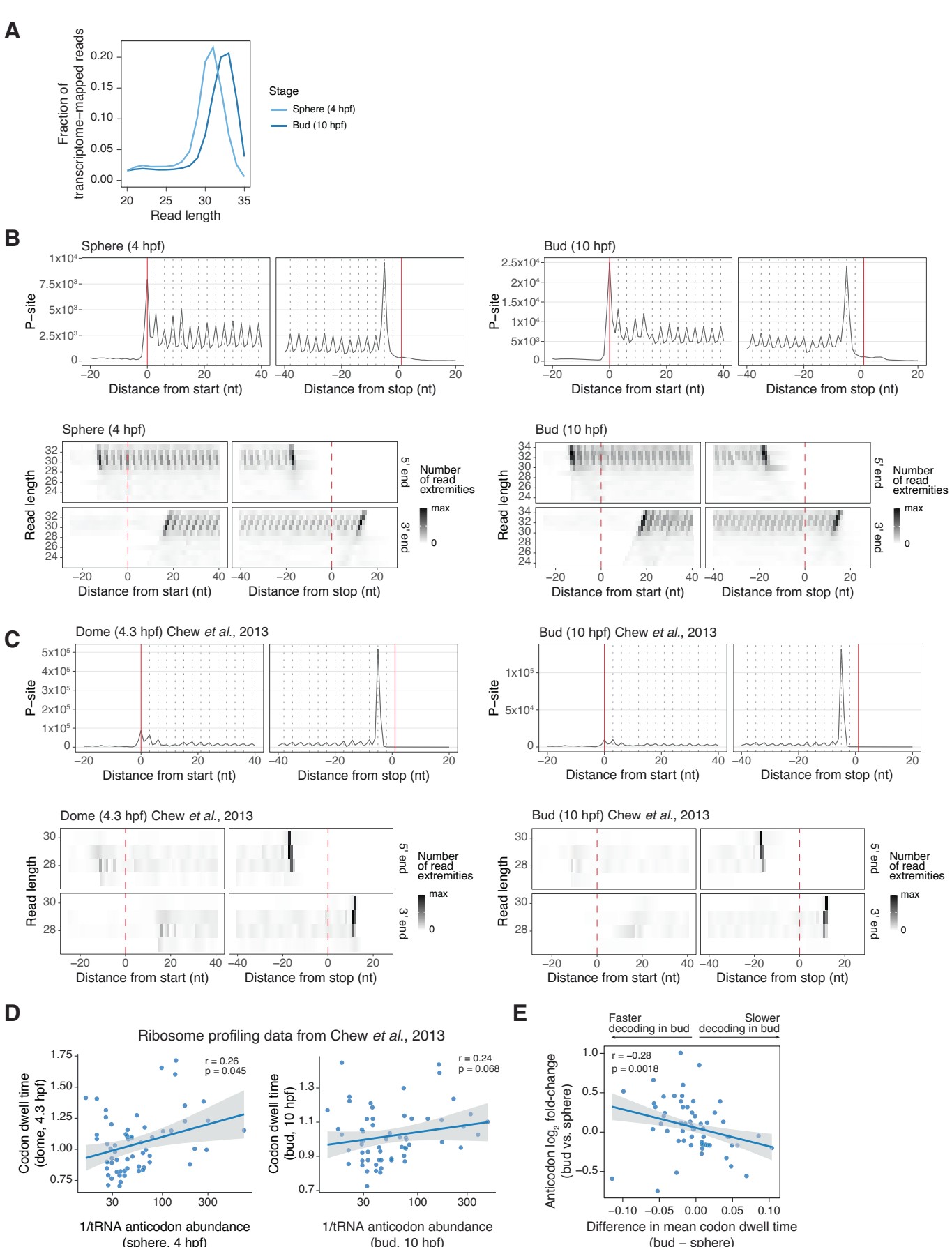

◀ **Figure EV3. Relationship between tRNA levels and ribosome decoding speed at the sphere and bud stages of zebrafish embryogenesis.**

(A) Representative read length distributions of ribosome footprints from sphere (4 hpf) and bud (10 hpf) stage samples generated by halting elongation with cycloheximide and tigecycline (B) Top: Meta-profiles depicting the periodicity of ribosome footprints at a transcriptome-wide scale at each stage of development based on P-site identification with riboWaltz. Bottom: Meta-gene heatmap depicting the signal associated to the 5′ and 3′ ends of the reads aligning around the start and stop codons for different ribosome footprint lengths at each stage of development. (C) Same as in (B) for publicly available ribosome profiling data by Chew et al (2013). (D) Correlation plots of codon-specific dwell times calculated with Scikit-ribo using ribosome profiling data from Chew et al (2013) generated by halting elongation only with cycloheximide. The abundance of cognate tRNA levels was determined by mim-tRNA seq (this study) at equivalent developmental time-points. Pearson's $r$ values and their associated $p$-values are shown. (E) Correlation plots of log2 fold changes in tRNA anticodon abundance at the bud (10 hpf) relative to the sphere (4 hpf) stage (p-adj ≤ 0.05) and differences in mean codon dwell time between the two stages. Solid lines: linear regression model; shaded gray: 95% confidence interval (CI); Pearson's $r$ value and associated $p$-value is shown.

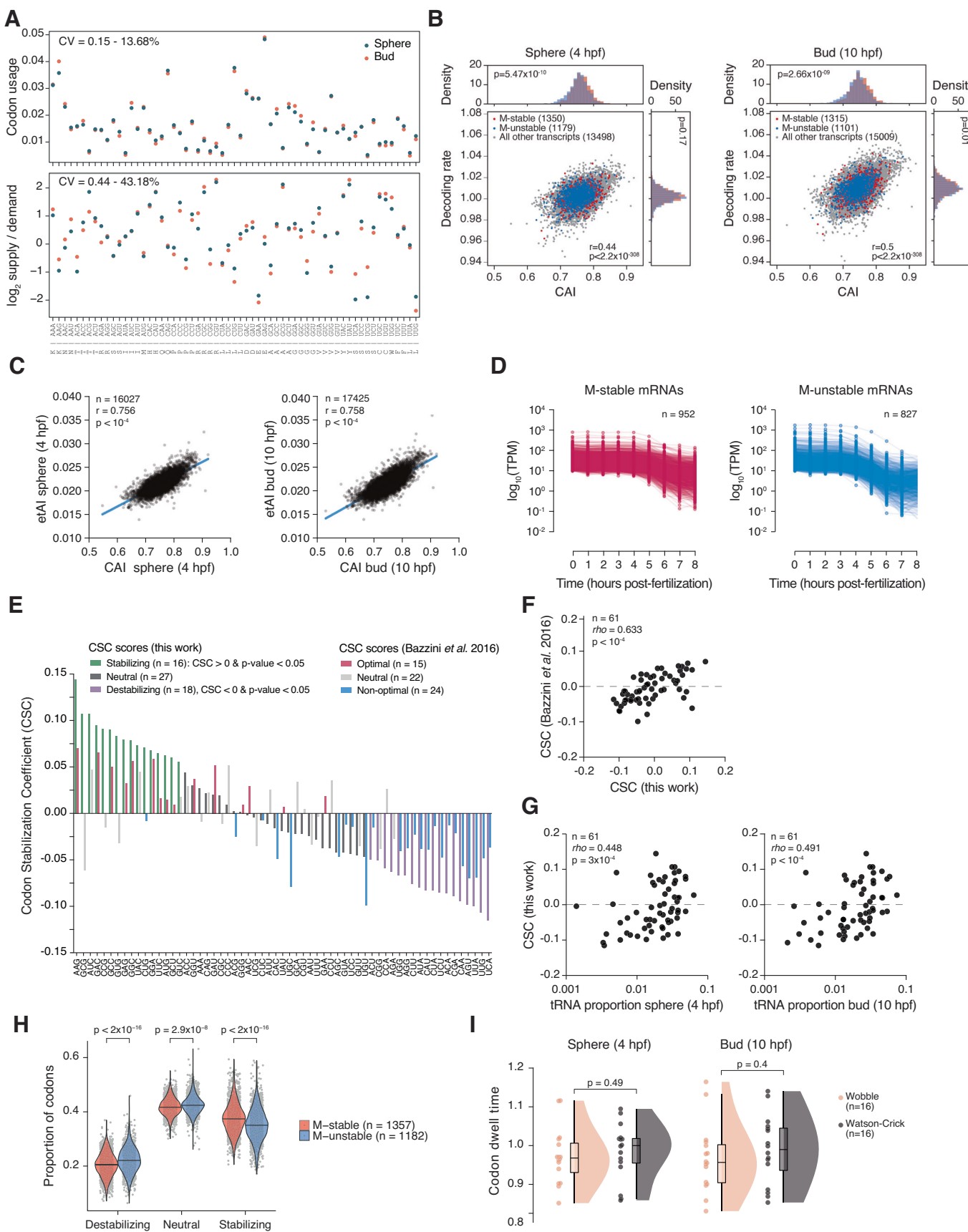

**Figure EV4. Relationship between mRNA codon content and codon optimality metrics (CAI and CSC) with decoding speed and tRNA levels.**

(A) Top: Mean aggregated codon usage weighted by transcript expression (TPM) across all reference transcripts ($n = 24{,}573$) based on RNA-seq ($n = 2$, this study). Values shown correspond to the mean weighted codon usage at the sphere (4 hpf) and bud (10 hpf) stages and are represented as proportions of total codon usage values. Bottom: log2 tRNA anticodon supply to codon demand ratios per codon. Mean tRNA anticodon abundance as a proportion of all tRNA-mapped reads from mim-tRNAseq for each stage ($n = 2$) were divided by proportional mean weighted codon usages for corresponding codons. X-axis labels: codon sequence and corresponding amino acid in single-letter code. (B) Correlation plots of Codon Adaptation Index (CAI) values (calculated from the top 5% most highly expressed genes) and decoding rates at the sphere (4 hpf) and bud (10 hpf) stages. Red dots represent maternal (M)-stable mRNAs, blue dots M-unstable mRNAs and gray dots all other expressed transcripts ($n = 16{,}027$ with mean TPM > 0.5 at 4 hpf; $n = 17{,}425$ with mean TPM > 0.5 at 10 hpf in RNA-seq). Marginal density histograms for transcript decoding rates and CAI values are shown for maternal (M)-stable and M-unstable transcripts (Mann–Whitney U Test $p$-values). (C) Correlation plots of expression-based tRNA adaptation index (etAI) values and Codon Adaptation Index (CAI) values at each stage of development. Pearson's correlation $r$ values are shown with their significance ($p$-value). (D) Transcript levels of maternal (M)-stable ($n = 952$, TPM > 0.5) and M-unstable ($n = 827$, TPM > 0.5) mRNAs (classified according to Bhat et al (2023)) over time that were used for determining mRNA half-lives (this study) during zebrafish MZT. TPM time-course data from Medina-Muñoz et al (2021). (E) Bar plot displaying the Codon Stabilization Coefficient (CSC) value of each codon and its associated codon effect on mRNA stability during the zebrafish MZT inferred by this study and in previous work (Bazzini et al, 2016). Codons with a statistically significant ($p < 0.05$), positive Pearson correlation coefficient between occurrences of each codon in individual transcripts and their mRNA half-lives were defined as stabilizing, and codons with a statistically significant ($p < 0.05$) negative Pearson correlation coefficient were defined as destabilizing (see Dataset EV4). (F) Scatter plot showing the relationship between CSC values inferred by this study and in previous work (Bazzini et al, 2016). Spearman's $rho$ value and associated $p$-value is shown. (G) Scatter plots showing the relationship between CSC values inferred by this work and corresponding anticodon tRNA proportions at each stage of development (4 and 10 hpf). Spearman's $rho$ values and associated $p$-values are shown (H) Violin plots representing the proportion of destabilizing, neutral and stabilizing codons in maternal (M)-stable transcripts ($n = 1357$, mean TPM > 0.5 at 4 and 10 hpf) and M-unstable transcripts ($n = 1182$, mean TPM > 0.5 at 4 and 10 hpf) expressed during the MZT (Mann–Whitney U Test $p$-values). (I) Raincloud plots of codon dwell times calculated from ribosome footprints for wobble-decoded ($n = 16$) and Watson-Crick decoded ($n = 16$) codons at the indicated developmental stage (Mann–Whitney U Test $p$-values). Center line of box plots: median; box limits: upper and lower quartiles; whiskers: 1.5 × interquartile range. Codon dwell times were calculated from one replicate of ribosome profiling data per time-point.

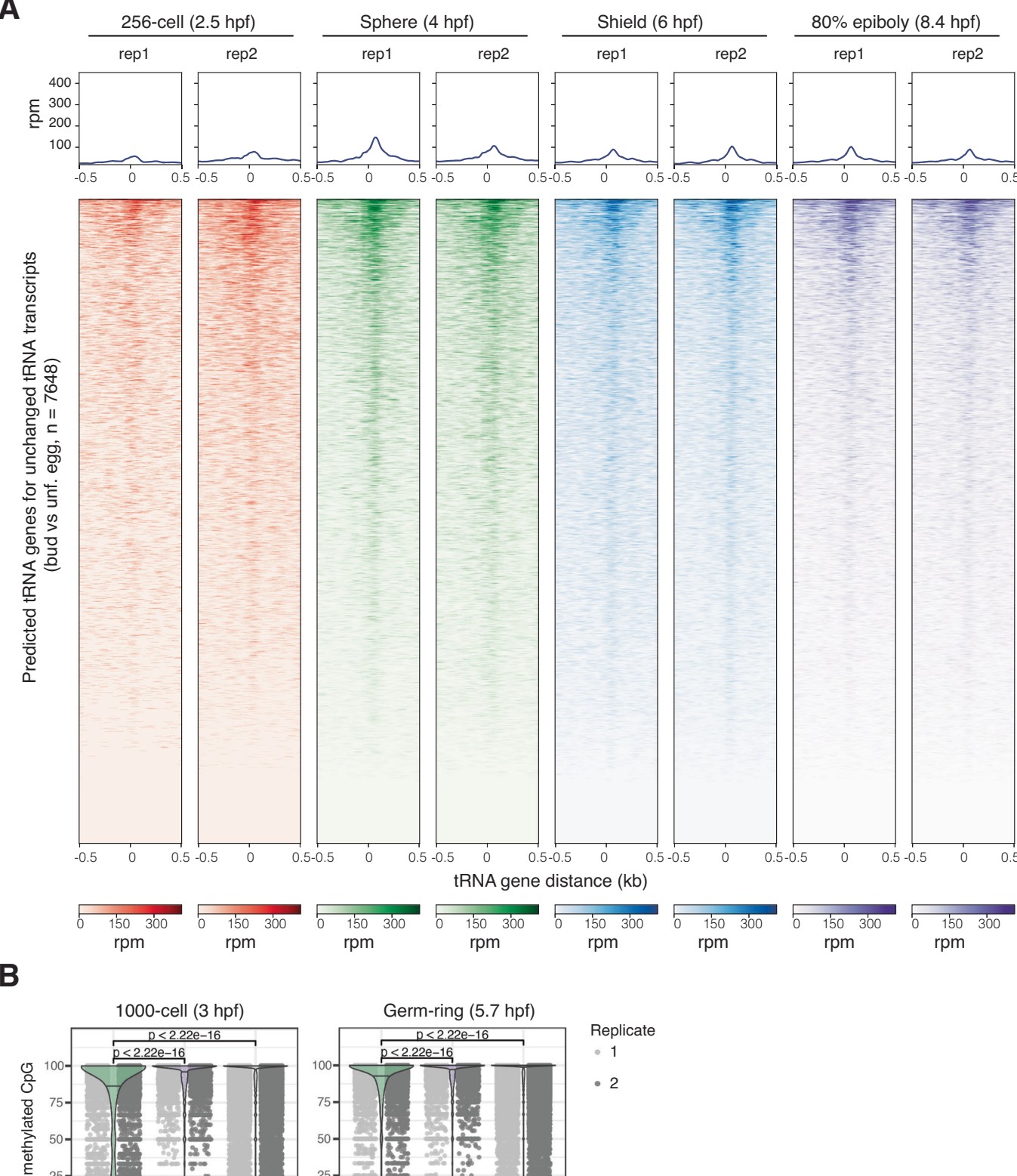

◀ **Figure EV5. Chromatin accessibility and methylation status of predicted tRNA genes during the zebrafish MZT.**

(A) Global changes in chromatin accessibility at tRNA genes encoding transcripts that do not change significantly in abundance in bud (10 hpf) vs unfertilized egg ($n = 7648$) for each stage (rpm: reads per million) during the zebrafish MZT. Top: Average normalized ATAC-seq nucleosome-free region signal intensity at predicted tRNA genes. Bottom: tRNA gene start site-centered heatmaps of chromatin accessibility. Bulk ATAC-seq data from Pálfy et al, (2020). (B) Violin plots of CpG methylation proportions at tRNA genes (+125 bp upstream sequence; center line: median) encoding transcripts upregulated ("Up"), downregulated ("Down"), or not differentially expressed ("non-DE") during the zebrafish MZT. Data from whole-genome bisulfite sequencing of 1000-cell (3 hpf) and germ ring (5.7 hpf) embryos in two biological replicates (Jiang et al, 2013). *P*-values are from Wilcoxon tests.

