## [Peer Review File · The EMBO Journal]

The dynamics and functional impact of tRNA repertoires during early embryogenesis in zebrafish

Danny Nedialkova, Madalena Madeira Reimão Pinto, Andrew Behrens, Sergio Forcelloni, Klemens Fröhlich, and Selay Kaya

Corresponding author(s): Danny Nedialkova (nedialkova@biochem.mpg.de), Madalena Madeira Reimão Pinto (madalena.pinto@unibas.ch)

Review Timeline:

Submission Date:	7th Mar 24
Editorial Decision:	14th May 24
Revision Received:	4th Aug 24
Editorial Decision:	13th Sep 24
Revision Received:	23rd Sep 24
Accepted:	27th Sep 24

Editor: Cornelius Schneider

Transaction Report:

Dear Prof. Nedialkova,

Thank you for submitting your manuscript for consideration by the EMBO Journal. It has now been seen by three referees whose comments are shown below.

As you will see from the reports, the reviewers appreciate the work, but also indicate a number of concerns that would require additional experiments. I think that the requested revisions are fair and reasonable and would therefore like to invite you to submit a revised version of the manuscript, addressing the comments of all three reviewers. I should add that it is EMBO Journal policy to allow only a single round of revision, and acceptance of your manuscript will therefore depend on the completeness of your responses in this revised version. I am happy to discuss the revision with you in more detail via email or phone/videoconferencing if any additional questions arise.

Thank you for the opportunity to consider your work for publication. I look forward to your revision.

Yours sincerely,

Cornelius Schneider, PhD
Editor
The EMBO Journal
c.schneider@embojournal.org

Please remember: Digital image enhancement is acceptable practice, as long as it accurately represents the original data and conforms to community standards. If a figure has been subjected to significant electronic manipulation, this must be noted in the

figure legend or in the 'Materials and Methods' section. The editors reserve the right to request original versions of figures and the original images that were used to assemble the figure.

We realize that it is difficult to revise to a specific deadline. In the interest of protecting the conceptual advance provided by the work, we recommend a revision within 3 months (12th Aug 2024). Please discuss the revision progress ahead of this time with the editor if you require more time to complete the revisions. Use the link below to submit your revision:

Referee #1:

In this manuscript, this Reimão-Pinto et al. address the contribution of tRNAs in determining the translation and codon-dependent RNA stability during zebrafish development. tRNAs are known to carry several mutations and it has been challenging to sequence them. Here, the authors, using their previously developed tRNA sequencing technique, mim-tRNAseq, characterize the tRNA repertoire across various stages of embryonic development. In doing so, they identify two distinct pools of tRNAs - maternally supplied and the zygotically transcribed tRNAs (~ 5hpf). While the authors show that the tRNA levels are in accordance with the codon demand during gastrulation and bud stage, the change in tRNA levels that occurs at the onset of gastrulation does sufficiently influence codon translation rates. Moreover, the authors show a relationship between "codon optimality" (based on CSC), cognate tRNA levels and their decoding rates, with exceptions. In addition, the authors claim that the zygotic expression of tRNAs is under the influence of mTOR activity that phosphorylates Maf1a/b thereby preventing its PolIII inhibitory activity.

Overall, this paper is both exciting and very timely given the growing interest in measuring the impact of tRNAs in regulating translation and codon optimality-mediated decay. However, there are some concerns that should be addressed:

Concerns:

1. It is interesting that only ~61% of the tRNAs sequenced are full length. It is unclear if the authors only considered full length tRNA sequences for their subsequent analysis.
2. In Fig 1G, the authors show a ~100% modification rate at A34 which raises the question if this is indeed a modification or a naturally occurring SNP. Moreover, while the authors explain a couple of modifications, it seems insufficient to gain a complete understanding of the tRNA modifications in zebrafish. Maybe they can move this figure to the supplementary as it does not sufficiently add to the major claims of the paper.
3. In Fig 2B, some of the replicates seems to have a lot of variation. It is understandable that sequencing tRNAs, especially with long protocols can introduce noise. Can the authors provide a correlation between replicates to get an understanding of the variation between replicates?
4. In Fig 2E, it might be worth it to plot trendlines for easier visualization. Also, it is unclear how the authors normalized the northern blots. Conventionally, a loading control such as U6 snRNA is used to normalize the blots. Moreover, the protocol that the authors used for Northern Blot is missing in the methods section.
5. Figure 3E, is the change in correlation from 0.42 to 0.59 a substantial change? The difference in maternal-Stable vs Maternal-Unstable also seems minimal.
6. In the manuscript the authors use an RNA-Seq dataset to derive CSC values. This may obscure the correlations, as endogenous transcript stabilities are influenced by UTRs. Moreover, with bulk RNA-Seq data, it is difficult to tease apart the contribution of transcription vs mRNA stability. The authors claim they use Ribo0 library. If this is the case, did the authors use spliced vs un-spliced RNA as a proxy for decay? Also, how well does the RNA-Seq derived CSC correlate with previously measured CSC?
7. In Fig. 4B, while there is a difference in the Codon DT between Stabilizing and destabilizing codons as expected, but interestingly, the neutral codons also exhibit the same codon DT as the stabilizing codons, can the authors speculate on this behaviour? How do the tRNA proportions in Fig 4C vary for neutral codons?
8. To conclusively show the effect of mTOR in the activation of tRNA transcription, the authors must perform a Northern blot for candidate tRNAs (those that increase in expression during gastrulation) in the presence/absence of Rapamycin.

Other minor comments:

1. Given the prevalence of tRNA-derived small RNAs (tsRNAs), and their expression in zebrafish, could the smaller sized sequences be tsRNAs?
2. Do mTOR levels change during gastrulation or is it just the increase in activity?
3. Table sS1: m1A58 shows only values of 8 transcripts not 10 (as mentioned in manuscript)
4. Line 295 Figure 4D should be 3D.

5. Line 276 - 290: could be reworked to make the message more clear.
6. Line 342: Figure 3E
7. Line 367: "and 18 a significant"

Referee #2:

In the manuscript, the authors systematically described the tRNA expression profile before and after the maternal-to-zygotic transition (MZT) of the zebrafish embryo. As the mim-tRNAseq could quantify both the expression level and modification status of each tRNA copy, the authors illustrate the signature of tRNA reprogramming during MZT from these two aspects. Based on their results, tRNA transcription is repressed before MZT while activated dramatically from the sphere stage. The nuclear-encoded tRNA profile is reprogrammed after MZT, with distinct tRNA profile at different stages, while the modification status remains quite stable. To verify the relationship between tRNA profile and translation, the authors established improved ribosome profiling and suggest that the changes on the tRNA level adapt the demand for translation increase and mRNA decay after MZT, but could not be the determining factor. Analysis with the published ATAC-seq data, the changes in tRNA expression might correlate with the selective chromatin accessibility of tRNA genes, although most tRNA genes remain inaccessible. Via molecular study, the transcription and translation activation after MZT might correlate with mTORC1 related translation and transcription regulation.

The novelty of this manuscript is systematic investigation of the tRNA expression profiles during zebrafish early embryogenesis. However, the connection between translation or mRNA stability and tRNA level is a long-standing subject. Although the investigation of the molecular mechanism behind tRNA transcription activity is interesting, additional proof is advised.

Major:

1. The authors' conclusion that zygotic tRNA expression is established after gastrulation onset conflicts with a previous report that the total mature tRNA level obviously increases from the 256-cell to the sphere stage in zebrafish embryos (Chen et al, 2021, PMID:34797706). Nevertheless, more convincing data revealing the onset of zygotic tRNA gene expression are needed. BrUTP incorporation experiment may help address that issue.
2. The relationships between global translation and codon/anticodon usage were deduced from omics data. Validation by exemplifying one or two specific mRNAs or tRNAs would strengthen the conclusion, through analysis of exogenous codon mutated mRNAs and morpholino knockdown of anticodon-specific tRNAs.
3. The results suggest that the translation efficiency is not highly correlated with the total tRNA level, but it is unknown whether it could be more positively correlated with the aminoacylated tRNA level. It was reported that some tRNAs, like tRNA-Ser and tRNA-Thr, have significantly lower aminoacylation levels in HEK293t cells and H1299 cells (Evans et al., 2017 and Davidsen et al., 2023). And in this manuscript, the authors found that tRNA-Ser-UGA and tRNA-Thr-UGU expression increased during gastrulation. There might be a possibility that the up-regulation of the tRNA level is a compensation for the low aminoacylation level. It is better to check the aminoacylation level of some specific tRNAs (like tRNA-Ser-UGA and tRNA-Thr-UGU) whose expression increased after MZT.
4. In Figure 5b the heatmap indicates that the average NRE signal of the up-regulated tRNAs (Bud stage to egg) gradually increased from the sphere to 80% epiboly stage. But from figure S5, the average NRE signal of all tRNAs at the sphere stage is equal to or even higher than the latter two stages. And the NRE signal on tRNA genes at the 256c stage is also quite obvious. This may suggest an underestimation of tRNA transcription during earlier stages. The transcription activity could be detected by using a more direct method, like labeling the nascent tRNAs or doing ChIP-seq with a PolIII antibody.
5. The phosphorylation status of Maf1a/1b changed obviously after rapamycin treatment, but it is unclear whether tRNA transcription is activated after treatment with rapamycin. More direct evidence is needed to prove that mTORC and Maf1a/1b repress tRNA transcription before MZT.
6. As mentioned in the cDNA preparation method, there is no de-acylation process. The aminoacylated tRNA might not be successfully ligated to 3' adaptor. There might be some bias toward tRNAs with different aminoacylation levels. It is better to compare the sequencing result with or without an individual de-acylation process.
7. As shown in figure 1C, the reads of some tRNAs, such as Tyr/Trp/Lys, are highly truncated at 5'end, which indicates a high level of modification at the truncated site. It is unclear how the modification data was calculated from these truncated reads of these tRNA isotypes.
8. The changes of whole amount of tRNAs were not mentioned. The up-/down-regulated tRNAs are classified by the relative proportion among tRNAs. If the whole tRNA proportion in total RNA is elevated during embryogenesis, some up-regulated genes might be underestimated. It is better to normalize the tRNA expression to the spike tRNA or total RNA.

9. Line 255, the citation "Most ribosomes in the early zebrafish embryo are in a translationally inactive state, and global translation gradually increases from 6 hpf onwards (Leesch et al., 2023)" is not accurate. In Leesch's paper (Fig. 1c), polysome fraction obviously increased at the 2 hpf (64-cell) stage compared to the 1 hpf (4-cell) stage; actually, polysome fraction slightly increased from the egg to 1 hpf.

Minor:

10. Line 4, "Developmental control is then handed over to the zygote ---" is not correct. It would be "--- to the zygotic genome control ---".

11. Fig. 1A, "Maternal mRNA demand vs maternal tRNA supply" should be "Maternal mRNA demand and maternal tRNA supply"; same change for "Zygotic mRNA demand vs zygotic tRNA supply".

12. Fig. 1B, E and F, The title of the Y-axis "Proportion ---" should be "Proportion of ---".

13. Lines 148-150, "The decrease in the proportion of mt-tRNAs at the shield and bud (6 and 10 hpf, respectively) stages could indicate a relative increase in nuclear-encoded tRNA levels throughout MZT.". This may also be explained by a decrease in the number of mitochondria per cell as embryonic development progresses.

14. Line 180, "--sequences; (Figure 2A).", in which the semicolon should be deleted.

15. Line 193, "the effect size for ---", in which the term "effective size" is inappropriate.

16. Figure 4a, some codons do not have correspondent tRNAs with exact anticodons, and codons are usually decoded in a wobble manner, then only the isotypes but not the isoacceptors could be correlated to the codon DT or stabilizing/destabilizing codons. The

17. Figure 6c "sphere 6hpf" label wrong (6 hpf is actually equivalent to the shield stage)

Referee #3:

Pinto et al 2024

The manuscript by Pinto et al sets out to quantitatively profile tRNA repertoire in zebrafish embryos during the maternal to zygotic transition. Traditionally tRNA quantification has been tricky, and to address many of the challenges associated with such analysis, Pinto et al employ a recently developed method called mim-tRNAseq. This method is combined with ribosome profiling to establish the functional impact of tRNA pools on mRNA decoding rates and stability. In addition to the general utility of the generated data sets, the authors make several significant insights. First, they find that bulk of zygotic tRNA expression is established as the embryo enters gastrulation (~5 hpf), rather than when bulk mRNA transcription initiates at ~3 hpf. Previous studies have suggested that a subset of maternal mRNAs is degraded in a translation-dependent manner based on their codon composition during the zebrafish MZT, and in this study the authors extend on this observation by demonstrating that it is maternal mRNAs with a codon content poorly matched to tRNA pools that are destabilized by slower decoding upon increases in protein synthesis during gastrulation. Mechanistically they suggest tRNA gene derepression is temporally coordinated by target of rapamycin complex 1 (TORC1) mediated inactivation of the RNA Polymerase III (Pol III) repressor Maf1a/b through phosphorylation, although this model is not experimentally confirmed.

Overall this study is a valuable contribution to the scientific literature that will be of use to those studying both tRNA regulation and early development. I have only minor concerns.

The authors note that they are using GRCz11 reference genome, but this reference genome is known to have gaps and possible duplications, particularly near centromeres as well as on chromosome 4 where many tRNA clusters are located. Similarly, the reference to the number of zebrafish tRNA genes (Chan and Lowe 2016) in the introduction also almost certainly relies on an incomplete genome. It could be helpful for the authors to make note of this point and any impacts on analysis.

In panel 2 D and H are relatively difficult to read, with it being particularly hard to discriminate between the three faintly shaded colors.

In Panel 2F it is somewhat difficult to discriminate between the colors for Ser and Ala on such small circles

In Figure 6E it is confusing to have values that are differently normalized on the same graph. Time course and Rapa treated vs DMSO should be broken out if they involve distinct normalization.

The authors convincingly show that inhibition of Torc1 prevents maf1a/b phosphorylation at least to some extent at some sites. While this seems like a plausible mechanism for regulating tRNA transcription via polIII, the authors don't really make that final connection experimentally. The case would be further strengthened by data showing Rapa leads to a change in the timing of bulk tRNA transcription if such experiments are possible.

At author's discretion:

There is also good DNA methylation data for early zebrafish development, it could be worth considering looking at methylation status of tRNAs with respect to their expression, could this be part of the explanation for delayed transcription?

EMBOJ-2024-117203 - Point-by-point response

We thank the editor and reviewers for their time and efforts on our manuscript, and for their positive reviews and constructive feedback. We have added new data and clarifications that address all the points raised below. The relevant sections of the revised manuscript are marked in blue.

Referee #1:

In this manuscript, this Reimão-Pinto et al. address the contribution of tRNAs in determining the translation and codon-dependent RNA stability during zebrafish development. tRNAs are known to carry several mutations and it has been challenging to sequence them. Here, the authors, using their previously developed tRNA sequencing technique, mim-tRNAseq, characterize the tRNA repertoire across various stages of embryonic development. In doing so, they identify two distinct pools of tRNAs - maternally supplied and the zygotically transcribed tRNAs (~ 5hpf). While the authors show that the tRNA levels are in accordance with the codon demand during gastrulation and bud stage, the change in tRNA levels that occurs at the onset of gastrulation does sufficiently influence codon translation rates. Moreover, the authors show a relationship between "codon optimality" (based on CSC), cognate tRNA levels and their decoding rates, with exceptions. In addition, the authors claim that the zygotic expression of tRNAs is under the influence of mTOR activity that phosphorylates Maf1a/b thereby preventing its PolIII inhibitory activity.

Overall, this paper is both exciting and very timely given the growing interest in measuring the impact of tRNAs in regulating translation and codon optimality-mediated decay.

We thank the reviewer very much for the positive feedback and their enthusiasm for our work.

However, there are some concerns that should be addressed:

Concerns:

1. It is interesting that only ~61% of the tRNAs sequenced are full length. It is unclear if the authors only considered full length tRNA sequences for their subsequent analysis.

We appreciate the opportunity to clarify our analysis workflow. We now specify in the *Results and Methods* section that we considered all uniquely mapped reads for subsequent analyses, regardless of their length. In our analysis in Figure 1D, "full length" requires complete 5' to 3' coverage of a tRNA without a single nucleotide missing from the 5' end. A median of 61%-72% uniquely mapped reads from nuclear-encoded tRNAs fulfill this strict definition across all stages of zebrafish embryogenesis we profiled, and we have previously shown that mim-tRNAseq

substantially outperforms other tRNA sequencing approaches in terms of coverage (Behrens *et al*, 2021).

2. In Fig 1G, the authors show a ~100% modification rate at A34 which raises the question if this is indeed a modification or a naturally occurring SNP. Moreover, while the authors explain a couple of modifications, it seems insufficient to gain a complete understanding of the tRNA modifications in zebrafish. Maybe they can move this figure to the supplementary as it does not sufficiently add to the major claims of the paper.

Position A34 in eukaryotic tRNAs with ANN anticodons is indeed invariantly modified to inosine (I), which leads to a G mismatch during sequencing. This modification is present in tRNA transcripts with ANN anticodons from eight anticodon families. I34 enables these tRNAs to read both U- and C-ending synonymous codons, as there are no corresponding GNN tRNAs encoded in the genome (Ehrlich *et al*, 2021). We annotate the presence of I34 in 28 tRNAs transcripts (Table EV1), and we thus consider it highly unlikely that the A to G misincorporations at all these positions are due to naturally occurring SNPs.

While there are currently no methods that enable a complete understanding of all tRNA modifications in an organism, we believe that our analysis in Figure 1G provides a valuable resource to the community. There are so far no modifications annotated for any zebrafish tRNA in the major RNA modifications database MODOMICS (<https://genesilico.pl/modomics/>). We now predict 327 zebrafish tRNA modifications based on RT misincorporation signatures (Table EV1). Moreover, we have previously shown that misincorporation proportions in mim-tRNAseq reflect the stoichiometry of a subset of tRNA modifications with near-complete RT readthrough (m¹G, m^{2,2}G, m¹A, I, m¹I, m³C, yW, acp³U) (Behrens *et al.*, 2021). Each of these modifications is present at conserved positions in multiple tRNA transcripts, where they play critical roles in maintaining tRNA structure and decoding accuracy. A wide range of human diseases are linked to tRNA modification defects (Suzuki, 2021) and zebrafish can be a useful vertebrate model system to identify the underlying molecular mechanisms; our analysis in Figure 1G identifies key similarities and differences in tRNA modification stoichiometry among eukaryotic model organisms that will guide such work in the future.

3. In Fig 2B, some of the replicates seems to have a lot of variation. It is understandable that sequencing tRNAs, especially with long protocols can introduce noise. Can the authors provide a correlation between replicates to get an understanding of the variation between replicates?

We have now analysed the correlation between DESeq2-normalised read counts per tRNA transcript (**Reviewer Figure 1A**) and tRNA anticodon family (**Reviewer Figure 1B**), which were used to generate the principal component analysis plots in Figure 2B and Figure EV2A. We find a Pearson's correlation of 0.95 - 0.99 between biological replicates from the same developmental stage, confirming the reproducibility of our tRNA abundance estimates.

Reviewer Figure 1

Reviewer Figure 1. Correlation plots between mim-tRNAseq datasets from biological replicates for individual tRNA transcripts (A) and tRNA anticodon families (B). Counts were normalised with DESeq2; Pearson's r is shown.

4. In Fig 2E, it might be worth it to plot trendlines for easier visualization. Also, it is unclear how the authors normalized the northern blots. Conventionally, a loading control such as U6 snRNA is used to normalize the blots. Moreover, the protocol that the authors used for Northern Blot is missing in the methods section.

We sincerely apologise for this omission and we thank the reviewer for pointing it out. We have now added a detailed northern blotting protocol to the *Methods* section of the manuscript. We loaded equal amounts of total RNA from each sample, and we now provide images of the SYBR Gold-stained gels prior to membrane transfer as new panels in Figure 2E to confirm equal loading. As we have no data on the dynamics of U6 abundance during zebrafish embryogenesis, we decided against using it (or other small RNA species such as 5S rRNA or 5.8S rRNA) as a loading control. Instead, we probed for two tRNA transcripts that are significantly upregulated (tRNA-Ser-UGA-19 and tRNA-Thr-UGU-8) and one that remains unchanged (tRNA-Ala-AGC-3) during early (tRNA-Ala-AGC-3) based on our mim-tRNAseq data. We then normalised the band intensity in each lane to the average signal for the same probe in the two replicates from unfertilized eggs, which we denote on the y axis of the plot in Figure 2F. This analysis is analogous to the one we performed to quantify differential tRNA expression by mim-tRNAseq, which we also performed in comparison to unfertilized eggs. To maintain the readability of the plot in Figure 2F that contains data from both northern blotting and mim-tRNAseq for all three tRNAs, we opted not to include trendlines.

5. Figure 3E, is the change in correlation from 0.42 to 0.59 a substantial change? The difference in maternal-Stable vs Maternal-Unstable also seems minimal.

We have now compared the difference in magnitude between the two correlations in Figure 3E with the *cocor* R package (Diedenhofen & Musch, 2015), and found it to be statistically significant ($p < 0.001$, Fisher's tests on z-statistics). The difference in decoding rates of maternal-stable and maternal-unstable mRNAs we find in bud stage embryos in Figure 3E is also statistically significant.

6. In the manuscript the authors use an RNA-Seq dataset to derive CSC values. This may obscure the correlations, as endogenous transcript stabilities are influenced by UTRs. Moreover, with bulk RNA-Seq data, it is difficult to tease apart the contribution of transcription vs mRNA stability. The authors claim they use Ribo0 library. If this is the case, did the authors use spliced vs un-spliced RNA as a proxy for decay? Also, how well does the RNA-Seq derived CSC correlate with previously measured CSC?

We appreciate the opportunity to clarify our rationale and workflow for CSC calculations, which we derived from maternal mRNAs. Throughout the manuscript, we were specifically interested in how tRNA pools impact the codon-dependent destabilisation of a subset of maternal mRNAs: this depends on active translation, and only manifests at later stages of the zebrafish MZT (from 4 - 6 hpf onward) (Bazzini *et al*, 2016; Medina-Muñoz *et al*, 2021; Mishima *et al*, 2022; Mishima & Tomari, 2016). We therefore asked if a reprogramming of tRNA pools underlies the timing of codon-dependent maternal mRNA decay. We focused exclusively on endogenous mRNAs because we wanted to maintain the native 5' and 3' UTR regulatory context, which strongly influences ribosome loading (Bazzini *et al*, 2012; Lewis *et al*, 2024; Reimão-Pinto *et al*, 2023; Strayer *et al*, 2023), and poly(A) tail length during the MZT (Begik *et al*, 2023; Subtelný *et al*,

2014). Uncoupling coding sequences from their native UTRs may thus also uncouple the timing and extent of their translational regulation during the MZT.

To circumvent the difficulty in deducing the contribution of transcription and turnover to mRNA abundance measurements by bulk RNA-Seq, we only analysed the half-lives of zebrafish mRNAs that were recently annotated as strictly maternally provided in the embryo. This annotation was based on metabolic labelling of newly made mRNAs in zebrafish embryos with 4-thiouridine and maternally deposited mRNAs were defined as transcripts with no detectable label incorporation during the MZT (Bhat *et al.*, 2023). Based on this definition, Bhat *et al.* discovered two classes of maternal mRNAs: maternal-stable, which persist until 5.5 hpf, and maternal-unstable, with decreasing levels from ~4 hpf onward. As both classes of mRNAs are not transcribed during early zebrafish embryogenesis, we reasoned that we can use published RNA-Seq time-course data from wild-type embryos in unperturbed conditions to calculate their half-lives. This approach has the added benefit of not relying on α -amanitin to shut off Pol II transcription, as multiple studies have shown that this inhibitor can also block Pol III. We explain this rationale in the *Results* section of the manuscript (page 10) and we note that our half-life calculations from RNA-Seq data faithfully capture the accelerated decay of maternal-unstable transcripts from ~4 hpf onward (Figure EV4D).

We now also provide new data to show that our maternal mRNA decay-derived CSC scores display a significant positive correlation (Spearman's $\rho=0.63$, $p<0.001$) with previous CSC calculations in zebrafish embryos after α -amanitin transcriptional shut-off by Bazzini *et al.* (2016).

7. In Fig. 4B, while there is a difference in the Codon DT between Stabilizing and destabilizing codons as expected, but interestingly, the neutral codons also exhibit the same codon DT as the stabilizing codons, can the authors speculate on this behaviour? How do the tRNA proportions in Fig 4C vary for neutral codons?

We have now plotted the variation of tRNA proportions for neutral codons as new data in Figure 4C. We observe similar results as for codon dwell times in Figure 4B: destabilising codons are decoded by tRNAs with significantly lower proportions than those matching neutral or stabilising codons, but there is no statistically significant difference between the proportions of tRNAs matching neutral and stabilising codons. We agree with the reviewer that this is intriguing, and we interpret these findings to indicate that destabilising codons act in a tRNA-dependent manner, while the stabilising codon effects may be related to mRNA nucleotide composition rather than tRNA availability (page 11 and final paragraph of the *Discussion*).

8. To conclusively show the effect of mTOR in the activation of tRNA transcription, the authors must perform a Northern blot for candidate tRNAs (those that increase in expression during gastrulation) in the presence/absence of Rapamycin.

We appreciate the reviewer's suggestion, but we consider this experiment unlikely to yield conclusive data for the following reasons:

- Our phosphoproteomics data show that incubation of embryos in rapamycin-containing medium between 3 hpf and 6 hpf significantly reduces but does not abolish Maf1a/b phosphorylation at a subset of sites (Figure EV6G, H). The phosphorylation of another well-established direct target of TORC1 - Rps6kb1a - is similarly decreased but not abolished under the same experimental conditions (Figure EV6F). This is in line with prior data showing that rapamycin inhibits mTORC1 target phosphorylation only incompletely (Benjamin *et al*, 2011), and the incomplete inhibition of human MAF1 phosphorylation by rapamycin *in vitro* or in cultured cells (Gao *et al*, 2024; Michels *et al*, 2010). The route of rapamycin administration in our experiments could further decrease its activity, since we added the drug to the embryo culture medium at 3 hpf. We chose this delivery method over drug microinjection in one-cell embryos because of the large number of embryos required for Maf1a/b phosphorylation analysis by mass spectrometry (several thousand). However, it is unclear how well rapamycin is absorbed from the medium, particularly after the onset of gastrulation when the three germ layers begin to form. The partial dephosphorylation of TORC1 targets we observe at 6 hpf in the presence of rapamycin could thus also be partially due to poor drug penetrance in gastrulating embryos. Finally, human FKBP1A and zebrafish Fkbp1a/b, which are the protein targets of rapamycin (Saxton & Sabatini, 2017), share <82% sequence identity. Sequence differences in the target proteins may thus also contribute to lower inhibitory activity of rapamycin on zebrafish TORC1.
- Differences in phosphopeptide ionisation efficiency do not allow us to measure the total fraction of Maf1a and Maf1b proteins that remains phosphorylated in rapamycin-treated embryos by mass spectrometry. We have so far only been able to assay Maf1a/b phosphorylation by this highly labor- and cost-intensive method, which requires several hundred embryos per replicate. The low sequence identity between zebrafish Maf1a/b and mammalian MAF1 (75%) have so far precluded the use of more straightforward assays such as Phos-tag gel electrophoresis followed by western blotting, which we recently used to assay MAF1 phosphorylation status in human cells (Gao *et al.*, 2024). We tested four commercially available MAF1 antibodies, none of which detected zebrafish Maf1a/b (see *Methods*).

As it is unclear what fraction of phosphorylated Maf1 is required to abolish its Pol III repressor activity in zebrafish embryos, the incomplete inhibition of Maf1a/b phosphorylation by rapamycin in our view precludes the use of this drug for establishing the contribution of TORC1 signaling to the activation of tRNA transcription.

Given that Maf1a and Maf1b transcripts are most likely maternally provided (and zygotically re-expressed) since their mRNA levels do not change between 0 hpf and 8 hpf (**Reviewer Figure 2**), and that Maf1b protein is likely also maternally deposited (Figure 6D), the role of Maf1a/b in tRNA activation at the onset of gastrulation in our view should be tested by generating double knockout zebrafish lines, an effort that we estimate will require at least 12 months. We note that

in other recent work, we established a link between TORC1 signalling, MAF1, and selective tRNA gene expression during differentiation *in human cells* by MAF1 depletion with CRISPR interference (Gao *et al.*, 2024).

Reviewer Figure 2

Reviewer Figure 2. mRNA levels (in TPM) for *Maf1a* and *Maf1b* in a wild-type zebrafish embryo developmental time course. Data from RNA-Seq samples generated after rRNA depletion (Medina-Muñoz *et al.*, 2021).

Other minor comments:

1. Given the prevalence of tRNA-derived small RNAs (tsRNAs), and their expression in zebrafish, could the smaller sized sequences be tsRNAs?

We consider this unlikely since we purified RNA of 60 to 100 nt by gel size selection prior to library construction to avoid amplifying tsRNAs, which are usually <40 nt in size.

2. Do mTOR levels change during gastrulation or is it just the increase in activity?

While proteomic time-course data from zebrafish embryos is not available, we analysed *mtor* mRNA levels in data from an RNA-Seq time-course and we found that they remain stable between the unfertilized egg stage (0 hpf) and 75% epiboly stages (8 hpf, **Reviewer Figure 3**). This suggests that the increased phosphorylation of TORC1 targets we observe starting at 6 hpf (Figure 6A) is driven by increased TORC1 activity.

Reviewer Figure 3

Reviewer Figure 3. mRNA levels (in TPM) for *Maf1a* and *Maf1b* in a wild-type zebrafish embryo developmental time course. Data from RNA-Seq samples generated after rRNA depletion (Medina-Muñoz *et al.*, 2021).

3. Table EVS1: m1A58 shows only values of 8 transcripts not 10 (as mentioned in manuscript).

Thank you for pointing this out; the manuscript text has been corrected.

4. Line 295 Figure 4D should be 3D.

Thank you; the text has been corrected.

5. Line 276 - 290: could be reworked to make the message more clear.

We have now re-written this part of the text to clarify our message.

6. Line 342: Figure 3E

Thank you; the text has been corrected.

7. Line 367: "and 18 a significant"

Corrected.

Referee #2:

In the manuscript, the authors systematically described the tRNA expression profile before and after the maternal-to-zygotic transition (MZT) of the zebrafish embryo. As the mim-tRNAseq could quantify both the expression level and modification status of each tRNA copy, the authors illustrate the signature of tRNA reprogramming during MZT from these two aspects. Based on their results, tRNA transcription is repressed before MZT while activated dramatically from the sphere stage. The nuclear-encoded tRNA profile is reprogrammed after MZT, with distinct tRNA profile at different stages, while the modification status remains quite stable. To verify the relationship between tRNA profile and translation, the authors established improved ribosome profiling and suggest that the changes on the tRNA level adapt the demand for translation increase and mRNA decay after MZT, but could not be the determining factor. Analysis with the published ATAC-seq data, the changes in tRNA expression might correlate with the selective chromatin accessibility of tRNA genes, although most tRNA genes remain inaccessible. Via molecular study, the transcription and translation activation after MZT might correlate with mTORC1 related translation and transcription regulation.

The novelty of this manuscript is systematic investigation of the tRNA expression profiles during zebrafish early embryogenesis. However, the connection between translation or mRNA stability and tRNA level is a long-standing subject. Although the investigation of the molecular mechanism behind tRNA transcription activity is interesting, additional proof is advised.

We thank the reviewer for the encouraging feedback and constructive suggestions. We note that our manuscript presents the first in-depth characterization of the regulatory impact of endogenous tRNA repertoires on the decoding rates and stability of endogenous mRNAs.

Collectively, our data suggest that increased translational activity - rather than a switch in tRNA availability - underlies the codon-dependent destabilisation of maternal mRNAs in zebrafish embryos. Our findings thus provide key novel insights into the long-standing question of how the interplay between tRNA pools and codon content impacts mRNA stability in metazoans.

Major:

1. The authors' conclusion that zygotic tRNA expression is established after gastrulation onset conflicts with a previous report that the total mature tRNA level obviously increases from the 256-cell to the sphere stage in zebrafish embryos (Chen et al, 2021, PMID:34797706). Nevertheless, more convincing data revealing the onset of zygotic tRNA gene expression are needed. BrUTP incorporation experiment may help address that issue.

We thank the reviewer for the suggestions and for drawing our attention to the work of Chen and colleagues, which we have cited in the revised manuscript. In their publication, Chen et al. state that “*the level of total tRNAs exhibited a marked increase from the sphere to the shield stage, although not much changes happened from the 1c to the sphere stage (Fig. 4A)” . This conclusion is in line with our DESeq2 data in Figure 2 (panels B,C,G), which also places the onset of zygotic tRNA transcription between the sphere (4 hpf) and the shield (6 hpf) stage. Our data thus agree with the timing of zygotic tRNA expression in zebrafish embryos suggested by Chen and colleagues, as well as with a recent report published during the revision of our manuscript that also places it at the onset of gastrulation (Rappol et al, 2024).*

To our knowledge, BrUTP has not been used for nascent RNA analysis in zebrafish embryos, probably because metabolic labelling with such nucleotide analogs is challenging as the signal is detected following antibody enrichment. The antibody epitope however will likely be inaccessible when BrUTP is present in the highly structured mature tRNA molecules, leading to inaccurate estimates of nascent tRNA levels. Similar challenges would limit the use of other nucleotide analogs, which is why we are not aware of published methods for quantifying nascent tRNA production.

2. The relationships between global translation and codon/anticodon usage were deduced from omics data. Validation by exemplifying one or two specific mRNAs or tRNAs would strengthen the conclusion, through analysis of exogenous codon mutated mRNAs and morpholino knockdown of anticodon-specific tRNAs.

We thank the reviewer for the suggestion. We would like to emphasise that our work is the first study to probe the relationship between endogenous tRNA supply and endogenous mRNA codon demand in the dynamic process of vertebrate embryogenesis. Previous work in zebrafish embryos has already conclusively established the impact of changing tRNA anticodon availability (by interfering with aminoacylation) on synthetic reporter mRNA stability (Mishima et al., 2022). In this work, Mishima and colleagues resorted to expressing the bacterial enzyme AnsB, which hydrolyzes asparagine (Asn) to aspartic acid, to specifically reduce aminoacylated tRNA-Asn

levels. This is likely because the highly structured nature of mature tRNAs and their sequence similarity poses substantial challenges to their depletion by morpholinos. We note that a prior study that attempted morpholino-based tRNA knockdown found no effects on the function or stability of mature tRNAs in zebrafish embryos (Chen *et al*, 2021). Other nucleic acid probe-based approaches such as LNAs have also been shown to be ineffective for targeting tRNAs *in vivo* (Kim *et al.*, 2017 PMID: 29186115).

3. The results suggest that the translation efficiency is not highly correlated with the total tRNA level, but it is unknown whether it could be more positively correlated with the aminoacylated tRNA level. It was reported that some tRNAs, like tRNA-Ser and tRNA-Thr, have significantly lower aminoacylation levels in HEK293t cells and H1299 cells (Evans *et al.*, 2017 and Davidsen *et al.*, 2023). And in this manuscript, the authors found that tRNA-Ser-UGA and tRNA-Thr-UGU expression increased during gastrulation. There might be a possibility that the up-regulation of the tRNA level is a compensation for the low aminoacylation level. It is better to check the aminoacylation level of some specific tRNAs (like tRNA-Ser-UGA and tRNA-Thr-UGU) whose expression increased after MZT.

We appreciate the reviewer's point that aminoacylation levels are an important consideration when quantifying tRNA availability. We have therefore now performed a global quantitative analysis of tRNA charging in sphere (4 hpf) and bud (10 hpf) embryos (new data in Figure 3D). This analysis revealed that all tRNA anticodon families display >80% charging at both developmental stages, and we found no statistically significant differences in tRNA charging in samples from embryos before (4 hpf) or after gastrulation (10 hpf; n=3 biological replicates from each stage). These new data suggest that the upregulation of specific tRNA anticodon families at the onset of gastrulation does not serve to compensate for their low aminoacylation level.

We note that in contrast to human HEK293T cells and H1299 cells, where charging levels of tRNAs that carry Ser or Thr have indeed been reported to be lower than those of other tRNA isoforms, we did not observe such a trend in zebrafish embryos (new data in Figure 3D).

4. In Figure 5b the heatmap indicates that the average NRE signal of the up-regulated tRNAs (Bud stage to egg) gradually increased from the sphere to 80% epiboly stage. But from Figure EV5, the average NRE signal of all tRNAs at the sphere stage is equal to or even higher than the latter two stages. And the NRE signal on tRNA genes at the 256c stage is also quite obvious. This may suggest an underestimation of tRNA transcription during earlier stages. The transcription activity could be detected by using a more direct method, like labeling the nascent tRNAs or doing ChIP-seq with a PolIII antibody.

In our initial submission, the y axis of the plots in Figure 5 and Figure EV5 unfortunately did not have the same range, which precluded their direct comparison. We have now re-plotted these data using the same y-axis range, and we also separated predicted tRNA genes into up- or downregulated during embryogenesis (Figure 5) and not differentially expressed during embryogenesis (Figure EV5A). A direct comparison between the plots in Figure 5 and Figure

EV5A is now possible, and in our view it suggests that nucleosome-free region (NFR) signal gradually increases at genes coding for tRNAs upregulated during embryogenesis.

We appreciate the reviewer's point that this increase is already detectable at the sphere stage (4 hpf) in the global analysis in Figure 5B, and we have amended the main text accordingly. These data are compatible with our observations that tRNA transcript pools do not change significantly until ~6 hpf, as *increased chromatin accessibility generally precedes the transcription of associated genes* during the zebrafish MZT (Pálffy *et al*, 2020). We note that a very small fraction of predicted tRNA genes have NFR signal at the 256-cell stage (2.5 hpf). It is conceivable that this minor fraction may overlap Pol II-transcribed regions of the zebrafish genome that are activated during ZGA.

We resorted to ATAC-Seq as a proxy for tRNA gene activity during zebrafish embryogenesis because Pol III ChIP-Seq in zebrafish embryos is hampered by the lack of suitable antibodies. In our recent work in human stem-cell based models (Gao *et al.*, 2024), we established a low-input ChIP-Seq workflow for Pol III using a *commercial monoclonal antibody against RPC1*, the catalytic core of the 17-subunit Pol III complex. This was the only ChIP-grade antibody (out of 6 we tested) that yielded a robust and reproducible enrichment of Pol III target genes in *ChIP-Seq*. Unfortunately, its epitope is not preserved in zebrafish Rpc1 since the protein is not detectable with this antibody in western blotting (**Reviewer Figure 4**). Pol III ChIP-seq in zebrafish embryos will therefore require *the generation and validation of ChIP-grade antibodies against zebrafish Rpc1*, or the generation of biallelic *epitope-tagged Rpc1 fish lines* - an effort that based on our experience will require considerable resources and take at least *9 to 12 months*.

Reviewer Figure 4

Reviewer Figure 4. Western blotting analysis of protein extracts from a zebrafish developmental time course and from human HEK293 cells. The time course was performed in two biological replicates and membranes were probed with ChIP-grade anti-RPC1 monoclonal antibody validated in human cells (Gao *et al.*, 2024). The molecular weight of zebrafish Rpc1 and human RPC1 is 156 kDa.

5. The phosphorylation status of Maf1a/1b changed obviously after rapamycin treatment, but it is unclear whether tRNA transcription is activated after treatment with rapamycin. More direct evidence is needed to prove that mTORC and Maf1a/1b repress tRNA transcription before MZT.

We appreciate the reviewer's point and we have highlighted the technical challenges of addressing it in our response to Reviewer#1, point 8. We also note that we recently established a direct link between TORC1 signaling, MAF1, and selective tRNA gene expression in human cell differentiation (Gao *et al.*, 2024). Nevertheless, we have refrained from inferring causality in zebrafish throughout the manuscript and we instead refer to the temporal coincidence of TORC1 signaling increase and zygotic tRNA transcription at the onset of gastrulation. We would also like to emphasise that our key conclusion - that codon-dependent decay of maternal mRNAs is driven by increased global translation and not by tRNA pool changes - is independent of the precise mechanism(s) by which zygotic tRNA transcription is activated.

6. As mentioned in the cDNA preparation method, there is no de-acylation process. The aminoacylated tRNA might not be successfully ligated to 3' adaptor. There might be some bias toward tRNAs with different aminoacylation levels. It is better to compare the sequencing result with or without an individual de-acylation process.

We apologise for the brevity of the *Methods* section. Instead of referring to our prior work, we now include all relevant methodological details for tRNA sequencing library preparation. tRNAs were indeed deacylated prior to library construction by incubating total RNA in 75 mM Tris-HCl, pH=9.0 at 37°C for 45 min. In response to the reviewer's question #3 about aminoacylation levels, we now also include tRNA charging measurements in embryos collected before (4 hpf) and after (10 hpf) the onset of gastrulation. These new data in Figure 4D show that tRNA charging levels are comparably high among all tRNA anticodon families (80 - 90%), arguing against potential bias in our tRNA abundance measurements stemming from differential aminoacylation.

7. As shown in figure 1C, the reads of some tRNAs, such as Tyr/Trp/Lys, are highly truncated at 5'end, which indicates a high level of modification at the truncated site. It is unclear how the modification data was calculated from these truncated reads of these tRNA isotypes.

There is indeed a high level of premature cDNA synthesis termination in tRNA-Lys-UUU and tRNA-Tyr-GUA. We previously showed that this is due to the bulky ms²t⁶A modified nucleoside at position 37 in the nuclear-encoded tRNA-Lys-UUU, and the stretch of two consecutive m^{2,2}G nucleosides at positions 26 and 27 in the nuclear-encoded tRNA-Tyr-GUA. Both of these highly rare and unusual modifications pose an insurmountable obstacle to TGIRT even in our optimised RT protocol (Behrens *et al.*, 2021). Since we calculate modification fractions based on misincorporation rates alone, we do not estimate modification stoichiometries for these rare remaining RT-blocking modifications. We would like to emphasise that these stops do not affect our tRNA abundance measurements because the truncated cDNA fragments derived from them are sufficiently long (39–56 nt) to be unambiguously aligned with our computational pipeline.

8. The changes of whole amount of tRNAs were not mentioned. The up-/down-regulated tRNAs are classified by the relative proportion among tRNAs. If the whole tRNA proportion in total RNA is elevated during embryogenesis, some up-regulated genes might be underestimated. It is better to normalize the tRNA expression to the spike tRNA or total RNA.

We appreciate the reviewer's point and in our view the suggested analysis is not feasible with the assays and data currently available to us. We performed relative differential expression analysis because we wanted to establish whether changes in the relative abundance of tRNA with specific codons account for the codon-dependent destabilisation of a maternal mRNA subset at the onset of gastrulation. We indeed spiked in a synthetic *E.coli* tRNA-Lys-UUU during tRNA library preparation, but the transcript was added after total RNA isolation. Normalisation to this spike-in would therefore not account for differences in total RNA content during embryogenesis, as total RNA levels can vary substantially with cell size, cell cycle stage, and embryonic developmental stage (Coate & Doyle, 2015; Piko & Clegg, 1982). Deducing changes in the total amount of tRNAs is challenging for similar reasons: normalisation of tRNA signal to 5S rRNA and/or 5.8 rRNA after total RNA staining (employed by Chen *et al.* (2021)) is difficult to interpret because the onset of 5.8S rRNA transcription during zebrafish embryogenesis is unknown. In addition, 5S rRNA is also transcribed by Pol III and subject to MAF1 repression (Canella *et al.*, 2010). Because of these complexities, we have refrained from analysing total tRNA levels from the SYBR Gold-stained gels we now provide as a loading control for the northern blots in Figure 2E. To what extent total tRNA levels change during zebrafish embryogenesis will be best addressed in future studies that determine the total RNA content in individual cells from different developmental stages similarly to prior work for early mouse embryos (Piko & Clegg, 1982). Such studies will likely be highly challenging as they would require methods for efficient cell dissociation that do not perturb cell physiology.

9. Line 255, the citation "Most ribosomes in the early zebrafish embryo are in a translationally inactive state, and global translation gradually increases from 6 hpf onwards (Leesch *et al.*, 2023)" is not accurate. In Leesch's paper (Fig. 1c), polysome fraction obviously increased at the 2 hpf (64-cell) stage compared to the 1 hpf (4-cell) stage; actually, polysome fraction slightly increased from the egg to 1 hpf.

We appreciate the reviewer's point and we have changed the wording of this section in the manuscript (lines 229 to 233) to more accurately reflect the conclusions by Leesch *et al.* (2023). As the reviewer points out, Figure 1C of that paper indeed shows a gradual increase in polysome to 80S ratios as embryos develop. However, this increase was only statistically significant at 6 and 24 hpf. The authors accordingly state: "We found that ribosomes in the egg and in the one hour post-fertilization (hpf) embryo were almost exclusively present as monosomes, whereas the polysome fraction started to increase from 3–6hpf onwards (Fig. 1b,c). As an orthogonal approach, we calculated translational efficiency values based on ribosome-protected mRNA fragments over the course of zebrafish embryogenesis, using published ribosome profiling and RNA-seq datasets. We observed that translational efficiency increases over the course of embryogenesis, with a clear increase at 6hpf (Fig. 1d)."

Minor:

10. Line 4, "Developmental control is then handed over to the zygote ---" is not correct. It would be "--- to the zygotic genome control ---".

We have amended the text to read "*Developmental control is then handed over to the zygotic genome*".

11. Fig. 1A, "Maternal mRNA demand vs maternal tRNA supply" should be "Maternal mRNA demand and maternal tRNA supply"; same change for "Zygotic mRNA demand vs zygotic tRNA supply".

We have amended the schematic by leaving out "vs".

12. Fig. 1B, E and F, The title of the Y-axis "Proportion ---" should be "Proportion of ---".

Thank you, the axis titles have now been corrected throughout the manuscript.

13. Lines 148-150, "The decrease in the proportion of mt-tRNAs at the shield and bud (6 and 10 hpf, respectively) stages could indicate a relative increase in nuclear-encoded tRNA levels throughout MZT.". This may also be explained by a decrease in the number of mitochondria per cell as embryonic development progresses.

We thank the reviewer for pointing out this alternative explanation; we have now included it in the main text.

14. Line 180, "--sequences; (Figure 2A).", in which the semicolon should be deleted.

The text has been corrected.

15. Line 193, "the effect size for ---", in which the term "effective size" is inappropriate.

We have now rephrased to "*magnitude of expression changes*" to avoid ambiguity.

16. Figure 4a, some codons do not have correspondent tRNAs with exact anticodons, and codons are usually decoded in a wobble manner, then only the isotopes but not the isoacceptors could be correlated to the codon DT or stabilizing/destabilizing codons. The

There are indeed only 45 tRNA anticodon families (also known as isoacceptors) that decode the 61 sense codons in zebrafish. This is common as all organisms encode fewer than 61 tRNA anticodons (Marck & Grosjean, 2002). The rules of decoding the 61 codons with these smaller tRNA sets in eukaryotes are *well-established*, allowing us to correlate tRNA anticodon levels with specific properties of their cognate codons. For example, eight eukaryotic tRNA anticodon

families contain an adenosine at the wobble position (34), which is universally deaminated to inosine (I). This modification allows tRNAs with ANN anticodons to decode their cognate U- and C-ending synonymous codons, as the corresponding GNN tRNA anticodons are not encoded by eukaryotes genomes (e.g. the UCU and UCC codons for Ser are both decoded by tRNA-Ser-AGA since tRNA-Ser-GGA does not exist). For these eight ANN tRNA anticodon families, we have followed the nomenclature in Crick (1966) and denoted pairing with their cognate C-ending codon as “Watson-Crick” and with their cognate U-ending codons as “wobble” in Figure 4A. Similarly, a set of eight tRNA anticodon families with GNN anticodons decode both C- and U-ending synonymous codons because their ANN tRNA counterparts are missing. We have denoted this set of wobble instances with asterisks in Figure 4A, and we find no significant differences in the decoding rates of codons read via Watson-Crick or wobble pairing in zebrafish embryos we measured by ribosome profiling (Figure EV4I).

17. Figure 6c "sphere 6hpf" label wrong (6 hpf is actually equivalent to the shield stage)

Thank you for pointing out this error, which we have corrected in the revised manuscript.

Referee #3:

Pinto et al 2024

The manuscript by Pinto et al sets out to quantitatively profile tRNA repertoire in zebrafish embryos during the maternal to zygotic transition. Traditionally tRNA quantification has been tricky, and to address many of the challenges associated with such analysis, Pinto et al employ a recently developed method called mim-tRNAseq. This method is combined with ribosome profiling to establish the functional impact of tRNA pools on mRNA decoding rates and stability. In addition to the general utility of the generated data sets, the authors make several significant insights. First, they find that bulk of zygotic tRNA expression is established as the embryo enters gastrulation (~5 hpf), rather than when bulk mRNA transcription initiates at ~3 hpf. Previous studies have suggested that a subset of maternal mRNAs is degraded in a translation-dependent manner based on their codon composition during the zebrafish MZT, and in this study the authors extend on this observation by demonstrating that it is maternal mRNAs with a codon content poorly matched to tRNA pools that are destabilized by slower decoding upon increases in protein synthesis during gastrulation. Mechanistically they suggest tRNA gene derepression is temporally coordinated by target of rapamycin complex 1 (TORC1) mediated inactivation of the RNA Polymerase III (Pol III) repressor Maf1a/b through phosphorylation, although this model is not experimentally confirmed.

Overall this study is a valuable contribution to the scientific literature that will be of use to those studying both tRNA regulation and early development. I have only minor concerns.

We thank the reviewer very much for their enthusiastic support and appreciation of the advances offered by our work.

The authors note that they are using GRCz11 reference genome, but this reference genome is known to have gaps and possible duplications, particularly near centromeres as well as on chromosome 4 where many tRNA clusters are located. Similarly, the reference to the number of zebrafish tRNA genes (Chan and Lowe 2016) in the introduction also almost certainly relies on an incomplete genome. It could be helpful for the authors to make note of this point and any impacts on analysis.

We thank the reviewer for raising this important point. We used the GRCz11 reference-based set of tRNA gene predictions as it is the only one that is currently available to the community through the Genomic tRNA Database (GtRNAdb). To draw attention to the incomplete nature of this current assembly, we have now included the following text in the *Discussion*:

“We note that tRNA gene predictions in the GtRNAdb (Chan & Lowe, 2016) are only available for the current GRCz11 build of the zebrafish genome, which however has known gaps and possible duplications, particularly on the tRNA-rich chromosome 4.”

In panel 2 D and H are relatively difficult to read, with it being particularly hard to discriminate between the three faintly shaded colors.

We have now decreased the transparency of the colour boxes and increased the width of the boxes in the boxplots from Figure 2D and Figure 2h to improve readability.

In Panel 2F it is somewhat difficult to discriminate between the colors for Ser and Ala on such small circles

We have now changed the colours for Ser and Ala in Figure 2F to a more contrasting set to improve plot readability.

In Figure 6E it is confusing to have values that are differently normalized on the same graph. Time course and Rapa treated vs DMSO should be broken out if they involve distinct normalization.

Thank you for pointing this out. We have now split the time-course and rapamycin treatment data in separate panels in Figure 6 to reflect their distinct normalisation.

The authors convincingly show that inhibition of Torc1 prevents maf1a/b phosphorylation at least to some extent at some sites. While this seems like a plausible mechanism for regulating tRNA transcription via polIII, the authors don't really make that final connection experimentally. The case would be further strengthened by data showing Rapa leads to a change in the timing of bulk tRNA transcription if such experiments are possible.

We appreciate the reviewer's point, and we have highlighted the technical challenges of using rapamycin to probe the role of TORC1 in the timing of bulk tRNA transcription in our response to Reviewer#1, point 8. We would like to emphasize that there is ample experimental evidence from both yeast and mammalian experimental systems that MAF1 phosphorylation by TORC1 renders it unable to repress Pol III (e.g Michels *et al.*, 2010, Towpik *et al.*, 2008), and our own recent work also established a link between TORC1 and selective tRNA transcription during human stem cell differentiation by MAF1 depletion with CRISPR interference (Gao *et al.*, 2024).

At author's discretion:

There is also good DNA methylation data for early zebrafish development, it could be worth considering looking at methylation status of tRNAs with respect to their expression, could this be part of the explanation for delayed transcription?

We thank the reviewer for this insightful suggestion, and we have now analysed the DNA methylation status of predicted tRNA genes in embryos from blastula phase (1000-cell, 3 hpf) and gastrula phase (germ ring, 5.7 hpf) using whole-genome bisulfite sequencing datasets from Jiang *et al* (2013) (new data in Figure EV1D and Figure EV5B). Interestingly, we found that a large proportion of CpG in predicted zebrafish tRNA genes are highly methylated in both 1000-cell (3 hpf) and germ ring (5.7 hpf) embryos (Figure EV1D). CpG methylation was near-complete for tRNA genes encoded in large clusters on chromosomes 4 (5227 tRNA genes, 60%), 8 (353 genes, 4%) and 3 (261 genes, 3%). These data are consistent with our recent analysis of human tRNA expression dynamics during cellular differentiation, where we found that inactive tRNA genes (not occupied by Pol III in ChIP-Seq datasets) are marked by near-complete CpG methylation (Gao *et al.*, 2024). Of note, CpG methylation at tRNA genes encoding upregulated tRNA transcripts was significantly lower than that at other tRNA genes already at 3 hpf (Figure EV5B). This suggests that the onset of selective tRNA transcription is not linked to a change in DNA methylation status at the activated genes.

References:

Bazzini AA, Del Viso F, Moreno-Mateos MA, Johnstone TG, Vejnar CE, Qin Y, Yao J, Khokha MK, Giraldez AJ (2016) Codon identity regulates mRNA stability and translation efficiency during the maternal-to-zygotic transition. *Embo j* 35: 2087-2103

Bazzini AA, Lee MT, Giraldez AJ (2012) Ribosome profiling shows that miR-430 reduces translation before causing mRNA decay in zebrafish. *Science* 336: 233-237

Begik O, Diensthuber G, Liu H, Delgado-Tejedor A, Kontur C, Niazi AM, Valen E, Giraldez AJ, Beaudoin JD, Mattick JS *et al* (2023) Nano3P-seq: transcriptome-wide analysis of gene expression and tail dynamics using end-capture nanopore cDNA sequencing. *Nat Methods* 20: 75-85

Behrens A, Rodschinka G, Nedialkova DD (2021) High-resolution quantitative profiling of tRNA abundance and modification status in eukaryotes by mim-tRNAseq. *Mol Cell* 81: 1802-1815.e1807

Benjamin D, Colombi M, Moroni C, Hall MN (2011) Rapamycin passes the torch: a new generation of mTOR inhibitors. *Nat Rev Drug Discov* 10: 868-880

Bhat P, Cabrera-Quio LE, Herzog VA, Fasching N, Pauli A, Ameres SL (2023) SLAMseq resolves the kinetics of maternal and zygotic gene expression during early zebrafish embryogenesis. *Cell Rep* 42: 112070

Canella D, Praz V, Reina JH, Cousin P, Hernandez N (2010) Defining the RNA polymerase III transcriptome: Genome-wide localization of the RNA polymerase III transcription machinery in human cells. *Genome Res* 20: 710-721

Chen L, Xu W, Liu K, Jiang Z, Han Y, Jin H, Zhang L, Shen W, Jia S, Sun Q *et al* (2021) 5' Half of specific tRNAs feeds back to promote corresponding tRNA gene transcription in vertebrate embryos. *Sci Adv* 7: eabh0494

Coate JE, Doyle JJ (2015) Variation in transcriptome size: are we getting the message? *Chromosoma* 124: 27-43

Crick FH (1966) Codon--anticodon pairing: the wobble hypothesis. *J Mol Biol* 19: 548-555

Diedenhofen B, Musch J (2015) cocor: a comprehensive solution for the statistical comparison of correlations. *PLoS One* 10: e0121945

Ehrlich R, Davyt M, Lopez I, Chalar C, Marin M (2021) On the Track of the Missing tRNA Genes: A Source of Non-Canonical Functions? *Front Mol Biosci* 8: 643701

Evans ME, Clark WC, Zheng G, Pan T (2017) Determination of tRNA aminoacylation levels by high-throughput sequencing. *Nucleic Acids Res* 45: e133

Gao L, Behrens A, Rodschinka G, Forcelloni S, Wani S, Strasser K, Nedialkova DD (2024) Selective gene expression maintains human tRNA anticodon pools during differentiation. *Nat Cell Biol* 26: 100-112

Jiang L, Zhang J, Wang JJ, Wang L, Zhang L, Li G, Yang X, Ma X, Sun X, Cai J *et al* (2013) Sperm, but not oocyte, DNA methylome is inherited by zebrafish early embryos. *Cell* 153: 773-784

Leesch F, Lorenzo-Orts L, Pribitzer C, Grishkovskaya I, Roehsner J, Chugunova A, Matzinger M, Roitinger E, Belačić K, Kandolf S *et al* (2023) A molecular network of conserved factors keeps ribosomes dormant in the egg. *Nature* 613: 712-720

Lewis CJT, Xie L, Bhandarkar S, Jin D, Abdallah KS, Draycott AS, Chen Y, Thoreen CC, Gilbert WV (2024) Quantitative profiling of human translation initiation reveals regulatory elements that potently affect endogenous and therapeutically modified mRNAs. *bioRxiv*: 2024.2002.2028.582532

Locati MD, Pagano JF, Ensink WA, van Olst M, van Leeuwen S, Nehrdich U, Zhu K, Spaink HP, Girard G, Rauwerda H *et al* (2017) Linking maternal and somatic 5S rRNA types with different sequence-specific non-LTR retrotransposons. *RNA* 23: 446-456

Love MI, Huber W, Anders S (2014) Moderated estimation of fold change and dispersion for RNA-seq data with DESeq2. *Genome Biol* 15: 550

Marck C, Grosjean H (2002) tRNomics: analysis of tRNA genes from 50 genomes of Eukarya, Archaea, and Bacteria reveals anticodon-sparing strategies and domain-specific features. *RNA* 8: 1189-1232

Medina-Muñoz SG, Kushawah G, Castellano LA, Diez M, DeVore ML, Salazar MJB, Bazzini AA (2021) Crosstalk between codon optimality and cis-regulatory elements dictates mRNA stability. *Genome Biol* 22: 14

Michels AA, Robitaille AM, Buczynski-Ruchonnet D, Hodroj W, Reina JH, Hall MN, Hernandez N (2010) mTORC1 directly phosphorylates and regulates human MAF1. *Mol Cell Biol* 30: 3749-3757

Mishima Y, Han P, Ishibashi K, Kimura S, Iwasaki S (2022) Ribosome slowdown triggers codon-mediated mRNA decay independently of ribosome quality control. *Embo j* 41: e109256

Mishima Y, Tomari Y (2016) Codon Usage and 3' UTR Length Determine Maternal mRNA Stability in Zebrafish. *Mol Cell* 61: 874-885

Pálffy M, Schulze G, Valen E, Vastenhouw NL (2020) Chromatin accessibility established by Pou5f3, Sox19b and Nanog primes genes for activity during zebrafish genome activation. *PLoS Genet* 16: e1008546

Piko L, Clegg KB (1982) Quantitative changes in total RNA, total poly(A), and ribosomes in early mouse embryos. *Dev Biol* 89: 362-378

Rappol T, Waldl M, Chugunova A, Hofacker IL, Pauli A, Vilardo E (2024) tRNA expression and modification landscapes, and their dynamics during zebrafish embryo development. *Nucleic Acids Res*

Reimão-Pinto MM, Castillo-Hair SM, Seelig G, Schier AF (2023) The regulatory landscape of 5' UTRs in translational control during zebrafish embryogenesis. *bioRxiv*: 2023.2011.2023.568470

Saxton RA, Sabatini DM (2017) mTOR Signaling in Growth, Metabolism, and Disease. *Cell* 168: 960-976

Strayer EC, Krishna S, Lee H, Vejnar C, Beaudoin J-D, Giraldez AJ (2023) NaP-TRAP, a novel massively parallel reporter assay to quantify translation control. *bioRxiv*: 2023.2011.2009.566434

Subtelny AO, Eichhorn SW, Chen GR, Sive H, Bartel DP (2014) Poly(A)-tail profiling reveals an embryonic switch in translational control. *Nature* 508: 66-71

Suzuki T (2021) The expanding world of tRNA modifications and their disease relevance. *Nat Rev Mol Cell Biol* 22: 375-392

Towpik J, Graczyk D, Gajda A, Lefebvre O, Boguta M (2008) Derepression of RNA polymerase III transcription by phosphorylation and nuclear export of its negative regulator, Maf1. *J Biol Chem* 283: 17168-17174

Dear Dr Nedialkova,

Thank you for submitting a revised version of your manuscript. Your study has now been seen by all original referees, who find that their previous concerns have been addressed and now recommend publication of the manuscript. There remain only a few mainly editorial points that have to be addressed before I can extend formal acceptance of the manuscript:

- Please make sure that you add all the all funding sources to eJP: the International Max Planck Research School for Molecular Life Sciences; SFB 1035; The initials "D.D.N." should be removed from the list of the grant numbers in eJP
- Please rename the Conflict of Interest section into "Disclosure and Competing Interests Statement", in accordance with our updated Guide to Authors
- As we are switching from a free-text author contribution statement towards a more formal statement based on Contributor Role Taxonomy (CRediT) terms, please remove the present Author Contribution section and instead specify each author's contribution(s) directly in the Author Information page of our submission system during upload of the final manuscript. See <https://casrai.org/credit/> for more information.
- Please rename EV1-EV5 to Dataset EV1-EV5 with the legends removed from ms file and uploaded as a separate tab in each Excel file, with the corresponding callouts.
- Please reorganize the SD files - Source data files need to be saved in a scheme one figure/folder and then uploaded as .zip files. E.g. all the Source data files for figure 1 need to be saved in a single folder and this needs to be zipped and then uploaded as "SD figure 1.zip" file.
- Please provide suggestions for a short 'blurb' text prefacing and summing up the conceptual aspect of the study in two sentences (max. 250 characters), followed by 3-5 one-sentence 'bullet points' with brief factual statements of key results of the paper; they will form the basis of an editor-written 'Synopsis' accompanying the online version of the article. Please also provide an altered synopsis image, making sure that the aspect ratio conforms to our website's format - it should be exactly 550 pixels wide and between 300-600 pixels high.
- Please add the specific URLs for GSE241755 and MSV000094166 datasets to the data availability statement.

Our data editors have also raised several points regarding the figure legends.

1. Please note that the box plots need to be defined in terms of minima, maxima, in the legends of figures 1d; 2d, h; 4b-c; EV 1b.
 2. Please note that the box plots need to be defined in terms of minima, maxima, centre, bounds of box and whiskers, and percentile in the legends of figures EV 4i.
 3. Please note that information related to n is missing in the legends of figures 1d; 2d, h; EV 1d; EV 2e; EV 4i; EV 5b; EV 6d, f-g.
 4. Please note that n=2 in figure 1b. We do not require an additional biological replicate here but do not think that statistical analysis makes sense with n=2.
 5. Although 'n' is provided, please describe the nature of entity for 'n' in the legends of figures 4b-c; 6d; EV 4h.
 6. Please note that the error bars are not defined in the legends of figures 6d; EV 6d, f-g.
 7. Please note that the exact p values are not provided in the legends of figures EV 4b-c, f-h; EV 5b; EV 6d, f-h.
 8. Please indicate the statistical test used for data analysis in the legends of figures 2c; 3c, e; EV 4e.
- Since we can only accommodate up to 5 Expanded View Figures that will be directly included in the HTML version, at least one of the currently 16 EV Figures has to be turned into an "Appendix Figures" or fused with another EV figure.
 - Please adjust the order of the manuscript sections: Title page with complete author information, Abstract, Keywords, Introduction, Results, Discussion, Methods, Data Availability Section, Acknowledgements, Disclosure and Competing Interests Statement, References, Main figure legends, Tables, Expanded Figure Legends.
 - One of the Author email bounced: Andrew Behrens - behrens@biochem.mpg.de
Please update if possible.

With best regards,

Cornelius Schneider

Cornelius Schneider, PhD
Editor | The EMBO Journal
c.schneider@embojournal.org

We realize that it is difficult to revise to a specific deadline. In the interest of protecting the conceptual advance provided by the work, we recommend a revision within 3 months (12th Dec 2024). Please discuss the revision progress ahead of this time with the editor if you require more time to complete the revisions. Use the link below to submit your revision:

Referee #1:

I appreciate the authors efforts in revising their manuscript and sufficiently addressing all my queries. I recommend acceptance of their work for publication.

Referee #2:

Most of my previous criticisms or concerns have been adequately addressed in this revised version. It is recommended for acceptance for publication.

Referee #3:

All of my concerns have been addressed.

All editorial and formatting issues were resolved by the authors.

Dear Prof. Nedialkova,

I am pleased to inform you that your manuscript has been accepted for publication in the EMBO Journal.

Yours sincerely,

Cornelius Schneider, PhD
Editor
The EMBO Journal
c.schneider@embojournal.org
